# Block Coordinate Descent Methods for Optimization under J-Orthogonality Constraints with Applications

## Abstract

The J-orthogonal matrix, also referred to as the hyperbolic orthogonal matrix, is a class of special orthogonal matrix in hyperbolic space, notable for its advantageous properties. These matrices are integral to optimization under J-orthogonal constraints, which have widespread applications in statistical learning and data science. However, addressing these problems is generally challenging due to their non-convex nature and the computational intensity of the constraints. Currently, algorithms for tackling these challenges are limited. This paper introduces **JOBCD**, a novel Block Coordinate Descent method designed to address optimizations with J-orthogonality constraints. We explore two specific variants of **JOBCD**: one based on a Gauss-Seidel strategy (**GS-JOBCD**), the other on a variance-reduced and Jacobi strategy (**VR-J-JOBCD**). Notably, leveraging the parallel framework of a Jacobi strategy, **VR-J-JOBCD** integrates variance reduction techniques to decrease oracle complexity in the minimization of finite-sum functions. For both **GS-JOBCD** and **VR-J-JOBCD**, we establish the oracle complexity under mild conditions and strong limit-point convergence results under the Kurdyka-Lojasiewicz inequality. To demonstrate the effectiveness of our method, we conduct experiments on hyperbolic eigenvalue problems, hyperbolic structural probe problems, and the ultrahyperbolic knowledge graph embedding problem. Extensive experiments using both real-world and synthetic data demonstrate that **JOBCD** consistently outperforms state-of-the-art solutions, by large margins.

## 1 Introduction

A matrix $\mathbf{X} \in \mathbb{R}^{n \times n}$ is a J-orthogonal matrix if $\mathbf{X}^\top \mathbf{J} \mathbf{X} = \mathbf{J}$, where $\mathbf{J} = \begin{bmatrix} \mathbf{I}_p & \mathbf{0} \\ \mathbf{0} & -\mathbf{I}_{n-p} \end{bmatrix}$, and $\mathbf{I}_p$ is a $p \times p$ identity matrix. Here, $\mathbf{J} \in \mathbb{R}^{n \times n}$ is the signature matrix with signature $(p, n-p)$. In this paper, we mainly focus on the following optimization problem under J-orthogonality constraints:

$$\min_{\mathbf{X} \in \mathbb{R}^{n \times n}} f(\mathbf{X}) \triangleq \tfrac{1}{N} \sum_{i=1}^N f_i(\mathbf{X}), \text{ s.\,t. } \mathbf{X}^\top \mathbf{J} \mathbf{X} = \mathbf{J}. \tag{1}$$

Here, $f(\mathbf{X})$ could have a finite-sum structure, each component function $f_i(\mathbf{X})$ is assumed to be differentiable, and $N$ is the number of data points. For brevity, the J-orthogonality constraint $\mathbf{X}^\top \mathbf{J} \mathbf{X} = \mathbf{J}$ in Problem (1) is rewritten as $\mathbf{X} \in \mathcal{J}$.

We impose the following assumptions on Problem (1) throughout this paper. (𝔸-i) For any matrices $\mathbf{X}$ and $\mathbf{X}^+$, we assume $f_i : \mathbb{R}^{n \times n} \mapsto \mathbb{R}$ is continuously differentiable for some symmetric positive semidefinite matrix $\mathbf{H} \in \mathbb{R}^{nn \times nn}$ that:

$$f_i(\mathbf{X}^+) \le f_i(\mathbf{X}) + \langle \mathbf{X}^+ - \mathbf{X}, \nabla f_i(\mathbf{X}) \rangle + \tfrac{1}{2} \|\mathbf{X}^+ - \mathbf{X}\|_{\mathbf{H}}^2, \tag{2}$$

for all $i \in [N]$, where $\|\mathbf{H}\| \le L_f$ for some constant $L_f > 0$ and $\|\mathbf{X}\|_{\mathbf{H}}^2 \triangleq \text{vec}(\mathbf{X})^\top \mathbf{H} \text{vec}(\mathbf{X})$. Importantly, the function $f(\mathbf{X}) = \tfrac{1}{2} \text{tr}(\mathbf{X}^\top \mathbf{C} \mathbf{X} \mathbf{D}) = \tfrac{1}{2} \|\mathbf{X}\|_{\mathbf{H}}^2$ with $\mathbf{H} = \mathbf{D} \otimes \mathbf{C}$ satisfies the equality $\forall \mathbf{X}, \mathbf{X}^+, f(\mathbf{X}^+) = f(\mathbf{X}) + \langle \mathbf{X}^+ - \mathbf{X}, \nabla f(\mathbf{X}) \rangle + \tfrac{1}{2} \|\mathbf{X}^+ - \mathbf{X}\|_{\mathbf{H}}^2$ in (2), where $\mathbf{C} \in \mathbb{R}^{n \times n}$ and $\mathbf{D} \in \mathbb{R}^{n \times n}$ are arbitrary symmetric matrices. (𝔸-ii) For any matrices $\mathbf{X}$ and $\mathbf{X}^+$, we assume that: $\|\nabla f_i(\mathbf{X}) - \nabla f_i(\mathbf{X}^+)\|_{\mathsf{F}} \le L_f \|\mathbf{X} - \mathbf{X}^+\|_{\mathsf{F}}$ for all $i \in [N]$ and

$L_f$ is mentioned in ($\mathbb{A}$-i). ($\mathbb{A}$-iii) The function $f_i(\mathbf{X})$ is coercive for all $i \in N$, that is, $\lim_{\|\mathbf{X}\|_{\mathsf{F}} \to \infty} f_i(\mathbf{X}) = \infty$, $\forall i$.

Problem (1) defines an optimization framework that is fundamental to a wide range of models in statistical learning and data science, including hyperbolic eigenvalue problem [6; 42; 39], hyperbolic structural probe problem [20; 7], and ultrahyperbolic knowledge graph embedding [49]. Additionally, it is closely related to machine learning in hyperbolic spaces, including Lorentz model learning [34; 51; 8] and ultrahyperbolic neural networks [27; 56; 41]. It also intersects with hyperbolic linear algebra [3; 21], addressing problems such as the indefinite least squares problem, hyperbolic QR factorization, and indefinite polar decomposition.

## 1.1 Related Work

▶ **Block Coordinate Descent Methods**. Block Coordinate Descent (BCD) is a well-established iterative algorithm that sequentially minimizes along block coordinate directions. Its simplicity and efficiency have led to its widespread adoption in structured convex applications [36]. Recently, BCD has gained traction in non-convex problems due to its robust optimality guarantees and/or excellent empirical performance in areas including optimal transport [22], matrix optimization [12], fractional minimization [54], deep neural networks [5; 55; 31], federated learning[46], black-box optimization [4], and optimization with orthogonality constraints [52; 14]. To our knowledge, this is the first application of BCD methods to optimization under J-orthoginality constraints, with a focus on analyzing their theoretical guarantees and empirical efficacy.

▶ **Minimizing Smooth Functions under J-Orthogonality Constraints**. The J-orthogonal matrix belongs to a subset of generalized orthogonal matrices [16; 35; 23]. However, projecting onto the J-orthogonality constraint poses challenges, complicating the extension of conventional optimization algorithms to address optimization problems under these constraints [1; 16]. This contrasts with computing orthogonal projections using methods such as polar or SVD decomposition, or approximating them via QR factorization. Existing methods for addressing Problem (1) can be categorized into three classes. (***i***) CS-Decomposition Based Methods. These approaches involve parameterizing four orthogonal matrices (as described in Proposition 2.2) and subsequently minimizing a smooth function over these matrices in an alternating fashion. The involvement of $3 \times 3$ block matrices makes the implementation of these methods very challenging. Consequently, the work of [49] focuses on optimizing a reduced subspace of the CS decomposition parameters, albeit at the expense of losing some degrees of freedom. (***ii***) Unconstrained Multiplier Correction Methods [47; 48; 13; 14]. These methods leverage the symmetry and explicit closed-form expression of the Lagrangian multiplier at the first-order optimality condition. Consequently, they address an unconstrained problem, resulting in efficient first-order infeasible approaches. (***iii***) Alternating Direction Method of Multipliers [19]. This method reformulates the original problem into a bilinear constrained optimization problem by introducing auxiliary variables. It employs dual variables to handle bilinear constraints, iteratively optimizing primal variables while keeping other primal and dual variables fixed, and using a gradient ascent strategy to update the dual variables. This approach has become widely adopted for solving general nonconvex and nonsmooth composite optimization problems. Notably, all the aforementioned methods solely identify critical points of Problem (1).

▶ **Finite-Sum Problems via Stochastic Gradient Descent**. The finite-sum structure is prevalent in machine learning and statistical modeling, facilitating decomposition into smaller, more manageable components. This property is advantageous for developing efficient algorithms for large-scale problems, such as Stochastic Gradient Descent (SGD). Reducing variance is crucial in SGD because it can lead to more stable and faster convergence. Various techniques, such as mini-batch SGD, momentum methods, and variance reduction methods like SAGA [10], SVRG [25], SARAH [33], SPIDER [11; 43], SNVRG [57], and PAGE [30], have been developed to address this issue. Additionally, SGD for minimizing composite functions has also been investigated by the authors [15; 24; 29].

## 1.2 Contributions

This paper makes the following contributions. (***i***) Algorithmically: We introduce the **JOBCD** algorithm, a novel Block Coordinate Descent method specifically designed to tackle optimizations constrained by J-orthogonality. We explore two specific variants of

**JOBCD**, one based on a Gauss-Seidel strategy (**GS-JOBCD**), the other on a variance-reduced and Jacobi strategy (**VR-J-JOBCD**). Notably, **VR-J-JOBCD** incorporates a variance-reduction technique into a parallel framework to reduce oracle complexity in the minimization of finite-sum functions (See Section 2). (***ii***) Theoretically: We provide comprehensive optimality and convergence analyses for both algorithms (see Sections 3 and 4). (***iii***) Empirically: Extensive experiments across hyperbolic eigenvalue problems, structural probe problems, and ultrahyperbolic knowledge graph embedding, using both real-world and synthetic data, consistently show the significant superiority of **JOBCD** over state-of-the-art solutions (see Section 6).

## 2 The Proposed **JOBCD** Algorithm

This section proposes **JOBCD** for solving optimization problems under J-orthogonality constraints in Problem (1), which is based on randomized block coordinate descent. Two variants of **JOBCD** are explored, one based on a Gauss-Seidel strategy (**GS-JOBCD**), the other on a variance-reduced and Jocobi strategy (**VR-J-JOBCD**).

**Notations**. We define $[n] \triangleq \{1, 2, \ldots, n\}$. We denote $\Omega \triangleq \{\mathcal{B}_1, \mathcal{B}_2, \ldots, \mathcal{B}_{C_n^2}\}$ as all the possible combinations of the index vectors choosing 2 items from $n$ without repetition. For any $\mathtt{B} \in \Omega$, we define $\mathbf{U}_\mathtt{B} \in \mathbb{R}^{n \times 2}$ as $(\mathbf{U}_\mathtt{B})_{ji} = 1$ if $\mathtt{B}_i = j$, else 0 for all $j$ and $i$, leading to $\mathbf{U}_\mathtt{B}^\mathsf{T} \mathbf{X} = \mathbf{X}(\mathtt{B}, :) \in \mathbb{R}^{2 \times n}$. We denote $\mathcal{J}_\mathtt{B} \triangleq \{\mathbf{V} \mid \mathbf{V}^\mathsf{T} \mathbf{J}_{\mathtt{BB}} \mathbf{V} = \mathbf{J}_{\mathtt{BB}}\}$, where $\mathbf{J}_{\mathtt{BB}} \in \mathbb{R}^{2 \times 2}$ is the sub-matrix of $\mathbf{J}$ indexed by $\mathtt{B}$. Further notations are provided in Appendix A.1.

### 2.1 Gauss-Seidel Block Coordinate Descent Algorithm

This subsection describes the proposed **GS-JOBCD** algorithm. We consider Problem (1) with $N = 1$ only, without utilizing its finite-sum structure.

**GS-JOBCD** is an iterative algorithm that, in each iteration $t$, randomly and uniformly (with replacement) selects a coordinate $\mathtt{B}^t$ from the set $\Omega$ and then solves a small-sized subproblem. The row index $[n]$ of the decision variable $\mathbf{X}^t$ are separated to two sets $\mathtt{B}^t$ and $\mathtt{B}^{t^c}$, where $\mathtt{B}^t \in \Omega$ with $|\mathtt{B}^t| = 2$ is the working set and $\mathtt{B}^{t^c} = [n] \setminus \mathtt{B}^t$. For simplicity, we use $\mathtt{B}$ instead of $\mathtt{B}^t$. Following [52], we consider the following block coordinate update rule: $[\mathbf{X}^{t+1}(\mathtt{B}, :) = \mathbf{V} \mathbf{X}^t(\mathtt{B}, :)] \Leftrightarrow [\mathbf{X}^{t+1} = \mathbf{X}^t + \mathbf{U}_\mathtt{B}(\mathbf{V} - \mathbf{I})\mathbf{U}_\mathtt{B}^\mathsf{T} \mathbf{X}^t]$, where $\mathbf{V} \in \mathbb{R}^{2 \times 2}$ is some suitable matrix.

The following lemma illustrates matrix selection for enforcing J-orthogonality constraints via the update rule $\mathbf{X}^+ \Leftarrow \mathcal{X}_\mathtt{B}(\mathbf{V}) \triangleq \mathbf{X} + \mathbf{U}_\mathtt{B}(\mathbf{V} - \mathbf{I})\mathbf{U}_\mathtt{B}^\mathsf{T} \mathbf{X}$, and presents associated properties.

**Lemma 2.1.** *(Proof in Section C.1) For any $\mathtt{B} \in \Omega$, we define $\mathbf{X}^+ \triangleq \mathcal{X}_\mathtt{B}(\mathbf{V}) \triangleq \mathbf{X} + \mathbf{U}_\mathtt{B}(\mathbf{V} - \mathbf{I})\mathbf{U}_\mathtt{B}^\mathsf{T} \mathbf{X}$. We have: (**a**) If $\mathbf{V} \in \mathcal{J}_\mathtt{B}$ and $\mathbf{X} \in \mathcal{J}$, then $\mathbf{X}^+ \in \mathcal{J}$. (**b**) $\|\mathbf{X}^+ - \mathbf{X}\|_\mathsf{F}^2 \leq \|\mathbf{X}\|_\mathsf{F}^2 \cdot \|\mathbf{V} - \mathbf{I}\|_\mathsf{F}^2$. (**c**) $\|\mathbf{X}^+ - \mathbf{X}\|_\mathbf{H}^2 \leq \|\mathbf{V} - \mathbf{I}\|_\mathbf{Q}^2$ for all $\mathbf{Q} \succcurlyeq \underline{\mathbf{Q}} \triangleq (\mathbf{Z}^\top \otimes \mathbf{U}_\mathtt{B})^\top \mathbf{H}(\mathbf{Z}^\top \otimes \mathbf{U}_\mathtt{B})$, $\mathbf{Z} \triangleq \mathbf{U}_\mathtt{B}^\mathsf{T} \mathbf{X} \in \mathbb{R}^{k \times n}$.*

▶ **The Main Algorithm**. Using the above update rule, we consider the following iterative procedure: $\mathbf{X}^{t+1} \Leftarrow \mathcal{X}_\mathtt{B}^t(\bar{\mathbf{V}}^t)$, where $\bar{\mathbf{V}}^t \in \arg\min_\mathbf{V} f(\mathcal{X}_\mathtt{B}^t(\mathbf{V}))$. However, the resulting subproblem could be still difficult to solve. This inspires us to use sequential majorization minimization [37; 32] to address it. This technique iteratively constructs a surrogate function that upper-bounds the objective function, allowing for effective optimization and gradual reduction of the objective function. We derive:

$$
\begin{aligned}
f(\mathcal{X}_\mathtt{B}^t(\mathbf{V})) \quad &\overset{①}{\leq} \quad f(\mathbf{X}^t) + \tfrac{1}{2}\|\mathcal{X}_\mathtt{B}^t(\mathbf{V}) - \mathbf{X}^t\|_\mathbf{H}^2 + \langle \mathcal{X}_\mathtt{B}^t(\mathbf{V}) - \mathbf{X}^t, \nabla f(\mathbf{X}^t) \rangle \\
&\overset{②}{\leq} \quad f(\mathbf{X}^t) + \tfrac{1}{2}\|\mathbf{V} - \mathbf{I}\|_{\mathbf{Q}+\theta\mathbf{I}}^2 + \langle \mathbf{V} - \mathbf{I}, [\nabla f(\mathbf{X}^t)(\mathbf{X}^t)^\top]_{\mathtt{BB}} \rangle \triangleq \mathcal{G}(\mathbf{V}; \mathbf{X}^t, \mathtt{B}^t), \quad (3)
\end{aligned}
$$

where step ① uses Inequality (2); step ② uses Claim (**c**) of Lemma 2.1, $\theta \geq 0$ and the fact that $\langle \mathbf{U}_\mathtt{B}(\mathbf{V} - \mathbf{I})\mathbf{U}_\mathtt{B}^\mathsf{T} \mathbf{X}, \nabla f(\mathbf{X}) \rangle = \langle \mathbf{V} - \mathbf{I}, [\nabla f(\mathbf{X})\mathbf{X}^\top]_{\mathtt{BB}} \rangle$, and the choice of $\mathbf{Q} \in \mathbb{R}^{4 \times 4}$ that:

$$
\mathbf{Q} = \underline{\mathbf{Q}}, \text{ or } \mathbf{Q} = \varsigma\mathbf{I}_4, \text{ with } \|\underline{\mathbf{Q}}\| \leq \varsigma \leq L_f. \quad (4)
$$

Therefore, the function $\mathcal{G}(\mathbf{V}; \mathbf{X}^t, \mathtt{B}^t)$ becomes a majorization function of $f(\mathbf{X})$ at $\mathbf{X}^t \in \mathcal{J}$ for all $\mathtt{B}^t \in \Omega$. We can consider the following optimization problem to find $\bar{\mathbf{V}}^t$: $\bar{\mathbf{V}}^t \in \arg\min_\mathbf{V} \mathcal{G}(\mathbf{V}; \mathbf{X}^t, \mathtt{B}^t)$.

---

**Algorithm 1: GS-JOBCD**: Block Coordinate Descent Methods using a Gauss-Seidel Strategy for Solving Problem (1)

---

**Init.:** Set $\mathbf{X}^0$ to satisfy J-orthogonality constraints (e.g., via Hyperbolic CS Decomposition), $\theta$ in Inequality (3) (e.g., 1e-6).

**for** $t$ from 0 to $T$ **do**

    (S1) Choose a coordinate $\mathsf{B}^t$ with $|\mathsf{B}^t| = 2$ from the set $\Omega$ randomly and uniformly (with replacement) for the $t$-th iteration. Denote $\mathsf{B} = \mathsf{B}^t$.

    (S2) Choose a matrix $\mathbf{Q} \in \mathbb{R}^{4 \times 4}$ using Formula (4).

    (S3) Solve the following small-size subproblem globally.

$$
\begin{aligned}
\overline{\mathrm{V}}^t \quad &\in \quad \arg\min_{\mathbf{V} \in \mathcal{J}_{\mathsf{B}}} \ \tfrac{1}{2}\|\mathbf{V} - \mathbf{I}\|_{\mathbf{Q}+\theta\mathbf{I}}^2 + \langle \mathbf{V} - \mathbf{I}, [\nabla f(\mathbf{X}^t)(\mathbf{X}^t)^{\mathsf{T}}]_{\mathsf{BB}} \rangle + f(\mathbf{X}^t) \quad (5) \\
&= \quad \arg\min_{\mathbf{V} \in \mathcal{J}_{\mathsf{B}} \in \mathbb{R}^{2 \times 2}} \ \tfrac{1}{2}\|\mathbf{V}\|_{\dot{\mathbf{Q}}}^2 + \langle \mathbf{V}, \mathbf{P} \rangle + c \quad\quad\quad\quad (6)
\end{aligned}
$$

    where $\mathbf{P} \triangleq [\nabla f(\mathbf{X}^t)(\mathbf{X}^t)^{\mathsf{T}}]_{\mathsf{BB}} - \mathrm{mat}(\dot{\mathbf{Q}} \mathrm{vec}(\mathbf{I}_2))$, $\dot{\mathbf{Q}} = \mathbf{Q} + \theta\mathbf{I}$ and $c \triangleq f(\mathbf{X}^t) - \langle \mathbf{I}_2, [\nabla f(\mathbf{X}^t)(\mathbf{X}^t)^{\mathsf{T}}]_{\mathsf{BB}} \rangle + \tfrac{1}{2}\|\mathbf{I}\|_{\dot{\mathbf{Q}}}^2$ is a constant.

    (S4) $\mathbf{X}^{t+1}(\mathsf{B}, :) = \overline{\mathrm{V}}^t \mathbf{X}^t(\mathsf{B}, :)$

**end**

---

We summarize the proposed **GS-JOBCD** in Algorithm 1.

Although the J-orthogonality constraint typically has a sorted diagonal with $\mathrm{diag}(\mathbf{J}) \in \{-1, +1\}^n$, **GS-JOBCD** is also applicable to problems with more general constraints $\mathbf{X}^{\mathsf{T}}\mathbf{J}\mathbf{X} = \mathbf{J}$ where $\mathrm{diag}(\mathbf{J}) \in \{\pm1\}^n$ is unsorted.

▶ **Solving the Small-Sized Subproblem**. We now elaborate on how to find the global optimal solution of Problem (6). We notice that $\mathbf{V} \in \mathcal{J}_{\mathsf{B}} \triangleq \{\mathbf{V} \,|\, \mathbf{V}^{\mathsf{T}}\mathbf{J}_{\mathsf{BB}}\mathbf{V} = \mathbf{J}_{\mathsf{BB}}\}$, where $\mathbf{J}_{\mathsf{BB}} \in \{\left(\begin{smallmatrix} 1 & 0 \\ 0 & -1 \end{smallmatrix}\right), \left(\begin{smallmatrix} 1 & 0 \\ 0 & 1 \end{smallmatrix}\right), \left(\begin{smallmatrix} -1 & 0 \\ 0 & -1 \end{smallmatrix}\right)\}$. We now concentrate on the first case where $\mathbf{J}_{\mathsf{BB}} = \left(\begin{smallmatrix} 1 & 0 \\ 0 & -1 \end{smallmatrix}\right)$. The following proposition provides a strategy to decompose any J-orthogonal matrix.

**Proposition 2.2.** *(Hyperbolic CS Decomposition [40]) Let $\mathbf{V}$ be J-orthogonal with signature $(p, n-p)$. Assume that $n - p \leq p$. Then there exist vectors $\dot{c}, \dot{s} \in \mathbb{R}^{n-p}$ with $\dot{c} \odot \dot{c} - \dot{s} \odot \dot{s} = \mathbf{1}$, and orthogonal matrices $\mathbf{U}_1, \mathbf{V}_1 \in \mathbb{R}^{p \times p}$ and $\mathbf{U}_2, \mathbf{V}_2 \in \mathbb{R}^{(n-p) \times (n-p)}$ such that:* $\mathbf{V} = \left[\begin{smallmatrix} \mathbf{U}_1 & 0 \\ 0 & \mathbf{U}_2 \end{smallmatrix}\right]\left[\begin{smallmatrix} \mathrm{Diag}(\dot{c}) & 0 & \mathrm{Diag}(\dot{s}) \\ 0 & I_{p-(n-p)} & 0 \\ \mathrm{Diag}(\dot{s}) & 0 & \mathrm{Diag}(\dot{c}) \end{smallmatrix}\right]\left[\begin{smallmatrix} \mathbf{V}_1^{\mathsf{T}} & 0 \\ 0 & \mathbf{V}_2^{\mathsf{T}} \end{smallmatrix}\right].$

Applying Proposition 2.2 with $n = 2$, $p = 1$, and $\mathbf{U}_1 = \mathbf{U}_2 = \mathbf{V}_1 = \mathbf{V}_2 = \pm1$, $\tilde{c}^2 - \tilde{s}^2 = 1$ with $\tilde{c}, \tilde{s} \in \mathbb{R}$, we parametrize $\mathbf{V}$ as: $\mathbf{V} = \left(\begin{smallmatrix} \pm1 & 0 \\ 0 & \pm1 \end{smallmatrix}\right) \cdot \left(\begin{smallmatrix} \tilde{c} & \tilde{s} \\ \tilde{s} & \tilde{c} \end{smallmatrix}\right) \cdot \left(\begin{smallmatrix} \pm1 & 0 \\ 0 & \pm1 \end{smallmatrix}\right)$, where we denote $\tilde{s}$ as $\sinh(\mu)$, $\tilde{c}$ as $\cosh(\mu)$, and $\tilde{t}$ as $\tanh(\mu)$ for some $\mu \in \mathbb{R}$, for simplicity of notation. It is not difficult to show that Problem (6) reduces to the following one-dimensional search problem:

$$
\bar{\mu} \in \min_{\mu} \tfrac{1}{2} \mathrm{vec}(\mathbf{V})^{\mathsf{T}}\dot{\mathbf{Q}} \mathrm{vec}(\mathbf{V}) + \langle \mathbf{V}, \mathbf{P} \rangle, \ \mathrm{s.\,t.} \ \mathbf{V} \in \{\left(\begin{smallmatrix} \tilde{c} & \tilde{s} \\ \tilde{s} & \tilde{c} \end{smallmatrix}\right), \left(\begin{smallmatrix} \tilde{c} & -\tilde{s} \\ -\tilde{s} & \tilde{c} \end{smallmatrix}\right), \left(\begin{smallmatrix} -\tilde{c} & -\tilde{s} \\ \tilde{s} & \tilde{c} \end{smallmatrix}\right), \left(\begin{smallmatrix} \tilde{c} & -\tilde{s} \\ \tilde{s} & -\tilde{c} \end{smallmatrix}\right)\}. \quad (7)
$$

We apply a breakpoint search method to solve Problem (7). For simplicity, we provide an analysis only for the first case. A detailed discussion of all four cases can be found in Appendix Section B.1. For the case where $\mathbf{V} = \left(\begin{smallmatrix} \tilde{c} & \tilde{s} \\ \tilde{s} & \tilde{c} \end{smallmatrix}\right)$, Problem (7) reduces to the following problem:

$$
\min_{\tilde{c}, \tilde{s}} a\,\tilde{c} + b\,\tilde{s} + c\,\tilde{c}^2 + d\,\tilde{c}\,\tilde{s} + e\,\tilde{s}^2, \quad (8)
$$

where $a = \mathbf{P}_{11} + \mathbf{P}_{22}$, $b = \mathbf{P}_{12} + \mathbf{P}_{21}$, $c = \tfrac{1}{2}(\dot{\mathbf{Q}}_{11} + \dot{\mathbf{Q}}_{41} + \dot{\mathbf{Q}}_{14} + \dot{\mathbf{Q}}_{44})$, $d = \tfrac{1}{2}(\dot{\mathbf{Q}}_{21} + \dot{\mathbf{Q}}_{31} + \dot{\mathbf{Q}}_{12} + \dot{\mathbf{Q}}_{42} + \dot{\mathbf{Q}}_{13} + \dot{\mathbf{Q}}_{43} + \dot{\mathbf{Q}}_{24} + \dot{\mathbf{Q}}_{34})$, and $e = \tfrac{1}{2}(\dot{\mathbf{Q}}_{22} + \dot{\mathbf{Q}}_{32} + \dot{\mathbf{Q}}_{23} + \dot{\mathbf{Q}}_{33})$. Then we perform a substitution to convert Problem (8) into an equivalent problem that depends on the trigonometric functions: (*i*) $\tilde{c}^2 = \frac{1}{1-\tilde{t}^2}$; (*ii*) $\tilde{s}^2 = \frac{\tilde{t}^2}{1-\tilde{t}^2}$; (*iii*) $\tilde{t} = \frac{\tilde{s}}{\tilde{c}}$. The following lemma provides a characterization of the global optimal solution for Problem (8).

**Lemma 2.3.** *(Proof in Section C.2) We let $\breve{F}(\tilde{c}, \tilde{s}) \triangleq a\tilde{c} + b\tilde{s} + c\tilde{c}^2 + d\tilde{c}\tilde{s} + e\tilde{s}^2$. The optimal solution $\bar{\mu}$ to Problem (8) can be computed as: $[\cosh(\tilde{\mu}), \sinh(\tilde{\mu})] \in \arg\min_{[c,s]} \breve{F}(c, s)$, s.t. $[c, s] \in \{[\frac{1}{\sqrt{1-(\bar{t}_+)^2}}, \frac{\bar{t}_+}{\sqrt{1-(\bar{t}_+)^2}}], [\frac{-1}{\sqrt{1-(\bar{t}_-)^2}}, \frac{-\bar{t}_-}{\sqrt{1-(\bar{t}_-)^2}}]\}$, where $\bar{t}_+ \in \arg\min_t p(t) \triangleq \frac{a+bt}{\sqrt{1-t^2}} + \frac{w+dt}{1-t^2}$; $\bar{t}_- \in \arg\min_t \tilde{p}(t) \triangleq \frac{-a-bt}{\sqrt{1-t^2}} + \frac{w+dt}{1-t^2}$. Here $w = c + e$.*

We now describe how to find the optimal solution $\bar{t}_+$, where $\bar{t}_+ \in \arg\min_t p(t) \triangleq \frac{a+bt}{\sqrt{1-t^2}} + \frac{w+dt}{1-t^2}$; this strategy can naturally be extended to find $\bar{t}_-$. Initially, we have the following first-order optimality conditions for the problem: $0 = \nabla p(t) = [b(1-t^2) + (a+bt)t]\sqrt{1-t^2} + [d(1-t^2) + (w+dt)(2t)] \Leftrightarrow dt^2 + 2wt + d = -[b+at]\sqrt{1-t^2}$. Squaring both sides yields the following quartic equation: $c_4 t^4 + c_3 t^3 + c_2 t^2 + c_1 t + c_0 = 0$, where $c_4 = d^2 + a^2$, $c_3 = 4wd + 2ab$, $c_2 = 4w^2 + 2d^2 - a^2 + b^2$, $c_1 = 4wd - 2ab$, $c_0 = d^2 - b^2$. This equation can be solved analytically by Lodovico Ferrari's method [45], resulting in all its real roots $\{\bar{t}_1, \bar{t}_2, \ldots, \bar{t}_j\}$ with $1 \le j \le 4$.

For the second and third cases, Problem (6) essentially boils down to optimization under orthogonality constraints. The work of [52] derives a breakpoint search method for finding the optimal solution for Problem (6) with $\mathbf{J_{BB}} \in \{(\begin{smallmatrix} 1 & 0 \\ 0 & 1 \end{smallmatrix}), (\begin{smallmatrix} -1 & 0 \\ 0 & -1 \end{smallmatrix})\}$ using the Givens rotation and Jacobi reflection matrices.

## 2.2 Variance-Reduced Jacobi Block Coordinate Descent Algorithm

This subsection proposes the **VR-J-JOBCD** algorithm, a randomized block coordinate descent method derived from **GS-JOBCD**. Importantly, by leveraging the parallel framework of a Jacobi strategy [17; 9], **VR-J-JOBCD** integrates variance reduction techniques [38; 30; 18] to decrease oracle complexity in the minimization of finite-sum functions. This makes the algorithm effective for minimizing large-scale problems under J-orthogonality constraints.

**Notations**. We assume $n$ is an even number in this paper. We create $(n/2)$ pairs by non-overlapping grouping of the numbers in any arbitrary combination, with each pair containing two distinct numbers from the set $[n]$. It is not hard to verify that such grouping yields $\mathrm{C}_J = (n!)/(2^{n/2}\frac{n}{2}!)$ possible combinations. The set of these combinations is denoted as $\Upsilon \triangleq \{\tilde{\mathcal{B}}_i\}_{i=1}^{\mathrm{C}_J} \triangleq \{\tilde{\mathcal{B}}_1, \tilde{\mathcal{B}}_2, \ldots, \tilde{\mathcal{B}}_{\mathrm{C}_J}\}$ [1].

▶ **Variance Reduction Strategy**. We incorporate state-of-the-art variance reduction strategies from the literature [30; 5] into our algorithm to solve Problem (1). These methods iteratively generate a stochastic gradient estimator as follows:

$$\tilde{\mathbf{G}}^t = \begin{cases} \frac{1}{b}\sum_{i\in\mathbb{S}_+^t} \nabla f_i(\mathbf{X}^t), & \text{with probability } p; \\ \tilde{\mathbf{G}}^{t-1} + \frac{1}{b'}\sum_{i\in\mathbb{S}_*^t}(\nabla f_i(\mathbf{X}^t) - \nabla f_i(\mathbf{X}^{t-1})), & \text{with probability } 1-p. \end{cases} \tag{9}$$

Here, $\{\mathbb{S}_+^t, \mathbb{S}_*^t\}$ are uniform random minibatch samples with $|\mathbb{S}_+^t| = b$, $|\mathbb{S}_*^t| = b'$, and $\tilde{\mathbf{G}}^0 = \frac{1}{b}\sum_{i\in\mathbb{S}_+^0} \nabla f_i(\mathbf{X}^0)$. We drop the superscript $t$ for $\{\mathbb{S}_+^t, \mathbb{S}_*^t\}$ as $t$ can be inferred from context. We only focus on the default setting that [30; 5]: $b = N$, $b' = \sqrt{b}$ and $p = \frac{b'}{b+b'}$.

▶ **Jacobi Block Coordinate Descent Method**. The proposed algorithm is built upon the parallel framework of a Jacobi strategy. In each iteration $t$, we randomly and uniformly (with replacement) select a coordinate set $\mathtt{B}^t \triangleq \{\mathtt{B}_{(1)}^t, \mathtt{B}_{(2)}^t, \cdots, \mathtt{B}_{(\frac{n}{2})}^t\}$ from the set $\Upsilon$ with $\mathtt{B}^t \in \mathbb{N}^{\frac{n}{2}\times 2}$ and $\mathtt{B}_{(i)}^t \in \mathbb{N}^2$. For all $t$, we have: $\mathtt{B}_{(i)}^t \cap \mathtt{B}_{(j)}^t = \emptyset$ and $\cup_{i=1}^{n/2}(\mathtt{B}_{(i)}^t) = [n]$. We drop the superscript $t$ if $t$ can be inferred from context.

The following lemma shows how to choose a suitable matrix $\mathbf{Q}$ so that the Jacobi strategy can be applied.

---

[1]Taking $n = 4$ for example, we have: $\Upsilon = \{\{(1, 2), (3, 4)\}, \{(1, 3), (2, 4)\}, \{(1, 4), (2, 3)\}\}$.

**Lemma 2.4.** *(Proof in Section C.3) We let $\mathtt{B}^t \triangleq \{\mathtt{B}^t_{(1)}, \mathtt{B}^t_{(2)}, \cdots, \mathtt{B}^t_{(\frac{n}{2})}\} \in \Upsilon$ for all $t$. We let $\mathbf{Q} = \varsigma \mathbf{I}_4$, where $\varsigma$ is some suitable constant with $\varsigma \leq L_f$. For any $\mathtt{B}^t_{(i)}$ and $\mathtt{B}^t_{(j)}$ with $i \neq j$, their corresponding objective functions as in Equation (3) are independent.*

We consider the following block coordinate update rule in **VR-J-JOBCD**: $\mathbf{X}^{t+1} \Leftarrow \tilde{\mathcal{X}}^t_\mathtt{B}(\mathbf{V}_:) \triangleq \mathbf{X}^t + [\sum_{i=1}^{n/2} \mathbf{U}_{\mathtt{B}_{(i)}}(\mathbf{V}_i - \mathbf{I}_2)\mathbf{U}^\top_{\mathtt{B}_{(i)}}]\mathbf{X}^t$. The following lemma provides properties of this rule.

**Lemma 2.5.** *(Proof in Section C.4) We let $\mathtt{B} \in \Upsilon$, $\mathbf{V}_i \in \mathcal{J}_{\mathtt{B}_{(i)}}$, $\mathbf{X} \in \mathcal{J}$, and $i \in [\frac{n}{2}]$. We define $\mathbf{X}^+ \triangleq \tilde{\mathcal{X}}_\mathtt{B}(\mathbf{V}_:) \triangleq \mathbf{X} + [\sum_{i=1}^{n/2} \mathbf{U}_{\mathtt{B}_{(i)}}(\mathbf{V}_i - \mathbf{I}_2)\mathbf{U}^\top_{\mathtt{B}_{(i)}}]\mathbf{X}$. We have: (a) $\sum_{i=1}^{\frac{n}{2}} \|\mathbf{U}_{\mathtt{B}_{(i)}}(\mathbf{V}_i - \mathbf{I}_2)\mathbf{U}^\top_{\mathtt{B}_{(i)}}\mathbf{X}\|^2_\mathsf{F} = \|\sum_{i=1}^{\frac{n}{2}} \mathbf{U}_{\mathtt{B}_{(i)}}(\mathbf{V}_i - \mathbf{I}_2)\mathbf{U}^\top_{\mathtt{B}_{(i)}}\mathbf{X}\|^2_\mathsf{F}$. (b) $\|\mathbf{X}^+ - \mathbf{X}\|^2_\mathsf{F} \leq \|\mathbf{X}\|^2_\mathsf{F} \cdot \sum_{i=1}^{n/2} \|\mathbf{V}_i - \mathbf{I}_2\|^2_\mathsf{F}$. (c) $\|\mathbf{X}^+ - \mathbf{X}\|^2_\mathbf{H} \leq \sum_{i=1}^{n/2} \|\mathbf{V}_i - \mathbf{I}_2\|^2_\mathbf{Q}$ with $\mathbf{Q} = \varsigma \mathbf{I}_4$. (d) For all $\tilde{\mathbf{G}} \in \mathbb{R}^{n \times n}$, it follows that: $2\sum_{i=1}^{n/2}\langle \mathbf{V}_i - \mathbf{I}_2, [(\nabla f(\mathbf{X}) - \tilde{\mathbf{G}})\mathbf{X}^\top]_{\mathtt{B}_{(i)}\mathtt{B}_{(i)}}\rangle \leq \|\mathbf{X}\|^2_\mathsf{F}\sum_{i=1}^{n/2}\|\mathbf{V}_i - \mathbf{I}_2\|^2_\mathsf{F} + \|[\nabla f(\mathbf{X}) - \tilde{\mathbf{G}}]\|^2_\mathsf{F}$.*

▶ **The Main Algorithm**. Using the update rule above, we consider the following iterative procedure: $\mathbf{X}^{t+1} \Leftarrow \tilde{\mathcal{X}}^t_\mathtt{B}(\mathbf{V}_:)$, where $\bar{\mathbf{V}}^t_: \in \arg\min_{\mathbf{V}_:} f(\tilde{\mathcal{X}}^t_\mathtt{B}(\mathbf{V}_:))$. We establish the majorization function for $f(\tilde{\mathcal{X}}^t_\mathtt{B}(\mathbf{V}_:))$, as follows:

$$f(\tilde{\mathcal{X}}^t_\mathtt{B}(\mathbf{V}_:)) \overset{①}{\leq} f(\mathbf{X}^t) + \langle \tilde{\mathcal{X}}^t_\mathtt{B}(\mathbf{V}_:) - \mathbf{X}^t, \nabla f(\mathbf{X}^t)\rangle + \frac{1}{2}\|\tilde{\mathcal{X}}^t_\mathtt{B}(\mathbf{V}_:) - \mathbf{X}^t\|^2_\mathbf{H}$$

$$\overset{②}{\leq} f(\mathbf{X}^t) + \sum_{i=1}^{n/2}\{\langle \mathbf{V}_i - \mathbf{I}_2, [\nabla f(\mathbf{X})(\mathbf{X})^\top]_{\mathtt{B}_{(i)}\mathtt{B}_{(i)}}\rangle + \frac{1}{2}\|\mathbf{V}_i - \mathbf{I}_2\|^2_{(\theta+\varsigma)\mathbf{I}}\} \quad (10)$$

where step ① uses the results of telescoping Inequality (2) over $i$ from 1 to $N$; step ② uses $\mathbf{X}^{t+1} - \mathbf{X}^t = [\sum_{i=1}^{n/2} \mathbf{U}_{\mathtt{B}_{(i)}}(\mathbf{V}_i - \mathbf{I}_2)\mathbf{U}^\top_{\mathtt{B}_{(i)}}]\mathbf{X}^t$, Claim (c) of Lemma 2.5, $\theta \geq 0$, and $\mathbf{Q} = \varsigma \mathbf{I}$.

Instead of computing the exact Euclidean gradient $\nabla f(\mathbf{X}^t)$ as **GS-JOBCD**, **VR-J-JOBCD** maintains and updates a recursive gradient estimator $\tilde{\mathbf{G}}^t$ using a variance-reduced strategy as in Formula (9). We consider minimizing the following function instead of the one on the right-hand side of Inequality (10):

$$\mathcal{T}(\mathbf{V}_:; \mathbf{X}^t, \mathtt{B}^t) \triangleq f(\mathbf{X}^t) + \sum_{i=1}^{n/2}\langle \mathbf{V}_i - \mathbf{I}_2, [\tilde{\mathbf{G}}^t(\mathbf{X}^t)^\top]_{\mathtt{B}_{(i)}\mathtt{B}_{(i)}}\rangle + \frac{1}{2}\|\mathbf{V}_i - \mathbf{I}_2\|^2_{\check{\mathbf{Q}}}. \quad (11)$$

Here, $\mathcal{T}(\mathbf{V}_:; \mathbf{X}^t, \mathtt{B}^t)$ can be termed as a stochastic majorization function of $f(\tilde{\mathcal{X}}^t_\mathtt{B}(\mathbf{V}_:))$ at the current solution $\mathbf{X}^t$. Therefore, we can consider the following optimization problem to find $\{\mathbf{V}_:\}$ using: $\bar{\mathbf{V}}^t_: \in \arg\min_{\mathbf{V}_:} \mathcal{T}(\mathbf{V}_:; \mathbf{X}^t, \mathtt{B}^t)$, which can be decomposed into $(n/2)$ independent subproblems and solved in parallel. It is important to note that each $\mathbf{V}_i$ in Problem (12) is identical to Problem (6), which can be efficiently solved in $\mathcal{O}(1)$ using the breakpoint search method, as in **GS-JOBCD**.

We summarize the proposed **VR-J-JOBCD** in Algorithm 2. Notably, when $N = 1$, **VR-J-JOBCD** simplifies to a direct Jacobi strategy for solving Problem (1), which we refer to as **J-JOBCD**.

## 3 Optimality Analysis

This section provides an optimality analysis for the proposed algorithms.

Initially, we define the first-order optimality condition for Problem (1). Since the matrix $\mathbf{X}^\top \mathbf{J} \mathbf{X}$ is symmetric, the Lagrangian multiplier $\Lambda$ corresponding to the constraints $\mathbf{X}^\top \mathbf{J} \mathbf{X} = \mathbf{J}$ is also a symmetric matrix. The Lagrangian function of problem (1) is $\mathcal{L}(\mathbf{X}, \Lambda) = f(\mathbf{X}) - \frac{1}{2}\langle \Lambda, \mathbf{X}^\top \mathbf{J} \mathbf{X} - \mathbf{J}\rangle$.

We obtain the following lemma for the first-order optimality condition for Problem (1).

**Lemma 3.1.** *(Proof in Section D.1, First-Order Optimality Condition) We let $\mathcal{J} \triangleq \{\mathbf{X} \,|\, \mathbf{X}^\top \mathbf{J} \mathbf{X} = \mathbf{J}\}$. We have (a) A solution $\check{\mathbf{X}} \in \mathcal{J}$ is a critical point of problem (1) if and only if: $\mathbf{0} = \nabla_\mathcal{J} f(\check{\mathbf{X}}) \triangleq \nabla f(\check{\mathbf{X}}) - \mathbf{J}\check{\mathbf{X}}[\nabla f(\check{\mathbf{X}})]^\top\check{\mathbf{X}}\mathbf{J}$. The associated Lagrangian multiplier can be computed as $\Lambda = \mathbf{J}\check{\mathbf{X}}^\top\nabla f(\check{\mathbf{X}})$. (b) The critical point condition is equivalent to the requirement that the matrix $\mathbf{X}\nabla f(\check{\mathbf{X}})^\top\mathbf{J}$ is symmetric, which is expressed as $\mathbf{X}\mathbf{G}^\top\mathbf{J} = [\mathbf{X}\mathbf{G}^\top\mathbf{J}]^\top$.*

---

**Algorithm 2: VR-J-JOBCD**: Block Coordinate Descent Methods using a variance-reduced and Jacobi strategy for Solving Problem 1

---

**Init.:** Set $\mathbf{X}^0$ to satisfy J-orthogonality constraints (e.g., via Hyperbolic CS Decomposition), $\theta$ in Inequality (3) (e.g., 1e-6) and $\xi$ satisfy Inequality (4).

**for** $t$ from 0 to $T$ **do**

   (S1) Choose a coordinate $\mathtt{B}^t$ from the set $\Upsilon$ randomly and uniformly (with replacement) for the $t$-th iteration. Denote $\mathtt{B} = \mathtt{B}^t$. In our implementation, we simply randomly permute the set $\{1, 2, ..., n\}$ and then output the grouping $\{[1, 2], [3, 4], [5, 6], \cdots, , [n - 1, n]\}$.

   (S2) Use a variance-reduced strategy (9) to obtain $\tilde{\mathbf{G}}^t$.

   (S3) Solve small-sized subproblems in parallel with $\mathbf{Q} = \varsigma \mathbf{I} \in \mathbb{R}^{4 \times 4}$.

   **for** $i = 1$ **to** $n/2$ **in parallel do**

$$
\begin{aligned}
\overline{\mathrm{V}}_i^t \in \; & \arg\min_{\mathbf{V}_i \in \mathcal{J}_{\mathtt{B}_{(i)}}} \quad \tfrac{1}{2}\|\mathbf{V}_i - \mathbf{I}\|_{\ddot{\mathbf{Q}}}^2 + \langle \mathbf{V}_i - \mathbf{I}, [\nabla f(\mathbf{X}^t)(\mathbf{X}^t)^{\mathsf{T}}]_{\mathtt{B}_{(i)}\mathtt{B}_{(i)}} \rangle + f(\mathbf{X}^t) \\
= \; & \arg\min_{\mathbf{V}_i \in \mathcal{J}_{\mathtt{B}_{(i)}}} \quad \tfrac{1}{2}\|\mathbf{V}_i\|_{\ddot{\mathbf{Q}}}^2 + \langle \mathbf{V}_i, \mathbf{P}_i \rangle
\end{aligned} \tag{12}
$$

   where $\mathbf{P}_i \triangleq [\nabla f(\mathbf{X}^t)(\mathbf{X}^t)^{\mathsf{T}}]_{\mathtt{B}_{(i)}\mathtt{B}_{(i)}} - \mathrm{mat}(\ddot{\mathbf{Q}} \, \mathrm{vec}(\mathbf{I}_2)) - \theta \mathbf{I}_2$, $\ddot{\mathbf{Q}} = (\zeta + \theta)\mathbf{I}$.

   (S4) Update the solution $\mathbf{X}^{t+1}$ in parallel as follows:

   **for** $i = 1$ **to** $n/2$ **in parallel do**

     $\mathbf{X}^{t+1}(\mathtt{B}_{(i)}, :) = \overline{\mathbf{V}}_i^t \mathbf{X}^t(\mathtt{B}_{(i)}, :)$

**end**

---

**Remarks**. While our results in Lemma 3.1 show similarities to existing works focusing on problems under orthogonality constraints [44], this study marks the first investigation into the first-order optimality condition for optimization problems under J-orthogonality constraints.

The following definition is useful in our subsequent analysis of the proposed algorithms.

**Definition 3.2.** (Block Stationary Point, abbreviated as BS-point) Let $\theta > 0$. A solution $\ddot{\mathbf{X}} \in \mathcal{J}$ is termed as a block stationary point if, for all $\mathtt{B} \in \Omega \triangleq \{\mathcal{B}_1, \mathcal{B}_2, \ldots, \mathcal{B}_{\mathrm{C}_n^2}\}$, the following condition is satisfied: $\mathbf{I}_2 \in \arg\min_{\mathbf{V} \in \mathcal{J}_{\mathtt{B}}} \mathcal{G}(\mathbf{V}; \ddot{\mathbf{X}}, \mathtt{B})$.

The following theorem shows the relation between critical points and BS-points.

**Theorem 3.3.** *(Proof in Section D.2) Any* BS-point is a critical point, while the reverse is not necessarily true.

## 4 Convergence Analysis

This section provides a convergence analysis for **GS-JOBCD** and **VR-J-JOBCD**.

For **GS-JOBCD**, the randomness of output $(\overline{\mathrm{V}}^t, \mathbf{X}^{t+1})$ for all $t$ are influenced by the random variable $\xi^t \triangleq (\mathtt{B}^1; \mathtt{B}^2; \cdots; \mathtt{B}^t)$. For **VR-J-JOBCD**, the randomness of output $(\bar{\mathbf{V}}_:^t, \mathbf{X}^{t+1})$ are influenced by the random variables $\iota^t \triangleq (\mathtt{B}^1, \mathtt{S}_+^1, \mathtt{S}_*^1; \mathtt{B}^2, \mathtt{S}_+^2, \mathtt{S}_*^2; \cdots; \mathtt{B}^t, \mathtt{S}_+^t, \mathtt{S}_*^t)$.

We denote $\bar{\mathbf{X}}$ as the global optimal solution of Problem (1). To simplify notations, we define: $u^t = \|\tilde{\mathbf{G}}^t - \nabla f(\mathbf{X}^t)\|_{\mathsf{F}}^2$, and $\Delta_i = f(\mathbf{X}^i) - f(\bar{\mathbf{X}})$.

We impose the following additional assumptions on the proposed algorithms.

**Assumption 4.1.** There exists constants $\{\overline{\mathrm{X}}, \overline{\mathrm{V}}\}$ that: $\|\mathbf{X}^t\|_{\mathsf{F}} \leq \overline{\mathrm{X}}$, $\|\mathbf{V}^t\|_{\mathsf{F}} \leq \overline{\mathrm{V}}$ for all $t$.

**Assumption 4.2.** There exists a constant $\overline{\mathrm{G}}$ that: $\|\nabla f(\mathbf{X}^t)\|_{\mathsf{F}} \leq \overline{\mathrm{G}}$, $\|\tilde{\mathbf{G}}^t\|_{\mathsf{F}} \leq \overline{\mathrm{G}}$ for all $t$.

**Assumption 4.3.** For any $\mathbf{X} \in \mathbb{R}^{n \times n}$, $\mathbb{E}_i[\|\nabla f_i(\mathbf{X}^t) - \nabla f(\mathbf{X}^t)\|_{\mathsf{F}}^2] \leq \sigma^2$, where $i$ is drawn uniformly at random from $[N]$.

**Remarks**. (***i***) Assumption 4.1 is satisfied as the function $f_i(\mathbf{X})$ is coercive for all $i$. (***ii***) Assumption 4.2 imposes a bound on the (stochastic) gradient, a fairly moderate condition

frequently employed in nonconvex optimization [26]. (***iii***) Assumption 4.3 ensures that the variance of the stochastic gradient is bounded, which is a common requirement in stochastic optimization [30; 5].

### 4.1 GLOBAL CONVERGENCE

We define the $\epsilon$-BS-point as follows.

**Definition 4.4.** ($\epsilon$-BS-point) Given any constant $\epsilon > 0$, a point $\ddot{\mathbf{X}}$ is called an $\epsilon$-BS-point if: $\mathcal{E}(\ddot{\mathbf{X}}) \leq \epsilon$. Here, $\mathcal{E}(\mathbf{X})$ is defined as $\mathcal{E}(\mathbf{X}) \triangleq \frac{1}{C_n^2} \sum_{i=1}^{C_n^2} \text{dist}(\mathbf{I}_2, \arg\min_{\mathbf{V}} \mathcal{G}(\mathbf{V}; \mathbf{X}, \mathcal{B}_i))^2$ for For **GS-JOBCD**, while it is defined as $\mathcal{E}(\mathbf{X}) \triangleq \frac{1}{C_J} \sum_{i=1}^{C_J} \mathbb{E}_{\iota^t}[\text{dist}(\mathbf{I}_2, \arg\min_{\mathbf{V}_:} \mathcal{T}(\mathbf{V}_:; \mathbf{X}, \tilde{\mathcal{B}}_i))^2]$ for **VR-J-JOBCD**, where the expectation is with respect to the randomness inherent in the algorithm [30].

We have the following useful lemma for **VR-J-JOBCD**.

**Lemma 4.5.** *(Proof in Section E.1) Suppose Assumption 4.3 holds, then the variance $\mathbb{E}_{\iota^t}[u_k]$ of the gradient estimators $\{\tilde{\mathbf{G}}^t\}$ of Algorithm 2 is bounded by:* $\mathbb{E}_{\iota^t}[u^t] \leq \frac{p(N-b)}{b(N-1)}\sigma^2 + (1-p)\mathbb{E}_{\iota^{t-1}}[u^{t-1}] + \frac{L_f^2 \overline{X}^2(1-p)}{b'}\mathbb{E}_{\iota^{t-1}}[\sum_{i=1}^{n/2} \|\mathbf{V}_i^{t-1} - \mathbf{I}_2\|_{\mathsf{F}}^2]$

The following two theorems establish the iteration complexity (or oracle complexity) for **GS-JOBCD** and **VR-J-JOBCD**.

**Theorem 4.6.** *(Proof in Section E.2)* **GS-JOBCD** *finds an $\epsilon$-BS-point of Problem (1) within $\mathcal{O}(\frac{\Delta_0 N}{\epsilon})$ arithmetic operations.*

**Theorem 4.7.** *(Proof in Section E.3) Let $b = N$, $b' = \sqrt{N}$, and $p = \frac{b'}{b+b'}$.* **VR-J-JOBCD** *finds an $\epsilon$-BS-point of Problem (1) within $\mathcal{O}(nN + \frac{\Delta_0 \sqrt{N}}{\epsilon})$ arithmetic operations.*

**Remark.** Theorems 4.6 and 4.7 demonstrate that the arithmetic operation complexity of **GS-JOBCD** is linearly dependent on $N$, while **VR-J-JOBCD** is linearly dependent on $\sqrt{N}$. Therefore, **VR-J-JOBCD** reduces the iteration complexity significantly.

### 4.2 STRONG CONVERGENCE UNDER KL ASSUMPTION

We prove algorithms achieve strong convergence based on a non-convex analysis tool called Kurdyka-ojasiewicz inequality[2]. We impose the following assumption on Problem (1).

**Assumption 4.8.** (Kurdyka-ojasiewicz Property). Assume that $f^\circ(\mathbf{X}) = f(\mathbf{X}) + \mathcal{I}_{\mathcal{J}}(\mathbf{X})$ is a KL function. For all $\mathbf{X} \in \text{dom } f^\circ$, there exists $\sigma \in [0,1), \eta \in (0,+\infty]$ a neighborhood $\Upsilon$ of $\mathbf{X}$ and a concave and continuous function $\varphi(t) = ct^{1-\sigma}, c > 0, t \in [0,\eta)$ such that for all $\mathbf{X}' \in \Upsilon$ and satisfies $f^\circ(\mathbf{X}') \in (f^\circ(\mathbf{X}), f^\circ(\mathbf{X}) + \eta)$, the following holds: $\text{dist}(\mathbf{0}, \nabla f^\circ(\mathbf{X}'))\varphi'(f^\circ(\mathbf{X}') - f^\circ(\mathbf{X})) \geq 1$.

We establish strong limit-point convergence for **VR-J-JOBCD** and **GS-JOBCD**.

**Theorem 4.9.** *(Proof in Section E.5, a Finite Length Property). The sequence $\{\mathbf{X}^t\}_{t=0}^\infty$ of* **GS-JOBCD** *has finite length property that: $\forall t, \sum_{i=1}^t \mathbb{E}_{\xi^t}[\|\mathbf{X}^{t+1} - \mathbf{X}^t\|_{\mathsf{F}}] \leq \mathcal{O}(\varphi(\Delta_1)) < +\infty$, where $\varphi(\cdot)$ is the desingularization function defined in Proposition 4.8.*

**Theorem 4.10.** *(Proof in Section E.4, a Finite Length Property). Choosing $b = N$, $b' = \sqrt{N}$ and $p = \frac{b'}{b+b'}$, then the sequence $\{\mathbf{X}^t\}_{t=0}^\infty$ of* **VR-J-JOBCD** *has finite length property that: $\forall t, \sum_{i=1}^t \mathbb{E}_{\iota^t}[\|\mathbf{X}^{t+1} - \mathbf{X}^t\|_{\mathsf{F}}] \leq \mathcal{O}(\frac{\varphi(\Delta_1)}{N^{1/4}}) < +\infty$, where $\varphi(\cdot)$ is the desingularization function defined in Assumption 4.8.*

## 5 DISCUSSION

▶ **Differences with [53].** Since both our paper and [53] adopt the block coordinate descent method as the framework, we compare and analyze this paper with [53] in Table 1. Moreover, this paper is the first to propose the first-order optimality condition, the tangent space of the optimization manifold, the optimality condition, and the convergence properties for J orthogonality constraint problem for the first time.

Table 1: Comparisons with JOBCD and [53]

| | ours | [53] |
|---|---|---|
| Problem nature | unbounded | compact |
| parallelizability | ✓ | ✗ |
| Include Stochastic strategy | ✓ | ✗ |
| Convergence analysis under parallelization/Variance-Reduction strategy | ✓ | ✗ |
| New applications | ✓ | ✗ |

▶ **Comparisons with GS-JOBCD and J-JOBCD.** GS-JOBCD and **J-JOBCD** are complementary and mutually confirmatory. (**i**) The **Q** in **GS-JOBCD** (s2) has two selection methods as described in (3). However, in the **(VR)-J-JOBCD** (s1), only $\mathbf{Q} = \varsigma\mathbf{I} \in \mathbb{R}^{4\times 4}$ is applicable due to the requirement for independent updates of each block in the parallelization strategy. (**ii**) In small-scale problems, the benefits of parallel strategies are limited. Therefore, the sequential GS-JOBCD generally performs better.

## 6 Applications and Numerical Experiments

This section demonstrates the effectiveness and efficiency of **JOBCD** on three optimization tasks: (**i**) the hyperbolic eigenvalue problem, (**ii**) structural probe problem, and (**iii**) Ultra-hyperbolic Knowledge Graph Embedding problem. We provide experiments for the last problem in Section F.2.

▶ **Application to the Hyperbolic Eigenvalue Problem (HEVP)**. The hyperbolic eigenvalue problem refers to the generalized eigenvalue problem in hyperbolic spaces [39]. This problem is a fundamental component in machine learning models, such as Hyperbolic PCA [42; 6]. Given a data matrix $\mathbf{D} \in \mathbb{R}^{m\times n}$ and a signature matrix $\mathbf{J}$ with signature $(p, n-p)$, HEVP can be formulated as the following optimization problem: $\min_{\mathbf{X}} -\mathbf{tr}(\mathbf{X}^\top\mathbf{D}^\top\mathbf{D}\mathbf{X})$, s.t. $\mathbf{X}^\top\mathbf{J}\mathbf{X} = \mathbf{J}$.

▶ **Application to the Hyperbolic Structural Probe Problem (HSPP)**. The Structure Probe (SP) is a metric learning model aimed at understanding the intrinsic semantic information of large language models [20] [7]. Given a data matrix $\mathbf{D} \in \mathbb{R}^{m\times n}$ and its associated Euclidean distance metric matrix $\mathbf{T} \in \mathbb{R}^{m\times m}$, HSPP employs a smooth homeomorphic mapping function $\varphi(\cdot)$ to project the data $\mathbf{D}$ into ultra-hyperbolic space. Subsequently, it seeks an appropriate linear transformation $\mathbf{X} \in \mathbb{R}^{n\times n}$ constrained within a specific structure $\mathbf{X} \in \mathcal{J}$, such that the resulting transformed data $\mathbf{Q} \triangleq \varphi(\mathbf{D})\mathbf{X} \in \mathbb{R}^{m\times n}$ exhibits similarity to the original distance metric matrix $\mathbf{T}$ under the ultra-hyperbolic geodesic distance $d_\alpha(\mathbf{Q}_{i:}, \mathbf{Q}_{j:})$, expressed as $\mathbf{T}_{i,j} \approx d_\alpha(\mathbf{Q}_{i:}, \mathbf{Q}_{j:})$ for all $i, j \in [m]$, where $\mathbf{Q}_{i:}$ is $i$-th row of the matrix $\mathbf{Q} \in \mathbb{R}^{m\times n}$. This can be formulated as the following optimization problem: $\min_{\mathbf{X}} \frac{1}{m^2}\sum_{i,j\in m}(\mathbf{T}_{i,j} - d_\alpha(\mathbf{Q}_{i:}, \mathbf{Q}_{j:}))^2$, s.t. $\mathbf{Q} \triangleq \varphi(\mathbf{D})\mathbf{X}, \mathbf{X} \in \mathcal{J}$. For more details on the functions $\varphi(\cdot)$ and $d_\alpha(\cdot, \cdot)$, please refer to Appendix Section F.1.

▶ **Datasets**. To generate the matrix $\mathbf{D} \in \mathbb{R}^{m\times n}$, we use 8 real-world or synthetic data sets for both HEVP and HSPP tasks: 'Cifar', 'CnnCaltech', 'Gisette', 'Mnist', 'randn', 'Sector', 'TDT2', 'w1a'. We randomly extract a subset from the original data sets for the experiments.

▶ **Compared Methods**. We compare **GS-JOBCD** and **VR-J-JOBCD** with 3 state-of-the-art optimization algorithms under J-orthogonality constraints. (**i**) The CS Decomposition Method (**CSDM**) [49]. (**ii**) Stardard ADMM (**ADMM**) (Appendix Section F.3) [19]. (**iii**)**UMCM**: Unconstrained Multiplier Correction Method (Appendix Section F.4) [47; 48; 13].

▶ **Experiment Settings**. All methods are implemented using Pytorch on an Intel 2.6 GHz processor with an A40 (48GB). For **HSPP**, we fix $\alpha$ to 1. Each method employs the same random J-orthogonal matrix $\mathbf{X}^0$ with $\frac{1}{n^2}\sum_{ij}^n |\mathbf{X^0}^\top\mathbf{J}\mathbf{X^0} - \mathbf{J}|_{ij} \le 1e-9$. For **(VR)-J-JOBCD**, we define $\mathbf{X}$ as a high-dimensional tensor in PyTorch to achieve parallelization. The built-in solver *Adagrad* is used to solve the unconstrained minimization problem in **CSDM** and **UMCM**, the optimal learning rate is selected from the range [5e-4, 5e-3]. For more experimental details for **ADMM** and **UMCM**, please refer to Appendix Section F.5. We provide our code in the supplemental material.

Table 2: Comparisons of the objectives for HEVP across all the compared methods. The time limit is set to 90s. The notation '(+)' indicates that **GS-JOBCD** significantly improves upon the initial solution provided by **CSDM**. The $1^{st}$, $2^{nd}$, and $3^{rd}$ best results are colored with red, green and blue, respectively. The value in (·) stands for $\frac{1}{n^2} \sum_{ij}^{n} |\mathbf{X}^\top \mathbf{J} \mathbf{X} - \mathbf{J}|_{ij}$ and cells with this value greater than 1e-5 are highlighted in gray.

| dataname(m-n-p) | UMCM | ADMM | CSDM | GS-JOBCD | J-JOBCD | CSDM+GS-JOBCD |
|---|---|---|---|---|---|---|
| cifar(1000-100-50) | -1.11e+04(9.9e-10) | -7.32e+03(1.0e-07) | -1.07e+04(1.4e-09) | -1.53e+04(1.3e-08) | -3.43e+04(6.7e-08) | -6.97e+04(1.6e-08)(+) |
| CnnCaltech(2000-1000-500) | -2.62e+02(1.9e-10) | -2.51e+02(6.5e-08) | -2.80e+02(1.4e-10) | -2.51e+02(1.2e-10) | -1.96e+03(1.6e-08) | -2.87e+02(1.8e-10)(+) |
| gisette(3000-1000-500) | -1.02e+08(5.4e-05) | -1.49e+11(2.5e+00) | -1.77e+06(1.3e-10) | -1.41e+06(1.9e-10) | -3.58e+06(6.0e-09) | -1.81e+06(2.4e-10)(+) |
| mnist(1000-780-390) | -2.13e+05(1.4e-09) | -3.32e+06(1.4e-02) | -5.22e+04(1.6e-10) | -4.41e+04(2.9e-10) | -4.63e+05(2.1e-08) | -8.88e+04(5.6e-10)(+) |
| randn10(10-10-5) | -4.23e+01(8.4e-09) | -4.21e+01(1.1e-07) | -2.29e+02(2.7e-01) | -4.23e+01(1.5e-08) | -8.46e+01(7.5e-07) | -3.01e+02(2.7e-01)(+) |
| randn100(100-100-50) | -5.01e+03(1.4e-09) | -4.93e+03(1.1e-07) | -1.34e+04(3.4e-09) | -4.90e+03(2.6e-09) | -1.71e+04(1.2e-07) | -2.13e+04(5.8e-08)(+) |
| randn1000(1000-1000-500) | -5.64e+05(4.9e-07) | -5.63e+05(6.6e-08) | -5.46e+05(1.4e-10) | -5.01e+05(1.1e-10) | -2.84e+06(6.0e-08) | -5.52e+05(2.4e-10)(+) |
| sector(500-1000-500) | -1.42e+03(7.5e-10) | -1.51e+03(5.2e-08) | -1.63e+03(1.1e-10) | -1.54e+03(1.6e-10) | -2.52e+03(3.4e-09) | -1.63e+03(1.4e-10) |
| TDT2(1000-1000-500) | -1.05e+08(2.8e-06) | -7.34e+08(5.8e+00) | -1.93e+06(1.1e-10) | -1.81e+06(1.8e-10) | -3.21e+06(3.6e-09) | -1.93e+06(1.4e-10) |
| w1a(2470-290-145) | -1.46e+04(2.0e-09) | -1.39e+04(7.8e-08) | -1.45e+04(1.1e-05) | -1.38e+04(3.1e-09) | -3.60e+06(5.3e-07) | -1.63e+04(1.1e-05)(+) |

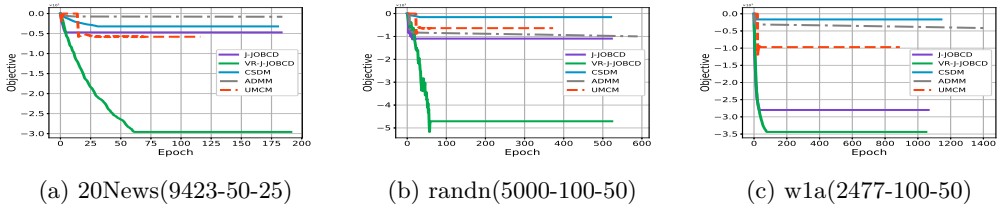

(a) cifar(1000-100-70)          (b) sector(500-1000-700)          (c) TDT2 (1000-1000-700)

Figure 1: The convergence curve for the HEVP across various datasets with different parameters $(m - n - p)$.

(a) 20News(9423-50-25)          (b) randn(5000-100-50)          (c) w1a(2477-100-50)

Figure 2: Comparisons of objective values $(F(\mathbf{X}) - F^0)$ of HSPP for all the compared methods with different parameters $(m - n - p)$.

▶ **Experiment Results.** Table 2 and Figure 1 display the accuracy and computational efficiency for HEVP, while Figure 2 presents the results for HSPP, leading to the following observations: (***i***) **GS-JOBCD** and **JJOBCD** consistently deliver better performance than the other methods. (***ii***) Other methods frequently encounter poor local minima, whereas **GS-JOBCD** effectively escapes these minima and typically achieves lower objective values, aligning with our theory that our methods locate stronger stationary points. (***iii***) **VR-J-JOBCD** outperforms both **J-JOBCD** and **CSDM** when dealing with a large dataset characterized by an finite-sum structure.

## 7 Conclusions

In this paper, we propose a new approach JOBCD, which is based on block coordinate descent, for solving the optimization problem under J-orthogonality constraints. We discuss two specific variants of JOBCD: one based on a Gauss-Seidel strategy (GS-JOBCD), the other on a variance-reduced Jacobi strategy. Both algorithms capitalize on specific structural characteristics of the constraints to converge to more favorable stationary solutions. Notably, **VR-J-JOBCD** incorporates a variance-reduction technique into a parallel framework to reduce oracle complexity in the minimization of finite-sum functions. For both **GS-JOBCD** and **VR-J-JOBCD**, we establish the oracle complexity under mild conditions and strong limit-point convergence results under the Kurdyka-Lojasiewicz inequality. Some experiments on the hyperbolic eigenvalue problem and structural probe problem show the efficiency and efficacy of the proposed methods.

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

# Appendix

The appendix is organized as follows.

Appendix A introduces some notations, technical preliminaries, and relevant lemmas.

Appendix B concludes some additional discussions.

Appendix C presents the proofs for Section 2.

Appendix D offers the proofs for Section 3.

Appendix E contains the proofs for Section 4.

Appendix F contains several extra experiments, extensions and discussions of the proposed methods.

## A  NOTATIONS, TECHNICAL PRELIMINARIES, AND RELEVANT LEMMAS

### A.1  NOTATIONS

In this paper, we denote the Lowercase boldface letters represent vectors, while uppercase letters represent real-valued matrices. We use the Matlab colon notation to denote indices that describe submatrices. The following notations are used throughout this paper.

- $\mathbb{N}$ : Set of natural numbers
- $\mathbb{R}$ : Set of real numbers
- $[n]$: $\{1, 2, ..., n\}$
- $\|\mathbf{x}\|$: Euclidean norm: $\|\mathbf{x}\| = \|\mathbf{x}\|_2 = \sqrt{\langle \mathbf{x}, \mathbf{x} \rangle}$
- $\mathbf{x}_i$: the $i$-th element of vector $\mathbf{x}$
- $\mathbf{X}_{i,j}$ or $\mathbf{X}_{ij}$ : the ($i^{\text{th}}$, $j^{\text{th}}$) element of matrix $\mathbf{X}$
- $\text{vec}(\mathbf{X})$ : $\text{vec}(\mathbf{X}) \in \mathbb{R}^{nn \times 1}$, the vector formed by stacking the column vectors of $\mathbf{X}$
- $\text{mat}(\mathbf{x}) \in \mathbb{R}^{n \times n}$, Convert $\mathbf{x} \in \mathbb{R}^{nn \times 1}$ into a matrix with $\text{mat}(\text{vec}(\mathbf{X})) = \mathbf{X}$
- $\mathbf{X}^{\mathsf{T}}$ : the transpose of the matrix $\mathbf{X}$
- $\text{sign}(t)$ : the signum function, $\text{sign}(t) = 1$ if $t \geq 0$ and $\text{sign}(t) = -1$ otherwise
- $\mathbf{X} \otimes \mathbf{Y}$ : Kronecker product of $\mathbf{X}$ and $\mathbf{Y}$
- $\det(\mathbf{D})$ : Determinant of a square matrix $\mathbf{D} \in \mathbb{R}^{n \times n} \mathbf{D} \in \mathbb{R}^{n \times n}$
- $\mathbf{C}_n^2$ : the number of possible combinations choosing $k$ items from $n$ without repetition.
- $\mathbf{0}_{n,r}$ : A zero matrix of size $n \times r$; the subscript is omitted sometimes
- $\mathbf{I}_r$ : $\mathbf{I}_r \in \mathbb{R}^{r \times r}$, Identity matrix
- $\mathbf{X} \succeq \mathbf{0}$(or $\succ \mathbf{0}$) : the Matrix $\mathbf{X}$ is symmetric positive semidefinite (or definite)
- $\text{Diag}(\mathbf{x})$: Diagonal matrix with $\mathbf{x}$ as the main diagonal entries.
- $\mathbf{tr}(\mathbf{A})$ : Sum of the elements on the main diagonal $\mathbf{A}$: $\mathbf{tr}(\mathbf{A}) = \sum_i \mathbf{A}_{i,i}$
- $\|\mathbf{X}\|_*$ : Nuclear norm: sum of the singular values of matrix $\mathbf{X}$
- $\|\mathbf{X}\|$ : Operator/Spectral norm: the largest singular value of $\mathbf{X}$
- $\|\mathbf{X}\|_{\mathsf{F}}$ : Frobenius norm: $(\sum_{ij} \mathbf{X}_{ij}^2)^{1/2}$
- $\nabla f(\mathbf{X})$ : classical (limiting) Euclidean gradient of $f(\mathbf{X})$ at $\mathbf{X}$
- $\nabla_{\mathcal{J}} f(\mathbf{X})$ : Riemannian gradient of $f(\mathbf{X})$ at $\mathbf{X}$
- $\mathcal{I}_\xi(\mathbf{X})$ : the indicator function of a set $\xi$ with $\mathcal{I}_\xi(\mathbf{X}) = 0$ if $\mathbf{X} \in \xi$ and otherwise $+\infty$
- $\text{dist}(\xi, \xi')$ : the distance between two sets with $\text{dist}(\xi, \xi') \triangleq \inf_{\mathbf{X} \in \xi, \mathbf{X}' \in \xi'} \|\mathbf{X} - \mathbf{X}'\|_{\mathsf{F}}$
- $\mathcal{I}_\xi(\mathbf{x})$ : the indicator function of a set $\xi$ with $\mathcal{I}_\xi(\mathbf{x}) = 0$ if $\mathbf{x} \in \xi$ and otherwise $+\infty$.

## A.2   Relevant Lemmas

**Lemma A.1.** *(Lemma 6.6 of [52]) For any $\mathbf{W} \in \mathbb{R}^{n \times n}$, we have:* $\sum_{i=1}^{C_n^k} \|\mathbf{W}(\mathcal{B}_i, \mathcal{B}_i)\|_{\mathsf{F}}^2 = \frac{k}{n} C_n^k \sum_i \mathbf{W}_{ii}^2 + C_{n-2}^{k-2} \sum_i \sum_{j, j \neq i} \mathbf{W}_{ij}^2$. *Here, the set $\{\mathcal{B}_1, \mathcal{B}_2, \cdots, \mathcal{B}_{C_n^k}\}$ represents all possible combinations of the index vectors choosing $k$ items from $n$ without repetition.*

**Lemma A.2.** *We have $\mathsf{S}_+$ be the set of $|\mathsf{S}_+| = b$ samples from $[N]$, drawn with replacement and uniformly at random. Then, $\forall t, \mathbf{X}^t \in \mathbb{R}^{n \times n}$, we have:*

$$\mathbb{E}_{\iota^t}[\|\tfrac{1}{b} \sum_{i \in \mathsf{S}_+} \nabla f_i(\mathbf{X}^t) - \nabla f(\mathbf{X}^t)\|_{\mathsf{F}}^2] = \tfrac{N-b}{b(N-1)} \mathbb{E}_{\iota^t}[\|\nabla f_i(\mathbf{X}^t) - \nabla f(\mathbf{X}^t)\|_{\mathsf{F}}^2].$$

*Proof.* The proof is exactly the same as in Lemma 2.8 of [5]. □

**Lemma A.3.** *The tangent space $\mathbf{T_X}\mathcal{J}$ of manifold constructed by $\mathbf{X}^\top \mathbf{JX} = \mathbf{J}$, with $\mathbf{X} \in \mathbb{R}^{n \times n}$, is :*

$$\mathbf{T_X}\mathcal{J} \triangleq \{\mathbf{Y} \in \mathbb{R}^{n \times n} \mid \mathbf{X}^\top \mathbf{JY} + \mathbf{Y}^\top \mathbf{JX} = 0\}, \tag{13}$$

*where $\mathbf{Y} = t\tilde{\mathbf{Y}}$ with $t$ being a positive scalar approaching 0.*

*Proof.* Assuming point $\mathbf{X} \in \mathbb{R}^{n \times n}$ lies on manifold $\mathcal{J}$, we have: $h(\mathbf{X}) = \mathbf{X}^\top \mathbf{JX} - \mathbf{J}$. Moving along $\mathbf{Y} \in \mathbb{R}^{n \times n}$ in the tangent space of $\mathbf{X}$, we obtain:

$$\begin{aligned}
h(\mathbf{X} + \mathbf{Y}) &= (\mathbf{X} + \mathbf{Y})^\top \mathbf{J}(\mathbf{X} + \mathbf{Y}) - \mathbf{J} \\
&= \mathbf{X}^\top \mathbf{JX} + \mathbf{X}^\top \mathbf{JY} + \mathbf{Y}^\top \mathbf{JX} + \mathbf{Y}^\top \mathbf{JY} - \mathbf{J} \\
&\overset{\text{①}}{=} \mathbf{X}^\top \mathbf{JY} + \mathbf{Y}^\top \mathbf{JX} + \mathbf{Y}^\top \mathbf{JY} \\
&\overset{\text{②}}{=} t\mathbf{X}^\top \mathbf{J}\tilde{\mathbf{Y}} + t\tilde{\mathbf{Y}}^\top \mathbf{JX} + t^2 \tilde{\mathbf{Y}}^\top \mathbf{J}\tilde{\mathbf{Y}}
\end{aligned}$$

where step ① uses $\mathbf{X}^\top \mathbf{JX} = \mathbf{J}$; step ② uses $\mathbf{Y} = t\tilde{\mathbf{Y}}$.

Since $t$ is a positive scalar approaching 0, we can ignore the higher-order term: $t^2 \tilde{\mathbf{Y}}^\top \mathbf{J}\tilde{\mathbf{Y}}$. According to the properties of the tangent space of any manifold, we have: $h(\mathbf{X} + \mathbf{Y}) = 0$, In other words, $\mathbf{X}^\top \mathbf{JY} + \mathbf{Y}^\top \mathbf{JX} = 0$, i.e. we obtain the defining equation for the tangent space: $\mathbf{T_X}\mathcal{J} \triangleq \{\mathbf{Y} \in \mathbb{R}^{n \times n} \mid \mathbf{X}^\top \mathbf{JY} + \mathbf{Y}^\top \mathbf{JX} = 0\}$. □

# B   Additional Discussions

## B.1   On the Global Optimal Solution for Problem (7)

In Section 2.1, we have demonstrated how to use the breakpoint search method to obtain an optimal solution for the case of $\mathbf{V} = \left(\begin{smallmatrix} \tilde{c} & \tilde{s} \\ \tilde{s} & \tilde{c} \end{smallmatrix}\right)$ of Problem (7). Since the structure of the other three cases $\mathbf{V} \in \{\left(\begin{smallmatrix} \tilde{c} & -\tilde{s} \\ -\tilde{s} & \tilde{c} \end{smallmatrix}\right), \left(\begin{smallmatrix} -\tilde{c} & -\tilde{s} \\ \tilde{s} & \tilde{c} \end{smallmatrix}\right), \left(\begin{smallmatrix} \tilde{c} & -\tilde{s} \\ \tilde{s} & -\tilde{c} \end{smallmatrix}\right)\}$ is exactly the same except for the coefficients of Problem (8), we will provide the corresponding coefficients in Problem (8): $\min_{\tilde{c}, \tilde{s}} a\,\tilde{c} + b\,\tilde{s} + c\,\tilde{c}^2 + d\,\tilde{c}\,\tilde{s} + e\,\tilde{s}^2$, and omit the specific analysis process.

**Case ($a$).** $\mathbf{V} = \left(\begin{smallmatrix} \tilde{c} & -\tilde{s} \\ -\tilde{s} & \tilde{c} \end{smallmatrix}\right)$: $a = \mathbf{P}_{11} + \mathbf{P}_{22}$, $b = -\mathbf{P}_{12} - \mathbf{P}_{21}$, $c = \frac{1}{2}(\dot{\mathbf{Q}}_{11} + \dot{\mathbf{Q}}_{41} + \dot{\mathbf{Q}}_{14} + \dot{\mathbf{Q}}_{44})$, $d = -\frac{1}{2}(\dot{\mathbf{Q}}_{21} + \dot{\mathbf{Q}}_{31} + \dot{\mathbf{Q}}_{12} + \dot{\mathbf{Q}}_{42} + \dot{\mathbf{Q}}_{13} + \dot{\mathbf{Q}}_{43} + \dot{\mathbf{Q}}_{24} + \dot{\mathbf{Q}}_{34})$, and $e = \frac{1}{2}(\dot{\mathbf{Q}}_{22} + \dot{\mathbf{Q}}_{32} + \dot{\mathbf{Q}}_{23} + \dot{\mathbf{Q}}_{33})$.

**Case ($b$).** $\mathbf{V} = \left(\begin{smallmatrix} -\tilde{c} & -\tilde{s} \\ \tilde{s} & \tilde{c} \end{smallmatrix}\right)$: $a = -\mathbf{P}_{11} + \mathbf{P}_{22}$, $b = -\mathbf{P}_{12} + \mathbf{P}_{21}$, $c = \frac{1}{2}(\dot{\mathbf{Q}}_{11} - \dot{\mathbf{Q}}_{41} - \dot{\mathbf{Q}}_{14} + \dot{\mathbf{Q}}_{44})$, $d = \frac{1}{2}(\dot{\mathbf{Q}}_{21} - \dot{\mathbf{Q}}_{31} + \dot{\mathbf{Q}}_{12} - \dot{\mathbf{Q}}_{42} - \dot{\mathbf{Q}}_{13} + \dot{\mathbf{Q}}_{43} - \dot{\mathbf{Q}}_{24} + \dot{\mathbf{Q}}_{34})$, and $e = \frac{1}{2}(\dot{\mathbf{Q}}_{22} - \dot{\mathbf{Q}}_{32} - \dot{\mathbf{Q}}_{23} + \dot{\mathbf{Q}}_{33})$.

**Case ($c$).** $\mathbf{V} = \left(\begin{smallmatrix} \tilde{c} & -\tilde{s} \\ \tilde{s} & -\tilde{c} \end{smallmatrix}\right)$: $a = \mathbf{P}_{11} - \mathbf{P}_{22}$, $b = -\mathbf{P}_{12} + \mathbf{P}_{21}$, $c = \frac{1}{2}(\dot{\mathbf{Q}}_{11} - \dot{\mathbf{Q}}_{41} - \dot{\mathbf{Q}}_{14} + \dot{\mathbf{Q}}_{44})$, $d = \frac{1}{2}(-\dot{\mathbf{Q}}_{21} + \dot{\mathbf{Q}}_{31} - \dot{\mathbf{Q}}_{12} + \dot{\mathbf{Q}}_{42} + \dot{\mathbf{Q}}_{13} - \dot{\mathbf{Q}}_{43} + \dot{\mathbf{Q}}_{24} - \dot{\mathbf{Q}}_{34})$, and $e = \frac{1}{2}(\dot{\mathbf{Q}}_{22} - \dot{\mathbf{Q}}_{32} - \dot{\mathbf{Q}}_{23} + \dot{\mathbf{Q}}_{33})$.

## C Proofs for Section 2

### C.1 Proof of Lemma 2.1

*Proof.* Defining $\mathbf{J}_{\mathtt{BB}} = \mathbf{J}(\mathbf{U}_{\mathtt{B}}, \mathbf{U}_{\mathtt{B}})$ , then we have: $\mathbf{J}\mathbf{U}_{\mathtt{B}} = \mathbf{U}_{\mathtt{B}}\mathbf{J}_{\mathtt{BB}}$, $\mathbf{U}_{\mathtt{B}}^{\top}\mathbf{J} = \mathbf{J}_{\mathtt{BB}}\mathbf{U}_{\mathtt{B}}^{\top}$, and $\mathbf{U}_{\mathtt{B}}^{\top}\mathbf{J}\mathbf{U}_{\mathtt{B}} = \mathbf{J}_{\mathtt{BB}}$.

**Part (a)**. For any $\mathbf{V} \in \mathbb{R}^{2\times2}$ and $\mathtt{B} \in \{\mathcal{B}_i\}_{i=1}^{\mathbf{C}_n^2}$, we have:

$$[\mathbf{X}^{+}]^{\top}\mathbf{J}\mathbf{X}^{+} - \mathbf{X}^{\top}\mathbf{J}\mathbf{X}$$

$$\overset{①}{=} \mathbf{X}^{\top}\mathbf{J}\mathbf{U}_{\mathtt{B}}(\mathbf{V} - \mathbf{I}_2)\mathbf{U}_{\mathtt{B}}^{\top}\mathbf{X} + [\mathbf{U}_{\mathtt{B}}(\mathbf{V} - \mathbf{I}_2)\mathbf{U}_{\mathtt{B}}^{\top}\mathbf{X}]^{\top}\mathbf{J}\mathbf{X}$$

$$+[\mathbf{U}_{\mathtt{B}}(\mathbf{V} - \mathbf{I}_2)\mathbf{U}_{\mathtt{B}}^{\top}\mathbf{X}]^{\top}\mathbf{J}[\mathbf{U}_{\mathtt{B}}(\mathbf{V} - \mathbf{I}_2)\mathbf{U}_{\mathtt{B}}^{\top}\mathbf{X}]$$

$$= \mathbf{X}^{\top}[\mathbf{J}\mathbf{U}_{\mathtt{B}}(\mathbf{V} - \mathbf{I}_2)\mathbf{U}_{\mathtt{B}}^{\top} + \mathbf{U}_{\mathtt{B}}(\mathbf{V} - \mathbf{I}_2)^{\top}\mathbf{U}_{\mathtt{B}}^{\top}\mathbf{J} + \mathbf{U}_{\mathtt{B}}(\mathbf{V} - \mathbf{I}_2)^{\top}\mathbf{U}_{\mathtt{B}}^{\top}\mathbf{J}\mathbf{U}_{\mathtt{B}}(\mathbf{V} - \mathbf{I}_2)\mathbf{U}_{\mathtt{B}}^{\top}]\mathbf{X}$$

$$= \mathbf{X}^{\top}[\mathbf{U}_{\mathtt{B}}\mathbf{J}_{\mathtt{BB}}(\mathbf{V} - \mathbf{I}_2)\mathbf{U}_{\mathtt{B}}^{\top} + \mathbf{U}_{\mathtt{B}}(\mathbf{V} - \mathbf{I}_2)^{\top}\mathbf{J}_{\mathtt{BB}}\mathbf{U}_{\mathtt{B}}^{\top} + \mathbf{U}_{\mathtt{B}}(\mathbf{V} - \mathbf{I}_2)^{\top}\mathbf{J}_{\mathtt{BB}}(\mathbf{V} - \mathbf{I}_2)\mathbf{U}_{\mathtt{B}}^{\top}]\mathbf{X}$$

$$= \mathbf{X}^{\top}\mathbf{U}_{\mathtt{B}}[\mathbf{J}_{\mathtt{BB}}(\mathbf{V} - \mathbf{I}_2) + (\mathbf{V} - \mathbf{I}_2)^{\top}\mathbf{J}_{\mathtt{BB}} + (\mathbf{V} - \mathbf{I}_2)^{\top}\mathbf{J}_{\mathtt{BB}}(\mathbf{V} - \mathbf{I}_2)]\mathbf{U}_{\mathtt{B}}^{\top}\mathbf{X}$$

$$= \mathbf{X}^{\top}\mathbf{U}_{\mathtt{B}}[\mathbf{V}^{\top}\mathbf{J}_{\mathtt{BB}}\mathbf{V} - \mathbf{J}_{\mathtt{BB}}]\mathbf{U}_{\mathtt{B}}^{\top}\mathbf{X}$$

$$\overset{②}{=} \mathbf{0}.$$

**Part (b)**. Using the update rule for $\mathbf{X}^{+} = \mathbf{X} + \mathbf{U}_{\mathtt{B}}(\mathbf{V} - \mathbf{I}_2)\mathbf{U}_{\mathtt{B}}^{\top}\mathbf{X} \in \mathbb{R}^{n\times n}$, we derive:

$$\|\mathbf{X}^{+} - \mathbf{X}\|_{\mathsf{F}} = \|\mathbf{U}_{\mathtt{B}}(\mathbf{V} - \mathbf{I}_2)\mathbf{U}_{\mathtt{B}}^{\top}\mathbf{X}\|_{\mathsf{F}}$$

$$\overset{①}{\leq} \|\mathbf{U}_{\mathtt{B}}\|_{\mathsf{F}} \cdot \|(\mathbf{V} - \mathbf{I}_2)\mathbf{U}_{\mathtt{B}}^{\top}\mathbf{X}\|_{\mathsf{F}},$$

$$\overset{②}{\leq} \|\mathbf{U}_{\mathtt{B}}\|_{\mathsf{F}} \cdot \|(\mathbf{V} - \mathbf{I}_2)\|_{\mathsf{F}} \cdot \|\mathbf{U}_{\mathtt{B}}^{\top}\|_{\mathsf{F}} \cdot \|\mathbf{X}\|_{\mathsf{F}},$$

$$\overset{③}{=} \|\mathbf{V} - \mathbf{I}_2\|_{\mathsf{F}} \cdot \|\mathbf{X}\|_{\mathsf{F}},$$

where step ① and step ② use the norm inequality that $\|\mathbf{A}\mathbf{X}\|_{\mathsf{F}} \leq \|\mathbf{A}\|_{\mathsf{F}} \cdot \|\mathbf{X}\|_{\mathsf{F}}$ for any $\mathbf{A}$ and $\mathbf{X}$; step ③ uses $\|\mathbf{U}_{\mathtt{B}}\| = \|\mathbf{U}_{\mathtt{B}}^{\top}\| = 1$.

**Part (c)**. We define $\mathbf{Z} \triangleq \mathbf{U}_{\mathtt{B}}^{\top}\mathbf{X}$. We derive:

$$\|\mathbf{X}^{+} - \mathbf{X}\|_{\mathbf{H}}^{2} = \|\mathbf{U}_{\mathtt{B}}(\mathbf{V} - \mathbf{I}_2)\mathbf{Z}\|_{\mathbf{H}}^{2}$$

$$\overset{①}{=} \mathrm{vec}(\mathbf{U}_{\mathtt{B}}(\mathbf{V} - \mathbf{I}_2)\mathbf{Z})^{\top}\mathbf{H}\mathrm{vec}(\mathbf{U}_{\mathtt{B}}(\mathbf{V} - \mathbf{I}_2)\mathbf{Z})$$

$$\overset{②}{=} \mathrm{vec}(\mathbf{V} - \mathbf{I}_2)^{\top}(\mathbf{Z}^{\top} \otimes \mathbf{U}_{\mathtt{B}})^{\top}\mathbf{H}(\mathbf{Z}^{\top} \otimes \mathbf{U}_{\mathtt{B}})\mathrm{vec}(\mathbf{V} - \mathbf{I}_2)$$

$$= \|\mathbf{V} - \mathbf{I}_2\|_{(\mathbf{Z}^{\top}\otimes\mathbf{U}_{\mathtt{B}})^{\top}\mathbf{H}(\mathbf{Z}^{\top}\otimes\mathbf{U}_{\mathtt{B}})}^{2}$$

$$\overset{③}{\leq} \|\mathbf{V} - \mathbf{I}_2\|_{\mathbf{Q}}^{2},$$

where step ① uses $\|\mathbf{X}\|_{\mathbf{H}}^{2} = \mathrm{vec}(\mathbf{X})^{\top}\mathbf{H}\mathrm{vec}(\mathbf{X})$; step ② uses $(\mathbf{Z}^{\top} \otimes \mathbf{R})\mathrm{vec}(\mathbf{U}) = \mathrm{vec}(\mathbf{R}\mathbf{U}\mathbf{Z})$ for all $\mathbf{R}$, $\mathbf{Z}$ and $\mathbf{U}$ of suitable dimensions; step ③ uses the choice of $\mathbf{Q} \succcurlyeq \underline{\mathbf{Q}} \triangleq (\mathbf{Z}^{\top} \otimes \mathbf{U}_{\mathtt{B}})^{\top}\mathbf{H}(\mathbf{Z}^{\top} \otimes \mathbf{U}_{\mathtt{B}})$. $\qquad\square$

### C.2 Proof of Lemma 2.3

*Proof.* We denote $w = c + e$. According to the properties of trigonometric functions, we have: ($i$) $\tilde{c}^2 = \frac{1}{1-\tilde{t}^2}$; ($ii$) $\tilde{s}^2 = \frac{\tilde{t}^2}{1-\tilde{t}^2}$; ($iii$)$\tilde{t} = \frac{\tilde{s}}{\tilde{c}}$, leading to: $\tilde{c} = \frac{\pm 1}{\sqrt{1-\tilde{t}^2}}, \tilde{s} = \frac{\pm\tilde{t}}{\sqrt{1-\tilde{t}^2}}$ with $|\tilde{t}| < 1$.

We discuss two cases for Problem (8).

**Case ($a$)**. $\tilde{c} = \frac{1}{\sqrt{1-\tilde{t}^2}}, \tilde{s} = \frac{\tilde{t}}{\sqrt{1-\tilde{t}^2}}$. Problem (8) is equivalent to the following problem: $\bar{\mu}_{+} = \arg\min_{\mu} \frac{a+\tilde{t}\,b}{\sqrt{1-\tilde{t}^2}} + \frac{w+\tilde{t}\,d}{1-\tilde{t}^2} - e$. Therefore, the optimal solution $\bar{\mu}_{+}$can be computed as:

$$\cosh(\bar{\mu}_{+}) = \frac{1}{\sqrt{1-(\bar{t}_{+})^2}}, \text{ and } \sinh(\bar{\mu}_{+}) = \frac{\bar{t}_{+}}{\sqrt{1-(\bar{t}_{+})^2}} \qquad (14)$$

**Case (b)**. $\tilde{c} = \frac{-1}{\sqrt{1-\tilde{t}^2}}, \tilde{s} = \frac{-\tilde{t}}{\sqrt{1-\tilde{t}^2}}$. Problem (8) is equivalent to the following problem: $\bar{\mu}_- = \arg\min_\mu \frac{-a-\tilde{t}\,b}{\sqrt{1-\tilde{t}^2}} + \frac{w+\tilde{t}\,d}{1-\tilde{t}^2} - e$. Therefore, the optimal solution $\bar{\mu}_-$ can be computed as:

$$\cosh(\bar{\mu}_-) = \frac{-1}{\sqrt{1-(\bar{t}_-)^2}}, \text{ and } \sinh(\bar{\mu}_-) = \frac{-\bar{t}_-}{\sqrt{1-(\bar{t}_-)^2}}. \tag{15}$$

We define the objective function as: $\breve{F}(\tilde{c},\tilde{s}) \triangleq a\tilde{c} + b\tilde{s} + c\tilde{c}^2 + d\tilde{c}\tilde{s} + e\tilde{s}^2$. In view of (14) and (15), the optimal solution pair $[cosh(\bar{\mu}, sinh(\bar{\mu})]$ for problem (8) can be computed as:

$$[\cosh(\bar{\mu}), \sinh(\bar{\mu})] = \arg\min_{[c,s]} \breve{F}(c,s),$$

$$\text{s. t. } [c,s] \in \{[\cosh(\bar{\mu}_+), \sinh(\bar{\mu}_+)], [\cosh(\bar{\mu}_-), \sinh(\bar{\mu}_-)]\}$$

Importantly, it is not necessary to compute the values $\bar{\mu}_+$ for (14) and $\bar{\mu}_-$ for (15). $\qquad\square$

## C.3  Proof of Lemma 2.4

*Proof.* The objective function for $\mathrm{B}^t_{(i)}$ as in Equation (3) is formulated as :

$$f(\mathbf{X}^t) + \tfrac{1}{2}\|\mathbf{V}_i - \mathbf{I}\|^2_{\mathbf{Q}+\theta\mathbf{I}} + \langle \mathbf{V}_i - \mathbf{I}, [\nabla f(\mathbf{X}^t)(\mathbf{X}^t)^\mathsf{T}]_{\mathrm{B}^t_{(i)}\mathrm{B}^t_{(i)}} \rangle$$

**Part (1).** For the part of $\tfrac{1}{2}\|\mathbf{V}_i - \mathbf{I}\|^2_{\mathbf{Q}+\theta\mathbf{I}}$, it is obviously irrelevant.

**Part (2).** For the part of $\langle \mathbf{V}_i - \mathbf{I}, [\nabla f(\mathbf{X}^t)(\mathbf{X}^t)^\mathsf{T}]_{\mathrm{B}^t_{(i)}\mathrm{B}^t_{(i)}} \rangle$, we note that $[\nabla f(\mathbf{X}^t)(\mathbf{X}^t)^\top]_{\mathrm{B}^t_{(i)}\mathrm{B}^t_{(i)}} = [\nabla f(\mathbf{X}^t)](\mathrm{B}^t_{(i)},:)[(\mathbf{X}^t)^\top](:,\mathrm{B}^t_{(i)}) = [\nabla f(\mathbf{X}^t)](\mathrm{B}^t_{(i)},:)[(\mathbf{X}^t)(\mathrm{B}^t_{(i)},:)]^\top$, which just use the information of block $\mathrm{B}^t_{(i)}$. The proof ends. $\qquad\square$

## C.4  Proof of Lemma 2.5

*Proof.* Part (**a**). For the purpose of analysis, we define the following: $\forall i \in [\frac{n}{2}], \mathbf{K}_i = \mathbf{U}_{\mathrm{B}_{(i)}}(\mathbf{V}_i - \mathbf{I}_2)\mathbf{U}^\top_{\mathrm{B}_{(i)}}\mathbf{X}$.

$$\|\sum_{i=1}^{\frac{n}{2}} [\mathbf{U}_{\mathrm{B}_{(i)}}(\mathbf{V}_i - \mathbf{I}_2)\mathbf{U}^\top_{\mathrm{B}_{(i)}}\mathbf{X}]\|^2_\mathsf{F} \overset{①}{=} \left\| \begin{bmatrix} \mathbf{K}_1 \\ \mathbf{K}_2 \\ \vdots \\ \mathbf{K}_{\frac{n}{2}} \end{bmatrix} \right\|^2_\mathsf{F}$$

$$\overset{②}{=} \|\mathbf{K}_1\|^2_\mathsf{F} + \|\mathbf{K}_2\|^2_\mathsf{F} + \cdots + \|\mathbf{K}_{\frac{n}{2}}\|^2_\mathsf{F}$$

$$\overset{③}{=} \sum_{i=1}^{\frac{n}{2}} [\|\mathbf{U}_{\mathrm{B}_{(i)}}(\mathbf{V}_i - \mathbf{I}_2)\mathbf{U}^\top_{\mathrm{B}_{(i)}}\mathbf{X}\|^2_\mathsf{F}]$$

where step ① uses the definition of $\mathbf{K}_i$ and the assumption that $\mathrm{B} \in \Upsilon$; step ② uses the definition of Squared Frobenius Norm; step ③ uses the definition of $\mathbf{K}_i$.

Part (**b**). Using the update rule for $\mathbf{X}^+ = \mathbf{X} + [\sum_{i=1}^{n/2} \mathbf{U}_{\mathrm{B}_{(i)}}(\mathbf{V}_i - \mathbf{I}_2)\mathbf{U}^\top_{\mathrm{B}_{(i)}}]\mathbf{X} \in \mathbb{R}^{n \times n}$, we have the following inequalities:

$$\|\mathbf{X}^+ - \mathbf{X}\|^2_\mathsf{F} = \|[\sum_{i=1}^{n/2} \mathbf{U}_{\mathrm{B}_{(i)}}(\mathbf{V}_i - \mathbf{I}_2)\mathbf{U}^\top_{\mathrm{B}_{(i)}}]\mathbf{X}\|^2_\mathsf{F} \tag{16}$$

$$\overset{①}{=} \sum_{i=1}^{n/2} \|[\mathbf{U}_{\mathrm{B}_{(i)}}(\mathbf{V}_i - \mathbf{I}_2)\mathbf{U}^\top_{\mathrm{B}_{(i)}}]\mathbf{X}\|^2_\mathsf{F} \tag{17}$$

$$\overset{②}{\leq} \sum_{i=1}^{n/2} \|\mathbf{V}_i - \mathbf{I}_2\|^2_\mathsf{F} \cdot \|\mathbf{X}\|^2_\mathsf{F}, \tag{18}$$

where step ① uses the conclusion of Part (**a**); step ② uses the same proof process of Part (**b**) of lemma 2.1.

Part (*c*). We derive the following results:

$$\frac{1}{2}\|\mathbf{X}^+ - \mathbf{X}\|_{\mathbf{H}}^2 = \frac{1}{2}\|[\sum_{i=1}^{n/2} \mathbf{U}_{\mathsf{B}_{(i)}}(\mathbf{V}_i - \mathbf{I}_2)\mathbf{U}_{\mathsf{B}_{(i)}}^\top]\mathbf{X}\|_{\mathbf{H}}^2$$

$$\overset{①}{=} \frac{1}{2}\sum_{i=1}^{n/2} \|[\mathbf{U}_{\mathsf{B}_{(i)}}(\mathbf{V}_i - \mathbf{I}_2)\mathbf{U}_{\mathsf{B}_{(i)}}^\top]\mathbf{X}\|_{\mathbf{H}}^2$$

$$\overset{②}{\leq} \frac{1}{2}\sum_{i=1}^{n/2} \|\mathbf{V}_i - \mathbf{I}_2\|_{\mathbf{Q}}^2$$

where step ① uses the conclusion of Part (*a*); step ② uses the same proof process of Part (*c*) of lemma 2.1.

Part (*d*). We derive the following results:

$$\sum_{i=1}^{n/2}\langle \mathbf{V}_i - \mathbf{I}_2, [(\nabla f(\mathbf{X}) - \tilde{\mathbf{G}})\mathbf{X}^\top]_{\mathsf{B}_i\mathsf{B}_i}\rangle$$

$$= \sum_{i=1}^{n/2}\langle [\mathbf{U}_{\mathsf{B}_{(i)}}(\mathbf{V}_i - \mathbf{I}_2)\mathbf{U}_{\mathsf{B}_{(i)}}^\top]\mathbf{X}, [(\nabla f(\mathbf{X}) - \tilde{\mathbf{G}})]\rangle$$

$$= \langle \mathbf{X}^+ - \mathbf{X}, [(\nabla f(\mathbf{X}) - \tilde{\mathbf{G}})]\rangle$$

$$\overset{①}{\leq} \frac{1}{2}\|\mathbf{X}^+ - \mathbf{X}\|_{\mathsf{F}}^2 + \frac{1}{2}\|[\nabla f(\mathbf{X}) - \tilde{\mathbf{G}}]\|_{\mathsf{F}}^2$$

$$\overset{②}{\leq} \frac{1}{2}\|\mathbf{X}\|_{\mathsf{F}}^2 \sum_{i=1}^{n/2}\|\mathbf{V}_i - \mathbf{I}_2\|_{\mathsf{F}}^2 + \frac{1}{2}\|[\nabla f(\mathbf{X}) - \tilde{\mathbf{G}}]\|_{\mathsf{F}}^2 \quad (19)$$

where step ① uses $\forall \mathbf{A}, \mathbf{B}, \frac{1}{2}\|\mathbf{A} - \mathbf{B}\|_{\mathsf{F}}^2 = \frac{1}{2}\|\mathbf{A}\|_{\mathsf{F}}^2 + \frac{1}{2}\|\mathbf{B}\|_{\mathsf{F}}^2 - \langle \mathbf{A}, \mathbf{B}\rangle \geq 0$, with $\mathbf{A} = \|\mathbf{X}^+ - \mathbf{X}\|_{\mathsf{F}}^2$ and $\mathbf{B} = \|[\nabla f(\mathbf{X}) - \tilde{\mathbf{G}}]\|_{\mathsf{F}}^2$; step ② uses the conclusion of Part (*b*). □

# D  PROOFS FOR SECTION 3

## D.1  PROOF OF LEMMA 3.1

*Proof.* We consider the Lagrangian function of problem (1):

$$\mathcal{L}(\mathbf{X}, \Lambda) = f(\mathbf{X}) - \frac{1}{2}\langle \Lambda, \mathbf{X}^\mathsf{T}\mathbf{J}\mathbf{X} - \mathbf{J}\rangle. \quad (20)$$

Setting the gradient of $\mathcal{L}(\mathbf{X}, \Lambda)$ *w.r.t.* $\mathbf{X}$ to zero yields:

$$\nabla f(\mathbf{X}) - \mathbf{J}\mathbf{X}\Lambda = \mathbf{0}. \quad (21)$$

**Part (a)**. Multiplying both sides by $\mathbf{X}^\mathsf{T}$ and using the fact that $\mathbf{X}^\mathsf{T}\mathbf{J}\mathbf{X} = \mathbf{J}$, we have $\mathbf{J}\Lambda = \mathbf{X}^\mathsf{T}\nabla f(\mathbf{X})$. Multiplying both sides by $\mathbf{J}^\mathsf{T}$ and using $\mathbf{J}^\top \mathbf{J} = \mathbf{I}$, we have $\Lambda = \mathbf{J}\mathbf{X}^\top\nabla f(\mathbf{X})$. Since $\Lambda$ is symmetric, we have $\Lambda = \nabla f(\mathbf{X})^\mathsf{T}\mathbf{X}\mathbf{J}$. Putting this equality into Equality (21) yields the following first-order optimality condition for Problem (1):

$$\nabla f(\mathbf{X}) = \mathbf{J}\mathbf{X}[\nabla f(\mathbf{X})]^\mathsf{T}\mathbf{X}\mathbf{J}. \quad (22)$$

**Part (b)**. We let $\mathbf{G} = \nabla f(\mathbf{X})$. We derive the following results:

$$\mathbf{G} = \mathbf{J}\mathbf{X}\mathbf{G}^\mathsf{T}\mathbf{X}\mathbf{J} \quad \overset{①}{\Rightarrow} \quad \mathbf{J}\mathbf{X}^\mathsf{T} \cdot \mathbf{G} = \mathbf{J}\mathbf{X}^\mathsf{T} \cdot \mathbf{J}\mathbf{X}\mathbf{G}^\mathsf{T}\mathbf{X}\mathbf{J}$$

$$\overset{②}{\Rightarrow} \quad \mathbf{J}\mathbf{X}^\mathsf{T}\mathbf{G} = \mathbf{G}^\mathsf{T}\mathbf{X}\mathbf{J}$$

$$\overset{③}{\Rightarrow} \quad \mathbf{X}(\mathbf{J}\mathbf{X}^\mathsf{T}\mathbf{G})\mathbf{X}^\mathsf{T} = \mathbf{X}(\mathbf{G}^\mathsf{T}\mathbf{X}\mathbf{J})\mathbf{X}^\mathsf{T}$$

$$\overset{④}{\Rightarrow} \quad \mathbf{X}\underbrace{\mathbf{J}\mathbf{X}^\mathsf{T}\mathbf{G}\mathbf{X}^\mathsf{T}\mathbf{J}}_{\triangleq \mathbf{G}^\mathsf{T}}\mathbf{J} = \mathbf{J}\underbrace{\mathbf{J}\mathbf{X}\mathbf{G}^\mathsf{T}\mathbf{X}\mathbf{J}}_{\triangleq \mathbf{G}}\mathbf{X}^\mathsf{T} \quad (23)$$

$$\overset{⑤}{\Rightarrow} \quad (\mathbf{X}\mathbf{G}^\mathsf{T}\mathbf{J}) \cdot \mathbf{J}\mathbf{X} = (\mathbf{J}\mathbf{G}\mathbf{X}^\mathsf{T}) \cdot \mathbf{J}\mathbf{X}$$

$$\overset{⑥}{\Rightarrow} \quad \mathbf{X}\mathbf{G}^\mathsf{T}\mathbf{X} = \mathbf{J}\mathbf{G}\mathbf{J}$$

$$\overset{⑦}{\Rightarrow} \quad \mathbf{J}\mathbf{X}\mathbf{G}^\mathsf{T}\mathbf{X}\mathbf{J} = \mathbf{G},$$

where step ① uses the results of left-multiplying both sides by $\mathbf{J}\mathbf{X}^\mathsf{T}$; step ② uses $\mathbf{J} \cdot \mathbf{X}^\mathsf{T}\mathbf{J}\mathbf{X} = \mathbf{J}\mathbf{J} = \mathbf{I}$; step ③ uses the results of left-multiplying both sides by $\mathbf{X}$ and subsequently right-multiplying them by $\mathbf{X}^\mathsf{T}$; ④ uses $\mathbf{G} = \mathbf{J}\mathbf{X}\mathbf{G}^\mathsf{T}\mathbf{X}\mathbf{J}$; step ⑤ uses the the results of right-multiplying both sides by $\mathbf{J}\mathbf{X}$; step ⑥ uses $\mathbf{J}\mathbf{J} = \mathbf{I}$ and $\mathbf{X}^\mathsf{T}\mathbf{J}\mathbf{X} = \mathbf{J}$; step ⑦ uses the results of left-multiply both sides by $\mathbf{J}$ and right-multiplied by $\mathbf{J}$.

Given Equality (23), we conclude that the critical point condition is equivalent to the requirement that the matrix $\mathbf{X}\nabla f(\check{\mathbf{X}})^{\mathsf{T}}\mathbf{J}$ is symmetric, which is expressed as $\mathbf{X}\mathbf{G}^{\mathsf{T}}\mathbf{J} = [\mathbf{X}\mathbf{G}^{\mathsf{T}}\mathbf{J}]^{\mathsf{T}}$. $\qquad\square$

### D.2 Proof of Theorem 3.3

*Proof.* We use $\ddot{\mathbf{X}}$ and $\check{\mathbf{X}}$ to denote any BS-point and critical point, respectively.

For all $\mathtt{B} \in \Omega \triangleq \{\mathcal{B}_1, \mathcal{B}_2, \ldots, \mathcal{B}_{\mathrm{C}_n^2}\}$, we have:

$$\mathbf{I}_2 \in \arg\min_{\mathbf{V} \in \mathcal{J}_{\mathtt{B}}} \mathcal{G}(\mathbf{V}; \ddot{\mathbf{X}}, \mathtt{B}).$$

where $\mathcal{G}(\mathbf{V}; \mathbf{X}, \mathtt{B}) \triangleq f(\mathbf{X}) + \frac{1}{2}\|\mathbf{V} - \mathbf{I}_2\|_{\mathbf{Q}+\theta\mathbf{I}}^2 + \langle\mathbf{V} - \mathbf{I}, [\nabla f(\mathbf{X})(\mathbf{X})^{\mathsf{T}}]_{\mathtt{BB}}\rangle$.

The Euclidean gradient of $\mathcal{G}(\mathbf{V}; \ddot{\mathbf{X}}, \mathtt{B})$ can be computed as:

$$\ddot{\mathbf{G}} \triangleq \mathrm{mat}((\mathbf{Q} + \theta\mathbf{I}_2)\,\mathrm{vec}(\mathbf{V} - \mathbf{I}_2)) + [\nabla f(\ddot{\mathbf{X}})(\ddot{\mathbf{X}})^{\mathsf{T}}]_{\mathtt{BB}}. \tag{24}$$

Given Lemma 3.1, we set the Riemannian gradient of $\mathcal{G}(\mathbf{V}; \ddot{\mathbf{X}}, \mathtt{B})$ *w.r.t.* $\mathbf{V}$ to zero, leading to the following first-order optimality condition:

$$\mathbf{0} = \nabla_{\mathcal{J}}\mathcal{G}(\mathbf{V}; \ddot{\mathbf{X}}, \mathtt{B}) = \ddot{\mathbf{G}} - \mathbf{U}_{\mathtt{B}}^{\mathsf{T}}\mathbf{J}\mathbf{V}\ddot{\mathbf{G}}^{\mathsf{T}}\mathbf{V}\mathbf{J}\mathbf{U}_{\mathtt{B}}. \tag{25}$$

Letting $\mathbf{V} = \mathbf{I}_2$, and using the definition of $\ddot{\mathbf{G}}$, we have:

$$\mathbf{0}_{2,2} = [\nabla f(\mathbf{X})(\mathbf{X})^{\mathsf{T}}]_{\mathtt{BB}} - \mathbf{J}_{\mathtt{BB}}\ddot{\mathbf{G}}^{\mathsf{T}}\mathbf{J}_{\mathtt{BB}}, \ \forall \mathtt{B} \in \{\mathcal{B}_i\}_{i=1}^{\mathbf{C}_n^2}$$

$$\Rightarrow \quad \mathbf{0}_{2,2} = \mathbf{U}_{\mathtt{B}}^{\mathsf{T}}[\nabla f(\ddot{\mathbf{X}})\ddot{\mathbf{X}}^{\mathsf{T}}]\mathbf{U}_{\mathtt{B}} - \mathbf{J}_{\mathtt{BB}}\mathbf{U}_{\mathtt{B}}^{\mathsf{T}}[\ddot{\mathbf{X}}\nabla f(\ddot{\mathbf{X}})^{\mathsf{T}}]\mathbf{U}_{\mathtt{B}}\mathbf{J}_{\mathtt{BB}}, \ \forall \mathtt{B} \in \{\mathcal{B}_i\}_{i=1}^{\mathbf{C}_n^2}$$

$$\overset{\text{①}}{\Rightarrow} \quad \mathbf{0}_{2,2} = \mathbf{U}_{\mathtt{B}}^{\mathsf{T}}[\nabla f(\ddot{\mathbf{X}})\ddot{\mathbf{X}}^{\mathsf{T}}]\mathbf{U}_{\mathtt{B}} - \mathbf{U}_{\mathtt{B}}^{\mathsf{T}}\mathbf{J}[\ddot{\mathbf{X}}\nabla f(\ddot{\mathbf{X}})^{\mathsf{T}}]\mathbf{J}\mathbf{U}_{\mathtt{B}}, \ \forall \mathtt{B} \in \{\mathcal{B}_i\}_{i=1}^{\mathbf{C}_n^2}$$

$$\overset{\text{②}}{\Rightarrow} \quad \mathbf{0}_{n,n} = [\nabla f(\ddot{\mathbf{X}})\ddot{\mathbf{X}}^{\mathsf{T}}] - \mathbf{J}[\ddot{\mathbf{X}}\nabla f(\ddot{\mathbf{X}})^{\mathsf{T}}]\mathbf{J},$$

$$\overset{\text{③}}{\Rightarrow} \quad [\mathbf{J}\nabla f(\ddot{\mathbf{X}})\ddot{\mathbf{X}}^{\mathsf{T}}] = [\mathbf{J}\nabla f(\ddot{\mathbf{X}})\ddot{\mathbf{X}}^{\mathsf{T}}]^{\mathsf{T}},$$

where step ① uses $\mathbf{U}_{\mathtt{B}}^{\mathsf{T}}\mathbf{J} = \mathbf{J}_{\mathtt{BB}}\mathbf{U}_{\mathtt{B}}^{\mathsf{T}}$ and $\mathbf{J}\mathbf{U}_{\mathtt{B}} = \mathbf{U}_{\mathtt{B}}\mathbf{J}_{\mathtt{BB}}$; step ② uses the the following results for any $\mathbf{W} \in \mathbb{R}^{n \times n}$:

$$(\forall \mathtt{B} \in \{\mathcal{B}_i\}_{i=1}^{\mathbf{C}_n^2}, \mathbf{0}_{2,2} = \mathbf{U}_{\mathtt{B}}^{\mathsf{T}}\mathbf{W}\mathbf{U}_{\mathtt{B}} = \mathbf{W}_{\mathtt{BB}}) \Rightarrow (\mathbf{W} = \mathbf{0}_{n,n}); \tag{26}$$

step ③ uses the fact that both sides are left-multiplied by $\mathbf{J}$. We conclude that the matrix $\mathbf{J}\nabla f(\ddot{\mathbf{X}})\ddot{\mathbf{X}}^{\mathsf{T}}$ is symmetric. Using Claim (*b*) of Lemma 3.1, we conclude that $\ddot{\mathbf{X}}$ is a also a critical point.

Notably, the condition in Equation (25) is a necessary but not sufficient condition. This is because BS-point is the global minimum of Problem: $\arg\min_{\mathbf{V} \in \mathcal{J}_{\mathtt{B}}} \mathcal{G}(\mathbf{V}; \ddot{\mathbf{X}}, \mathtt{B})$, according to Definition 3.2. $\qquad\square$

## E  Proofs for Section 4

### E.1 Proof of Lemma 4.5

*Proof.* By the definition of $\tilde{\mathbf{G}}^t$, we have

$$\mathbb{E}_{\iota^t}[\|\tilde{\mathbf{G}}^t - \nabla f(\mathbf{X}^t)\|_{\mathsf{F}}^2]$$

$$\overset{\text{①}}{=} \quad p\mathbb{E}_{\iota^t}[\|\tfrac{1}{b}\sum_{i=1}^b \nabla f_i(\mathbf{X}^t) - \nabla f(\mathbf{X}^t)\|_{\mathsf{F}}^2] +$$

$$(1-p)\mathbb{E}_{\iota^t}[\|\tilde{\mathbf{G}}^{t-1} + \tfrac{1}{b'}\sum_{i=1}^{b'}(\nabla f_i(\mathbf{X}^t) - \nabla f_i(\mathbf{X}^{t-1})) - \nabla f(\mathbf{X}^t)\|_{\mathsf{F}}^2]$$

$$\overset{\text{②}}{=} \quad p\mathbb{E}_{\iota^t}[\|\tfrac{1}{b}\sum_{i=1}^b \nabla f_i(\mathbf{X}^t) - \nabla f(\mathbf{X}^t)\|_{\mathsf{F}}^2] + (1-p)\mathbb{E}_{\iota^{t-1}}[\|\tilde{\mathbf{G}}^{t-1} - \nabla f(\mathbf{X}^{t-1})\|_{\mathsf{F}}^2]$$

$$+ (1-p)\mathbb{E}_{\iota^t}[\|\tfrac{1}{b'}\sum_{i=1}^{b'}(\nabla f_i(\mathbf{X}^t) - \nabla f_i(\mathbf{X}^{t-1})) - \nabla f(\mathbf{X}^t) + \nabla f(\mathbf{X}^{t-1})\|_{\mathsf{F}}^2]$$

where step ① uses formula (9); step ② uses that $\tilde{\mathbf{G}}^{t-1} - \nabla f(\mathbf{X}^{t-1})$ is measurable w.r.t. $\iota^{t-1}$ and $\mathbb{E}_{\iota^t}[\|\frac{1}{b'}\sum_{i=1}^{b'}(\nabla f_i(\mathbf{X}^t) - \nabla f_i(\mathbf{X}^{t-1})) - \nabla f(\mathbf{X}^t) + \nabla f(\mathbf{X}^{t-1})\|_{\mathsf{F}}^2] = 0$. We further have

$$\mathbb{E}_{\iota^t}[\|\tilde{\mathbf{G}}^t - \nabla f(\mathbf{X}^t)\|_{\mathsf{F}}^2]$$

$$\overset{①}{\leq} p\mathbb{E}_{\iota^t}[\|\tfrac{1}{b}\sum_{i=1}^b \nabla f_i(\mathbf{X}^t) - \nabla f(\mathbf{X}^t)\|_{\mathsf{F}}^2] + (1-p)\mathbb{E}_{\iota^{t-1}}[\|\tilde{\mathbf{G}}^{t-1} - \nabla f(\mathbf{X}^{t-1})\|_{\mathsf{F}}^2]$$

$$+ (1-p)\mathbb{E}_{\iota^t}[\|\tfrac{1}{b'}\sum_{i=1}^{b'}(\nabla f_i(\mathbf{X}^t) - \nabla f_i(\mathbf{X}^{t-1}))\|_{\mathsf{F}}^2]$$

$$\overset{②}{\leq} \tfrac{p(N-b)}{b(N-1)}\mathbb{E}_{\iota^t}[\|\nabla f_i(\mathbf{X}^t) - \nabla f(\mathbf{X}^t)\|_{\mathsf{F}}^2] + (1-p)\mathbb{E}_{\iota^{t-1}}\|\tilde{\mathbf{G}}^{t-1} - \nabla f(\mathbf{X}^{t-1})\|_{\mathsf{F}}^2$$

$$+ \tfrac{1-p}{b'}\mathbb{E}_{\iota^{t-1}}[\|\nabla f_i(\mathbf{X}^t) - \nabla f_i(\mathbf{X}^{t-1})\|_{\mathsf{F}}^2]$$

$$\overset{③}{\leq} \tfrac{p(N-b)}{b(N-1)}\sigma^2 + (1-p)\mathbb{E}_{\iota^{t-1}}[\|\tilde{\mathbf{G}}^{t-1} - \nabla f(\mathbf{X}^{t-1})\|_{\mathsf{F}}^2]$$

$$+ \tfrac{L_f^2 \overline{\mathbf{X}}^2(1-p)}{b'}\mathbb{E}_{\iota^{t-1}}[\sum_{i=1}^{n/2} \|\mathbf{V}_i^{t-1} - \mathbf{I}_2\|_{\mathsf{F}}^2] \qquad (27)$$

where step ① uses that for any random variable $\mathbf{X}, \mathbb{E}[(\mathbf{X} - \mathbb{E}[\mathbf{X}])^2] \leq \mathbb{E}[\mathbf{X}^2]$; step ② uses lemma A.2; step ③ uses assumption 4.3, Inequality (2) and Part ($b$) of lemma 2.5. □

### E.2    Proof of theorem 4.6

*Proof.* For simplicity, we use $\mathsf{B}$ instead of $\mathsf{B}^t$. We will show that the following inequality holds :

$$\tfrac{\theta}{2}\|\bar{\mathbf{V}}^t - \mathbf{I}_2\|_{\mathsf{F}}^2 \leq f(\mathbf{X}^t) - f(\mathbf{X}^{t+1}). \qquad (28)$$

Since $\bar{\mathbf{V}}^t$ is the global optimal solution of Problem (5), we have:

$$\mathcal{G}(\bar{\mathbf{V}}^t; \mathbf{X}^t, \mathsf{B}) \leq \mathcal{G}(\mathbf{V}; \mathbf{X}^t, \mathsf{B}), \mathbf{V} \in \mathcal{J}_{\mathsf{B}}$$

Letting $\mathbf{V} = \mathbf{I}_2$, we have: $\mathcal{G}(\bar{\mathbf{V}}^t; \mathbf{X}^t, \mathsf{B}) \leq \mathcal{G}(\mathbf{I}_2; \mathbf{X}^t, \mathsf{B})$. We further obtain:

$$\tfrac{1}{2}\|\bar{\mathbf{V}}^t - \mathbf{I}_2\|_{\mathbf{Q}+\theta\mathbf{I}}^2 + \langle \bar{\mathbf{V}}^t - \mathbf{I}, [\nabla f(\mathbf{X}^t)(\mathbf{X}^t)^\top]_{\mathsf{BB}}\rangle \leq 0. \qquad (29)$$

Using Inequality (2) with $N = 1$ and Part ($c$) of Lemma 2.1, we have:

$$f(\mathbf{X}^{t+1}) \leq f(\mathbf{X}^t) + \langle \bar{\mathbf{V}}^t - \mathbf{I}_2, [\nabla f(\mathbf{X}^t)(\mathbf{X}^t)^\top]_{\mathsf{BB}}\rangle + \tfrac{1}{2}\|\bar{\mathbf{V}}^t - \mathbf{I}_2\|_{\mathbf{Q}}^2. \qquad (30)$$

Adding Inequality (29) and (30) together, we obtain the inequality in (28). Using the result of Part ($b$) in Lemma 2.1 that $\frac{\|\mathbf{X}^+ - \mathbf{X}\|_{\mathsf{F}}^2}{\|\mathbf{X}\|_{\mathsf{F}}^2} \leq \|\mathbf{V} - \mathbf{I}_2\|_{\mathsf{F}}^2$, we have the following sufficient decrease condition:

$$f(\mathbf{X}^{t+1}) - f(\mathbf{X}^t) \leq -\tfrac{\theta}{2}\|\bar{\mathbf{V}}^t - \mathbf{I}_2\|_{\mathsf{F}}^2 \leq -\tfrac{\theta}{2}\tfrac{\|\mathbf{X}^{t+1} - \mathbf{X}^t\|_{\mathsf{F}}^2}{\|\mathbf{X}^t\|_{\mathsf{F}}^2} \qquad (31)$$

We now prove the global convergence. Taking the expectation for Inequality (31), we obtain a lower bound on the expected progress made by each iteration for Algorithm 1:

$$\mathbb{E}_{\xi^{t+1}}[f(\mathbf{X}^{t+1})] - \mathbb{E}_{\xi^t}[f(\mathbf{X}^t)] \leq -\mathbb{E}_{\xi^t}[\tfrac{\theta}{2}\|\bar{\mathbf{V}}^t - \mathbf{I}_2\|_{\mathsf{F}}^2].$$

Summing up the inequality above over $t = 0, 1, \ldots, T$, we have:

$$\mathbb{E}_{\xi^t}[\tfrac{\theta}{2}\sum_{t=0}^T \|\bar{\mathbf{V}}^t - \mathbf{I}_2\|_{\mathsf{F}}^2] \leq f(\mathbf{X}^0) - \mathbb{E}_{\xi^{T+1}}[f(\mathbf{X}^{T+1})] \leq f(\mathbf{X}^0) - f(\bar{\mathbf{X}}).$$

As a result, there exists an index $\bar{t}$ with $0 \leq \bar{t} \leq T$ such that

$$\mathbb{E}_{\xi^{\bar{t}}}[\|\bar{\mathbf{V}}^{\bar{t}} - \mathbf{I}_2\|_{\mathsf{F}}^2] \leq \tfrac{2}{\theta(T+1)}[f(\mathbf{X}^0) - f(\bar{\mathbf{X}})]. \qquad (32)$$

Furthermore, for any $t$, we have:

$$\mathcal{E}(\mathbf{X}^t) \triangleq \tfrac{1}{C_n^2}\sum_{i=1}^{C_n^2} \mathrm{dist}(\mathbf{I}_2, \arg\min_{\mathbf{V}} \mathcal{G}(\mathbf{V}; \mathbf{X}^t, \mathcal{B}_i))^2 = \mathbb{E}_{\xi^t}[\|\bar{\mathbf{V}}^t - \mathbf{I}_2\|_{\mathsf{F}}^2] \qquad (33)$$

Combining Inequality (32) and equality (33), we have the following result:

$$\mathbb{E}_{\xi^t}[\|\bar{\mathbf{V}}^t - \mathbf{I}_2\|_{\mathsf{F}}^2] = \mathcal{E}(\mathbf{X}^{\bar{t}}) \leq \tfrac{2(f(\mathbf{X}^0) - f(\bar{\mathbf{X}}))}{\theta(T+1)}. \qquad (34)$$

We will give the arithmetic operations of **GS-JOBCD**. By the chosen parameters and Inequality (34), we have

$$\mathcal{E}(\mathbf{X}^{\bar{t}}) \leq \frac{2(f(\mathbf{X}^0) - f(\bar{\mathbf{X}}))}{\theta(T+1)} \leq \epsilon.$$

We define $\Delta_0 = f(\mathbf{X}_0) - f(\bar{\mathbf{X}})$ and set $T + 1 = \frac{2\Delta_0}{\epsilon\theta}$. Denoting $m_t$ to be the number of arithmetic operations at $t$-th iteration, we have for $t \geq 1$:

$$\mathbb{E}_{\xi^t}[m_t] = \mathcal{O}(2N).$$

Then we have for $t \geq 1$, the total number of arithmetic operations $M^T$ in $T$ iterations to obtain $\epsilon$-BS-point is

$$\mathbb{E}_{\xi^T}[M^T] = \mathbb{E}_{\xi^t}[\textstyle\sum_{t=0}^{T} m_t] = 2(T+1)N = \mathcal{O}((T+1)N).$$

We have $(T+1)N = N\frac{2\Delta_0}{\epsilon\theta} \leq \mathcal{O}(\frac{\Delta_0 N}{\epsilon})$. $\qquad\qquad\qquad\qquad\qquad\qquad\qquad\square$

### E.3 Proof of Theorem 4.7

*Proof.* For simplicity, we use $\mathtt{B}$ instead of $\mathtt{B}^t$. Defining $\bar{\mathbf{V}}_{:}^{\ t}$ as the global optimal solution of $\arg\min_{\mathbf{V}_:} \mathcal{T}(\mathbf{V}_:; \mathbf{X}^t, \mathtt{B})$, we have:

$$\mathcal{T}(\bar{\mathbf{V}}_{:}^{\ t}; \mathbf{X}^t, \mathtt{B}) \leq \mathcal{T}(\mathbf{V}_:; \mathbf{X}^t, \mathtt{B}), \forall i, \mathbf{V}_i \in \mathcal{J}_{\mathtt{B}_{(i)}}$$

Letting $\mathbf{V}_i = \mathbf{I}_2, \forall i$, we have: $\mathcal{T}(\bar{\mathbf{V}}_{:}^{\ t}; \mathbf{X}^t, \mathtt{B}) \leq \mathcal{T}(\mathbf{I}_2; \mathbf{X}^t, \mathtt{B})$. We further obtain:

$$\tfrac{1}{2}\textstyle\sum_{i=1}^{n/2} \|\bar{\mathbf{V}}_i^t - \mathbf{I}_2\|_{(\zeta+\theta)\mathbf{I}}^2 + \sum_{i=1}^{n/2}\langle \bar{\mathbf{V}}_i^t - \mathbf{I}, [\tilde{\mathbf{G}}^t(\mathbf{X}^t)^\top]_{\mathtt{B}_{(i)}\mathtt{B}_{(i)}}\rangle \leq 0. \tag{35}$$

Using the results of telescoping Inequality (2) over $i$ from 1 to $N$ with Part (**c**) of Lemma 2.5, we have:

$$f(\mathbf{X}^{t+1}) \leq f(\mathbf{X}^t) + \textstyle\sum_{i=1}^{n/2}\langle\bar{\mathbf{V}}_i - \mathbf{I}_2, [\nabla f(\mathbf{X})\mathbf{X}^\top]_{\mathtt{B}_{(i)}\mathtt{B}_{(i)}}\rangle + \tfrac{1}{2}\sum_{i=1}^{n/2}\|\bar{\mathbf{V}}_i - \mathbf{I}_2\|_{\zeta\mathbf{I}}^2. \tag{36}$$

Adding inequality (35), and (36) together, we obtain the inequality in (37).

$$\begin{aligned}
&\tfrac{\theta}{2}\textstyle\sum_{i=1}^{n/2}\|\bar{\mathbf{V}}_i^t - \mathbf{I}_2\|_{\mathsf{F}}^2 \\
\leq\ & f(\mathbf{X}^t) - f(\mathbf{X}^{t+1}) + \textstyle\sum_{i=1}^{n/2}\langle\bar{\mathbf{V}}_i^t - \mathbf{I}, [(\nabla f(\mathbf{X}^t) - \tilde{\mathbf{G}}^t)(\mathbf{X}^t)^\top]_{\mathtt{B}_{(i)}\mathtt{B}_{(i)}}\rangle \\
\overset{①}{\leq}\ & f(\mathbf{X}^t) - f(\mathbf{X}^{t+1}) + \tfrac{1}{2}\|\mathbf{X}^t\|_{\mathsf{F}}^2\textstyle\sum_{i=1}^{n/2}\|\bar{\mathbf{V}}_i^t - \mathbf{I}_2\|_{\mathsf{F}}^2 + \tfrac{1}{2}\|[\nabla f(\mathbf{X}^t) - \tilde{\mathbf{G}}^t]\|_{\mathsf{F}}^2
\end{aligned} \tag{37}$$

where step ① uses Part (**d**) of Lemma 2.5.
Taking expectation on both sides of inequality (37) with respect to all randomness of the algorithm, and adding the inequality in Lemma 4.5 $\times \frac{1}{2p}$ to (37), we have:

$$\begin{aligned}
&(\tfrac{\theta - \overline{\mathbf{X}}^2}{2} - \tfrac{L_f^2\overline{\mathbf{X}}^2(1-p)}{2pb'})\mathbb{E}_{\iota^t}[\textstyle\sum_{i=1}^{n/2}\|\bar{\mathbf{V}}_i^t - \mathbf{I}_2\|_{\mathsf{F}}^2] \\
\leq\ & \mathbb{E}_{\iota^t}[f(\mathbf{X}^t)] - \mathbb{E}_{\iota^{t+1}}[f(\mathbf{X}^{t+1})] + \tfrac{(N-b)}{2b(N-1)}\sigma^2 + \tfrac{1-p}{2p}(\mathbb{E}_{\iota^t}[u^t] - \mathbb{E}_{\iota^{t+1}}[u^{t+1}])
\end{aligned} \tag{38}$$

Summing up the inequality above over $t = 0, 1, \ldots, T$, we have:

$$\begin{aligned}
&(\tfrac{\theta - \overline{\mathbf{X}}^2}{2} - \tfrac{L_f^2\overline{\mathbf{X}}^2(1-p)}{2pb'})\mathbb{E}_{\iota^T}[\textstyle\sum_{t=0}^{T}\sum_{i=1}^{n/2}\|\bar{\mathbf{V}}_i^t - \mathbf{I}_2\|_{\mathsf{F}}^2] \\
\leq\ & f(\mathbf{X}^0) - \mathbb{E}_{\iota^T}[f(\mathbf{X}^T)] + \tfrac{(T+1)(N-b)}{2b(N-1)}\sigma^2 + \tfrac{1-p}{2p}(u^0 - \mathbb{E}_{\iota^{T+1}}[u^{T+1}]) \\
\leq\ & f(\mathbf{X}^0) - f(\bar{\mathbf{X}}) + \tfrac{(T+1)(N-b)}{2b(N-1)}\sigma^2 + \tfrac{1-p}{2p}(u^0 - \mathbb{E}_{\iota^{T+1}}[u^{T+1}])
\end{aligned} \tag{39}$$

As a result, there exists an index $\bar{t}$ with $0 \leq \bar{t} \leq T$ such that

$$\begin{aligned}
&(\tfrac{\theta - \overline{\mathbf{X}}^2}{2} - \tfrac{L_f^2\overline{\mathbf{X}}^2(1-p)}{2pb'})(T+1)\mathbb{E}_{\iota^{\bar{t}}}[\textstyle\sum_{i=1}^{n/2}\|\bar{\mathbf{V}}_i^{\bar{t}} - \mathbf{I}_2\|_{\mathsf{F}}^2] \\
\leq\ & f(\mathbf{X}^0) - f(\bar{\mathbf{X}}) + \tfrac{(T+1)(N-b)}{2b(N-1)}\sigma^2 + \tfrac{1-p}{2p}(u^0 - \mathbb{E}_{\iota^{T+1}}[u^{T+1}])
\end{aligned} \tag{40}$$

Defining $\varpi = \frac{\theta - \overline{\mathrm{X}}^2}{2} - \frac{L_f^2 \overline{\mathrm{X}}^2 (1-p)}{2pb'}$, furthermore, for any $t$ and $\forall i$, we have:

$$\mathcal{E}(\mathbf{X}^t) = \tfrac{1}{C_J} \sum_{i=1}^{C_J} \mathbb{E}_{\iota^t}[\mathrm{dist}(\mathbf{I}_2, \arg\min_{\mathbf{V}_:} \mathcal{T}(\mathbf{V}_:; \mathbf{X}^t, \tilde{\mathcal{B}}_i))^2] = \mathbb{E}_{\iota^t}[\sum_{i=1}^{n/2} \|\bar{\mathbf{V}}_i^t - \mathbf{I}_2\|_{\mathsf{F}}^2] \qquad (41)$$

Combining inequality (40) and (41) , we have the following result:

$$\mathcal{E}(\mathbf{X}^{\bar{t}}) \leq \tfrac{1}{(T+1)\varpi}(f(\mathbf{X}^0) - f(\bar{\mathbf{X}}) + \tfrac{(T+1)(N-b)}{2b(N-1)}\sigma^2 + \tfrac{1-p}{2p}(u^0 - \mathbb{E}_{\iota^{T+1}}[u^{T+1}])) \qquad (42)$$

By the chosen parameters and Inequality (42), we have

$$\mathcal{E}(\mathbf{X}^{\bar{t}}) \leq \tfrac{1}{(T+1)\varpi}(f(\mathbf{X}^0) - f(\bar{\mathbf{X}}) + \tfrac{(T+1)(N-b)}{2b(N-1)}\sigma^2 + \tfrac{1-p}{2p}(u^0 - \mathbb{E}_{\iota^{T+1}}[u^{T+1}])) \leq \epsilon.$$

We define $\Delta_0 = f(\mathbf{X}_0) - f(\bar{\mathbf{X}})$ and set $T + 1 = \frac{\Delta_0}{\epsilon\varpi}$. Denoting $m_t^i$ to be the number of arithmetic operations to update the $i$-th block at $t$-th iteration, we have for $t \geq 1$

$$\mathbb{E}_{\iota^t}[m_t^i] \leq \mathcal{O}(2(pb + (1-p)b')).$$

Letting $m_t$ be the number of arithmetic operations in the $t$-the iteration, we have for $t \geq 1$

$$\mathbb{E}_{\iota^t}[m_t] = \mathbb{E}_{\iota^t}[\sum_{i=1}^{n/2} m_t^i] \leq \mathcal{O}((pb + (1-p)b')n/2 \times 2) = \mathcal{O}(n(pb + (1-p)b')).$$

Hence, the total number of arithmetic operations $M^T$ in $T$ iterations to obtain $\epsilon$-BS-point is

$$\mathbb{E}_{\iota^T}[M] = \mathbb{E}_{\iota^t}[\sum_{t=0}^T m_t] \leq \mathcal{O}(bn) + \mathbb{E}_{\iota^t}[\sum_{t=1}^T m_t] \leq \mathcal{O}(bn + Tn(pb + (1-p)b')).$$

Since $b = N, b' = \sqrt{b}$ and $p = \frac{b'}{b+b'}$, $\varpi = \frac{\theta - \overline{\mathrm{X}}^2}{2} - \frac{L_f^2 \overline{\mathrm{X}}^2(1-p)}{2pb'} = \frac{1}{2}(\theta - \overline{\mathrm{X}}^2 - L_f^2 \overline{\mathrm{X}}^2)$, we have

$$nT(pb + (1-p)b') = n\frac{\Delta_0}{\epsilon(\theta - \overline{\mathrm{X}}^2 - L_f^2\overline{\mathrm{X}}^2)}\frac{2bb'}{b+b'} \leq \frac{n\Delta_0}{\epsilon(\theta - \overline{\mathrm{X}}^2 - L_f^2\overline{\mathrm{X}}^2)}2b' \leq \mathcal{O}(\frac{\Delta_0\sqrt{N}}{\epsilon}).$$

$\square$

### E.4 Proof of Theorem 4.10

*Proof.* For simplicity, we use $\mathtt{B}$ instead of $\mathtt{B}^t$. We notice that the Riemannian gradient of $\mathcal{T}(\mathbf{V}_:; \mathbf{X}^t, \mathtt{B})$ at the point $\mathbf{V}_i = \mathbf{I}_2, \forall i$ . Defining $\mathbf{G} = \tilde{\mathbf{G}}^t[\mathbf{X}^t]^\top$ and using $\mathbf{J}\mathbf{U}_{\mathtt{B}} = \mathbf{U}_{\mathtt{B}}\mathbf{J}_{\mathtt{BB}}$, $\mathbf{U}_{\mathtt{B}}^\top\mathbf{J} = \mathbf{J}_{\mathtt{BB}}\mathbf{U}_{\mathtt{B}}^\top$, we have:

$$\nabla_{\mathcal{J}}\mathcal{T}(\mathbf{V}_: = \mathbf{I}_2; \mathbf{X}^t, \mathtt{B}) = \sum_{i=1}^{n/2} \mathbf{U}_{\mathtt{B}_{(i)}}^\top \mathbf{G}\mathbf{U}_{\mathtt{B}_{(i)}} - \mathbf{U}_{\mathtt{B}_{(i)}}^\top \mathbf{J}\mathbf{G}^\top\mathbf{J}\mathbf{U}_{\mathtt{B}_{(i)}} \qquad (43)$$

Then, we prove the following important lemmas.

**Lemma E.1.** *We have the following result for* **VR-J-JOBCD***:* $\mathbb{E}_{\iota^{t+1}}[\|\tilde{\mathbf{G}}^t - \tilde{\mathbf{G}}^{t+1}\|_{\mathsf{F}}] \leq p\mathbb{E}_{\iota^t}[\sqrt{u^t}] + L_f\mathbb{E}_{\iota^{t+1}}[\|\mathbf{X}^t - \mathbf{X}^{t+1}\|_{\mathsf{F}}]$

*Proof.* By the definition of $\tilde{\mathbf{G}}^t$, with the choice of $b = N$, $b' = \sqrt{b}$ and $p = \frac{b'}{b+b'}$, we have

$$\mathbb{E}_{\iota^{t+1}}[\|\tilde{\mathbf{G}}^t - \tilde{\mathbf{G}}^{t+1}\|_{\mathsf{F}}]$$

$$\overset{\text{①}}{=} \mathbb{E}_{\iota^{t+1}}[\|\tilde{\mathbf{G}}^t - \tfrac{p}{b}\sum_{i=1}^b \nabla f_i(\mathbf{X}^{t+1}) - \tfrac{1-p}{b'}\sum_{i=1}^{b'}(\nabla f_i(\mathbf{X}^{t+1}) - \nabla f_i(\mathbf{X}^t)) - (1-p)\tilde{\mathbf{G}}^t\|_{\mathsf{F}}]$$

$$= \mathbb{E}_{\iota^{t+1}}[\|p\tilde{\mathbf{G}}^t - \tfrac{p}{b}\sum_{i=1}^b \nabla f_i(\mathbf{X}^{t+1}) - \tfrac{1-p}{b'}\sum_{i=1}^{b'}(\nabla f_i(\mathbf{X}^{t+1}) - \nabla f_i(\mathbf{X}^t))\|_{\mathsf{F}}]$$

$$\overset{\text{②}}{\leq} p\mathbb{E}_{\iota^{t+1}}[\|\tilde{\mathbf{G}}^t - \nabla f(\mathbf{X}^{t+1})\|_{\mathsf{F}}] + \tfrac{1-p}{b'}\mathbb{E}_{\iota^t}[\|\sum_{i=1}^{b'} \nabla f_i(\mathbf{X}^{t+1}) - \nabla f_i(\mathbf{X}^t)\|_{\mathsf{F}}]$$

$$\overset{\text{③}}{\leq} p\mathbb{E}_{\iota^t}[\|\tilde{\mathbf{G}}^t - \nabla f(\mathbf{X}^t)\|_{\mathsf{F}}] + p\mathbb{E}_{\iota^{t+1}}[\|\nabla f(\mathbf{X}^t) - \nabla f(\mathbf{X}^{t+1})\|_{\mathsf{F}}]$$

$$\qquad + \tfrac{1-p}{b'}\mathbb{E}_{\iota^{t+1}}[\|\sum_{i=1}^{b'} \nabla f_i(\mathbf{X}^{t+1}) - \nabla f_i(\mathbf{X}^t)\|_{\mathsf{F}}]$$

$$\overset{\text{④}}{\leq} p\mathbb{E}_{\iota^t}[\sqrt{u^t}] + p\mathbb{E}_{\iota^{t+1}}[\|\nabla f(\mathbf{X}^t) - \nabla f(\mathbf{X}^{t+1})\|_{\mathsf{F}}] + (1-p)\mathbb{E}_{\iota^{t+1}}[\|\nabla f_i(\mathbf{X}^{t+1}) - \nabla f_i(\mathbf{X}^t)\|_{\mathsf{F}}]$$

$$\overset{\text{⑤}}{\leq} p\mathbb{E}_{\iota^t}[\sqrt{u^t}] + L_f\mathbb{E}_{\iota^{t+1}}[\|\mathbf{X}^t - \mathbf{X}^{t+1}\|_{\mathsf{F}}]$$

where step ① uses formula (9); step ② uses norm inequality and $\frac{1}{b}\sum_{i=1}^b \nabla f_i(\mathbf{X}^{t+1}) = \nabla f(\mathbf{X}^{t+1})$ with $b = N$ and norm inequality; step ③ uses triangle inequality that $\|\mathbf{A} - \mathbf{B}\|_{\mathsf{F}} \leq \|\mathbf{A} - \mathbf{C}\|_{\mathsf{F}} + \|\mathbf{C} - \mathbf{B}\|_{\mathsf{F}}$, for any $\mathbf{A}$, $\mathbf{B}$ and $\mathbf{C}$; step ④ the definition of $u^t$; step ⑤ uses Inequality (2) and the results of telescoping it over $i$ from 1 to $N$. $\square$

**Lemma E.2.** *(Riemannian gradient Lower Bound for the Iterates Gap) We define* $\phi \triangleq (3\overline{X} + \overline{VX})\overline{G} + (1 + \overline{V}^2 + \frac{n}{2}(\overline{X}^2 + \overline{V}^2\overline{X}^2))L_f + (1 + \overline{V}^2)\theta.$ *It holds that:* $\mathbb{E}_{\iota^{t+1}}[\text{dist}(\mathbf{0}, \nabla_{\mathcal{J}}\mathcal{T}(\mathbf{I}_2; \mathbf{X}^{t+1}, \mathtt{B}^{t+1}))] \leq \phi \cdot \mathbb{E}_{\iota^t}[\sum_{i=1}^{n/2} \|\bar{\mathbf{V}}_i^t - \mathbf{I}_2\|_{\mathsf{F}}] + \frac{np\sqrt{u^t}}{2}(\overline{X} + \overline{V}^2\overline{X}).$

*Proof.* For notation simplicity, we define:

$$\Omega_{i0} \triangleq \mathbf{U}_{\mathtt{B}_{(i)}}^\top [\tilde{\mathbf{G}}^{t+1}][\mathbf{X}^{t+1}]^\top \mathbf{U}_{\mathtt{B}_{(i)}}, \forall i \tag{44}$$

$$\Omega_{i1} \triangleq \mathbf{U}_{\mathtt{B}_{(i)}}^\top [\tilde{\mathbf{G}}^{t+1}][\mathbf{X}^t]^\top \mathbf{U}_{\mathtt{B}_{(i)}}, \forall i, \tag{45}$$

$$\Omega_{i2} \triangleq \mathbf{U}_{\mathtt{B}_{(i)}}^\top [\tilde{\mathbf{G}}^t - \tilde{\mathbf{G}}^{t+1}][\mathbf{X}^t]^\top \mathbf{U}_{\mathtt{B}_{(i)}}, \forall i. \tag{46}$$

First, using the optimality of $\bar{\mathbf{V}}_i^t, i \in \{1, \cdots, \frac{n}{2}\}$ for the subproblem, we have:

$$\mathbf{0}_{2,2} = \tilde{\mathbf{G}}_i - \mathbf{J}_{\mathtt{B}_{(i)}}\bar{\mathbf{V}}_i^t\tilde{\mathbf{G}}_i^\top\bar{\mathbf{V}}_i^t\mathbf{J}_{\mathtt{B}_{(i)}} \tag{47}$$

$$\text{where } \tilde{\mathbf{G}}_i = \underbrace{\text{mat}((\mathbf{Q} + \theta\mathbf{I}_2)\text{vec}(\bar{\mathbf{V}}_i^t - \mathbf{I}_2))}_{\triangleq \Upsilon_{i1}} + \underbrace{\mathbf{U}_{\mathtt{B}_{(i)}}^\top\tilde{\mathbf{G}}^t(\mathbf{X}^t)^\top\mathbf{U}_{\mathtt{B}_{(i)}}}_{\triangleq \Upsilon_{i2}}. \tag{48}$$

Using the relation that $\tilde{\mathbf{G}}_i = \Upsilon_{i1} + \Upsilon_{i2}$, we obtain the following results from the above equality:

$$\mathbf{0}_{2,2} = (\Upsilon_{i1} + \Upsilon_{i2}) - \mathbf{J}_{\mathtt{B}_{(i)}}\bar{\mathbf{V}}_i^t(\Upsilon_{i1} + \Upsilon_{i2})^\top\bar{\mathbf{V}}_i^t\mathbf{J}_{\mathtt{B}_{(i)}}$$

$$\stackrel{①}{\Rightarrow} \mathbf{0}_{2,2} = \Upsilon_{i1} + \Omega_{i1} + \Omega_{i2} - \mathbf{J}_{\mathtt{B}_{(i)}}\bar{\mathbf{V}}_i^t(\Upsilon_{i1} + \Omega_{i1} + \Omega_{i2})^\top\bar{\mathbf{V}}_i^t\mathbf{J}_{\mathtt{B}_{(i)}}$$

$$\Rightarrow \Omega_{i1} = \mathbf{J}_{\mathtt{B}_{(i)}}\bar{\mathbf{V}}_i^t(\Upsilon_{i1} + \Omega_{i1} + \Omega_{i2})^\top\bar{\mathbf{V}}_i^t\mathbf{J}_{\mathtt{B}_{(i)}} - \Upsilon_{i1} - \Omega_{i2}, \tag{49}$$

where step ① uses $\Upsilon_{i2} = \Omega_{i1} + \Omega_{i2}$. Then we derive the following results:

$$\mathbb{E}_{\iota^{t+1}}[\text{dist}(\mathbf{0}, \nabla_{\mathcal{J}}\mathcal{T}(\mathbf{V}_: = \mathbf{I}_2; \mathbf{X}^{t+1}, \mathtt{B}^{t+1}))] = \mathbb{E}_{\iota^{t+1}}[\|\nabla_{\mathcal{J}}\mathcal{T}(\mathbf{V}_: = \mathbf{I}_2; \mathbf{X}^{t+1}, \mathtt{B}^{t+1})\|_{\mathsf{F}}]$$

$$\stackrel{①}{=} \mathbb{E}_{\iota^{t+1}}[\| \sum_{i=1}^{n/2} \mathbf{U}_{\mathtt{B}_{(i)}^{t+1}}^\top(\tilde{\mathbf{G}}^{t+1}[\mathbf{X}^{t+1}]^\top - \mathbf{J}\mathbf{X}^{t+1}[\tilde{\mathbf{G}}^{t+1}]^\top\mathbf{J})\mathbf{U}_{\mathtt{B}_{(i)}^{t+1}}\|_{\mathsf{F}}]$$

$$\stackrel{②}{=} \mathbb{E}_{\iota^t}[\| \sum_{i=1}^{n/2} \mathbf{U}_{\mathtt{B}_{(i)}}^\top(\tilde{\mathbf{G}}^{t+1}[\mathbf{X}^{t+1}]^\top - \mathbf{J}\mathbf{X}^{t+1}[\tilde{\mathbf{G}}^{t+1}]^\top\mathbf{J})\mathbf{U}_{\mathtt{B}_{(i)}}\|_{\mathsf{F}}]$$

$$\stackrel{③}{=} \mathbb{E}_{\iota^t}[\| \sum_{i=1}^{n/2} \Omega_{i0} - \mathbf{J}_{\mathtt{B}_{(i)}}\Omega_{i0}^\top\mathbf{J}_{\mathtt{B}_{(i)}}\|_{\mathsf{F}}]$$

$$\stackrel{④}{=} \mathbb{E}_{\iota^t}[\| \sum_{i=1}^{n/2} (\Omega_{i0} - \Omega_{i1}) + \Omega_{i1} - (\mathbf{J}_{\mathtt{B}_{(i)}}\Omega_{i0}^\top\mathbf{J}_{\mathtt{B}_{(i)}} - \mathbf{J}_{\mathtt{B}_{(i)}}\Omega_{i1}^\top\mathbf{J}_{\mathtt{B}_{(i)}}) - \mathbf{J}_{\mathtt{B}_{(i)}}\Omega_{i1}^\top\mathbf{J}_{\mathtt{B}_{(i)}}\|_{\mathsf{F}}]$$

$$\stackrel{⑤}{\leq} \mathbb{E}_{\iota^t}[\| \sum_{i=1}^{n/2} \Omega_{i0} - \Omega_{i1}\|_{\mathsf{F}}] + \mathbb{E}_{\iota^{t+1}}[\| \sum_{i=1}^{n/2} \mathbf{J}_{\mathtt{B}_{(i)}}\Omega_{i0}^\top\mathbf{J}_{\mathtt{B}_{(i)}} - \mathbf{J}_{\mathtt{B}_{(i)}}\Omega_{i1}^\top\mathbf{J}_{\mathtt{B}_{(i)}}\|_{\mathsf{F}}]$$

$$+ \mathbb{E}_{\iota^{t+1}}[\| \sum_{i=1}^{n/2} \Omega_{i1} - \mathbf{J}_{\mathtt{B}_{(i)}}\Omega_{i1}^\top\mathbf{J}_{\mathtt{B}_{(i)}}\|_{\mathsf{F}}]$$

$$\stackrel{⑥}{\leq} \mathbb{E}_{\iota^t}[\| \sum_{i=1}^{n/2} \Omega_{i0} - \Omega_{i1}\|_{\mathsf{F}}] + \mathbb{E}_{\iota^{t+1}}[\| \sum_{i=1}^{n/2} \Omega_{i0}^\top - \Omega_{i1}^\top\|_{\mathsf{F}}] + \mathbb{E}_{\iota^{t+1}}[\| \sum_{i=1}^{n/2} \Omega_{i1} - \mathbf{J}_{\mathtt{B}_{(i)}}\Omega_{i1}^\top\mathbf{J}_{\mathtt{B}_{(i)}}\|_{\mathsf{F}}]$$

$$\stackrel{⑦}{\leq} 2\mathbb{E}_{\iota^t}[\| \sum_{i=1}^{n/2} \Omega_{i0} - \Omega_{i1}\|_{\mathsf{F}}] + \mathbb{E}_{\iota^{t+1}}[\| \sum_{i=1}^{n/2} \Omega_{i1} - \mathbf{J}_{\mathtt{B}_{(i)}}\Omega_{i1}^\top\mathbf{J}_{\mathtt{B}_{(i)}}\|_{\mathsf{F}}]$$

$$\stackrel{⑧}{=} 2\mathbb{E}_{\iota^t}[\| \sum_{i=1}^{n/2} \Omega_{i0} - \Omega_{i1}\|_{\mathsf{F}}]$$

$$+ \mathbb{E}_{\iota^t}[\| \sum_{i=1}^{n/2} \mathbf{J}_{\mathtt{B}_{(i)}}\bar{\mathbf{V}}_i^t(\Upsilon_{i1} + \Omega_{i1} + \Omega_{i2})^\top\bar{\mathbf{V}}_i^t\mathbf{J}_{\mathtt{B}_{(i)}} - \Upsilon_{i1} - \Omega_{i2} - \mathbf{J}_{\mathtt{B}_{(i)}}\Omega_{i1}^\top\mathbf{J}_{\mathtt{B}_{(i)}}\|_{\mathsf{F}}]$$

$$\stackrel{⑨}{\leq} 2\mathbb{E}_{\iota^t}[\| \sum_{i=1}^{n/2} \Omega_{i0} - \Omega_{i1}\|_{\mathsf{F}}] + \mathbb{E}_{\iota^t}[\| \sum_{i=1}^{n/2} \mathbf{J}_{\mathtt{B}_{(i)}}\bar{\mathbf{V}}_i^t\Upsilon_{i1}^\top\bar{\mathbf{V}}_i^t\mathbf{J}_{\mathtt{B}_{(i)}} - \Upsilon_{i1}\|_{\mathsf{F}}] +$$

$$\mathbb{E}_{\iota^t}[\| \sum_{i=1}^{n/2} \bar{\mathbf{V}}_i^t\Omega_{i1}^\top\bar{\mathbf{V}}_i^t - \Omega_{i1}^\top\|_{\mathsf{F}}] + \mathbb{E}_{\iota^t}[\| \sum_{i=1}^{n/2} \mathbf{J}_{\mathtt{B}_{(i)}}\bar{\mathbf{V}}_i^t\Omega_{i2}^\top\bar{\mathbf{V}}_i^t\mathbf{J}_{\mathtt{B}_{(i)}} - \Omega_{i2}\|_{\mathsf{F}}] \tag{50}$$

where step ① uses Equality (43) ; step ② uses the fact that both the working set $\mathtt{B}^t$ and $\mathtt{B}^{t+1}$ are selected randomly and uniformly; step ③ uses the definition of $\Omega_{i0}$ in (44); step ④ uses $-\Omega_{i1} + \Omega_{i1} = \mathbf{0}$ and $-\Omega_{i1}^\top + \Omega_{i1}^\top = \mathbf{0}$; step ⑤ uses the norm inequality; step ⑥ uses the norm inequality; step ⑦ uses the norm inequality; step ⑧ uses Equality (49); step ⑨ uses the norm inequality. We now establish individual bounds for each term for Inequality (50).

For the first term $2\mathbb{E}_{\iota^t}[\|\sum_{i=1}^{n/2}\Omega_{i0} - \Omega_{i1}\|_{\mathsf{F}}]$ in (50):

$$
\begin{aligned}
2\mathbb{E}_{\iota^t}[\|\textstyle\sum_{i=1}^{n/2}\Omega_{i0} - \Omega_{i1}\|_{\mathsf{F}}] &= 2\mathbb{E}_{\iota^t}[\|\textstyle\sum_{i=1}^{n/2}\mathbf{U}_{\mathsf{B}_{(i)}}^{\top}[\tilde{\mathbf{G}}^t][\mathbf{X}^t - \mathbf{X}^t]^{\top}\mathbf{U}_{\mathsf{B}_{(i)}}\|_{\mathsf{F}}]\\
&\overset{\text{\textcircled{1}}}{=} 2\mathbb{E}_{\iota^t}[\|\textstyle\sum_{i=1}^{n/2}[\tilde{\mathbf{G}}^t][\mathbf{U}_{\mathsf{B}_{(i)}}(\bar{\mathbf{V}}_i^t - \mathbf{I}_2)\mathbf{U}_{\mathsf{B}_{(i)}}\mathbf{X}^t]^{\top}\|_{\mathsf{F}}]\\
&\overset{\text{\textcircled{2}}}{\leq} 2\overline{\mathrm{X}}\,\overline{\mathrm{G}}\,\mathbb{E}_{\iota^t}[\|\textstyle\sum_{i=1}^{n/2}\bar{\mathbf{V}}_i^t - \mathbf{I}_2\|_{\mathsf{F}}]\\
&\overset{\text{\textcircled{3}}}{\leq} 2\overline{\mathrm{X}}\,\overline{\mathrm{G}}\,\mathbb{E}_{\iota^t}[\textstyle\sum_{i=1}^{n/2}\|\bar{\mathbf{V}}_i^t - \mathbf{I}_2\|_{\mathsf{F}}] \quad (51)
\end{aligned}
$$

where step ① uses $[\mathbf{X}^t - \mathbf{X}^t]_{\mathsf{B}_i\mathsf{B}_i} = \mathbf{U}_{\mathsf{B}_{(i)}}(\bar{\mathbf{V}}_i^t - \mathbf{I}_2)\mathbf{U}_{\mathsf{B}_{(i)}}^{\top}\mathbf{X}^t$; step ② uses the inequality $\|\mathbf{X}\mathbf{Y}\|_{\mathsf{F}} \leq \|\mathbf{X}\|_{\mathsf{F}}\|\mathbf{Y}\|_{\mathsf{F}}$ for all $\mathbf{X}$ and $\mathbf{Y}$ repeatedly and the fact that $\forall t, \|\tilde{\mathbf{G}}^t\|_{\mathsf{F}} \leq \overline{\mathrm{G}}$ and $\forall t, \|\mathbf{X}^t\|_{\mathsf{F}} \leq \overline{\mathrm{X}}$; step ③ uses the norm inequality.

For the second term $\mathbb{E}_{\iota^t}[\|\sum_{i=1}^{n/2}\mathbf{J}_{\mathsf{B}_{(i)}}\bar{\mathbf{V}}_i^t\Upsilon_{i1}^{\top}\bar{\mathbf{V}}_i^t\mathbf{J}_{\mathsf{B}_{(i)}} - \Upsilon_{i1}\|_{\mathsf{F}}]$ in (50):

$$
\begin{aligned}
\mathbb{E}_{\iota^t}&[\|\textstyle\sum_{i=1}^{n/2}\mathbf{J}_{\mathsf{B}_{(i)}}\bar{\mathbf{V}}_i^t\Upsilon_{i1}^{\top}\bar{\mathbf{V}}_i^t\mathbf{J}_{\mathsf{B}_{(i)}} - \Upsilon_{i1}\|_{\mathsf{F}}]\\
&\overset{\text{\textcircled{1}}}{\leq} \mathbb{E}_{\iota^t}[\|\textstyle\sum_{i=1}^{n/2}\bar{\mathbf{V}}_i^t\Upsilon_{i1}^{\top}\bar{\mathbf{V}}_i^t\|_{\mathsf{F}}] + \mathbb{E}_{\iota^t}[\|\textstyle\sum_{i=1}^{n/2}\Upsilon_{i1}\|_{\mathsf{F}}]\\
&\overset{\text{\textcircled{2}}}{\leq} (1+\overline{\mathrm{V}}^2)\mathbb{E}_{\iota^t}[\|\textstyle\sum_{i=1}^{n/2}\Upsilon_{i1}\|_{\mathsf{F}}]\\
&\overset{\text{\textcircled{3}}}{=} (1+\overline{\mathrm{V}}^2)\mathbb{E}_{\iota^t}[\|\textstyle\sum_{i=1}^{n/2}\mathrm{mat}((\mathbf{Q}+\theta\mathbf{I}_2)\,\mathrm{vec}(\bar{\mathbf{V}}_i^t - \mathbf{I}_2))\|_{\mathsf{F}}]\\
&\leq (1+\overline{\mathrm{V}}^2)\|\mathbf{Q}+\theta\mathbf{I}_2\|_{\mathsf{F}}\cdot\mathbb{E}_{\iota^t}[\|\textstyle\sum_{i=1}^{n/2}\bar{\mathbf{V}}_i^t - \mathbf{I}_2\|_{\mathsf{F}}]\\
&\overset{\text{\textcircled{4}}}{\leq} (1+\overline{\mathrm{V}}^2)(L_f+\theta)\cdot\mathbb{E}_{\iota^t}[\textstyle\sum_{i=1}^{n/2}\|\bar{\mathbf{V}}_i^t - \mathbf{I}_2\|_{\mathsf{F}}] \quad (52)
\end{aligned}
$$

where step ① uses the triangle inequality; step ② uses the inequality $\|\mathbf{X}\mathbf{Y}\|_{\mathsf{F}} \leq \|\mathbf{X}\|_{\mathsf{F}}\|\mathbf{Y}\|_{\mathsf{F}}$ for all $\mathbf{X}$ and $\mathbf{Y}$ and $\forall t, \|\mathbf{V}^t\|_{\mathsf{F}} \leq \overline{\mathrm{V}}$; step ③ uses the definition of $\Upsilon_{i1}$; step ④ uses the choice of $\mathbf{Q} \preceq L_f\mathbf{I}$ and the norm inequality.

For the third term $\mathbb{E}_{\iota^t}[\|\sum_{i=1}^{n/2}\bar{\mathbf{V}}_i^t\Omega_{i1}^{\top}\bar{\mathbf{V}}_i^t - \Omega_{i1}^{\top}\|_{\mathsf{F}}]$ in (50), we have:

$$
\begin{aligned}
\mathbb{E}_{\iota^t}&[\|\textstyle\sum_{i=1}^{n/2}\bar{\mathbf{V}}_i^t\Omega_{i1}^{\top}\bar{\mathbf{V}}_i^t - \Omega_{i1}^{\top}\|_{\mathsf{F}}]\\
&\overset{\text{\textcircled{1}}}{=} \mathbb{E}_{\iota^t}[\|\textstyle\sum_{i=1}^{n/2}\bar{\mathbf{V}}_i^t\Omega_{i1}^{\top}(\bar{\mathbf{V}}_i^t - \mathbf{I}_2) + (\bar{\mathbf{V}}_i^t - \mathbf{I}_2)\Omega_{i1}^{\top}\|_{\mathsf{F}}]\\
&\overset{\text{\textcircled{2}}}{\leq} (1+\overline{\mathrm{V}})\mathbb{E}_{\iota^t}[\textstyle\sum_{i=1}^{n/2}\|\Omega_{i1}\|_{\mathsf{F}}\cdot\|\bar{\mathbf{V}}_i^t - \mathbf{I}_2\|_{\mathsf{F}}]\\
&\overset{\text{\textcircled{3}}}{\leq} (\overline{\mathrm{X}}+\overline{\mathrm{V}}\overline{\mathrm{X}})\mathbb{E}_{\iota^t}[\textstyle\sum_{i=1}^{n/2}\|\tilde{\mathbf{G}}^t\|_{\mathsf{F}}\cdot\|\bar{\mathbf{V}}_i^t - \mathbf{I}_2\|_{\mathsf{F}}]\\
&\overset{\text{\textcircled{4}}}{\leq} (\overline{\mathrm{X}}+\overline{\mathrm{V}}\overline{\mathrm{X}})\overline{\mathrm{G}}\,\mathbb{E}_{\iota^t}[\textstyle\sum_{i=1}^{n/2}\|\bar{\mathbf{V}}_i^t - \mathbf{I}_2\|_{\mathsf{F}}] \quad (53)
\end{aligned}
$$

where step ① uses the fact that $-\bar{\mathbf{V}}_i^t\Omega_{i1}^{\top}\mathbf{I}_2 + \bar{\mathbf{V}}_i^t\Omega_{i1}^{\top} = \mathbf{0}$; step ② uses the norm inequality and $\forall t, \|\mathbf{V}^t\|_{\mathsf{F}} \leq \overline{\mathrm{V}}$; step ③ uses the fact that $\|\Omega_{i1}\|_{\mathsf{F}} = \|\mathbf{U}_{\mathsf{B}_{(i)}}^{\top}\tilde{\mathbf{G}}^t[\mathbf{X}^t]^{\top}\mathbf{U}_{\mathsf{B}_{(i)}}\|_{\mathsf{F}} \leq \overline{\mathrm{X}}\|\tilde{\mathbf{G}}^t\|_{\mathsf{F}}, \forall i$ which can be derived using the norm inequality ; step ④ uses the fact that $\forall\mathbf{X}, \|\tilde{\mathbf{G}}^t\|_{\mathsf{F}} \leq \overline{\mathrm{G}}$.

For the fourth term $\mathbb{E}_{\iota^t}[\|\sum_{i=1}^{n/2}\mathbf{J}_{\mathsf{B}_{(i)}}\bar{\mathbf{V}}_i^t\Omega_{i2}^{\top}\bar{\mathbf{V}}_i^t\mathbf{J}_{\mathsf{B}_{(i)}} - \Omega_{i2}\|_{\mathsf{F}}]$ in (50), we have:

$$
\begin{aligned}
\mathbb{E}_{\iota^t}&[\|\textstyle\sum_{i=1}^{n/2}\mathbf{J}_{\mathsf{B}_{(i)}}\bar{\mathbf{V}}_i^t\Omega_{i2}^{\top}\bar{\mathbf{V}}_i^t\mathbf{J}_{\mathsf{B}_{(i)}} - \Omega_{i2}\|_{\mathsf{F}}]\\
&\overset{\text{\textcircled{1}}}{\leq} \mathbb{E}_{\iota^t}[\|\textstyle\sum_{i=1}^{n/2}\bar{\mathbf{V}}_i^t\Omega_{i2}^{\top}\bar{\mathbf{V}}_i^t\|_{\mathsf{F}}] + \mathbb{E}_{\iota^t}[\|\textstyle\sum_{i=1}^{n/2}\Omega_{i2}\|_{\mathsf{F}}]\\
&\overset{\text{\textcircled{2}}}{\leq} (1+\overline{\mathrm{V}}^2)\mathbb{E}_{\iota^t}[\|\textstyle\sum_{i=1}^{n/2}\Omega_{i2}\|_{\mathsf{F}}]\\
&\overset{\text{\textcircled{3}}}{=} (1+\overline{\mathrm{V}}^2)\mathbb{E}_{\iota^t}[\|\textstyle\sum_{i=1}^{n/2}\mathbf{U}_{\mathsf{B}_{(i)}}^{\top}[\tilde{\mathbf{G}}^t - \tilde{\mathbf{G}}^t][\mathbf{X}^t]^{\top}\mathbf{U}_{\mathsf{B}_{(i)}}\|_{\mathsf{F}}]\\
&\overset{\text{\textcircled{4}}}{\leq} \tfrac{n}{2}(\overline{\mathrm{X}}+\overline{\mathrm{V}}^2\overline{\mathrm{X}})\mathbb{E}_{\iota^t}[\|[\tilde{\mathbf{G}}^t - \tilde{\mathbf{G}}^t]\|_{\mathsf{F}}]\\
&\overset{\text{\textcircled{5}}}{\leq} \tfrac{n}{2}(\overline{\mathrm{X}}+\overline{\mathrm{V}}^2\overline{\mathrm{X}})(p\mathbb{E}_{\iota^t}[\sqrt{u^t}] + L_f\mathbb{E}_{\iota^t}[\|\mathbf{X}^t - \mathbf{X}^t\|_{\mathsf{F}}])\\
&\overset{\text{\textcircled{6}}}{\leq} \tfrac{np}{2}(\overline{\mathrm{X}}+\overline{\mathrm{V}}^2\overline{\mathrm{X}})\mathbb{E}_{\iota^t}[\sqrt{u^t}] + \tfrac{nL_f}{2}(\overline{\mathrm{X}}^2+\overline{\mathrm{V}}^2\overline{\mathrm{X}}^2)\mathbb{E}_{\iota^t}[\textstyle\sum_{i=1}^{n/2}\|\bar{\mathbf{V}}_i^t - \mathbf{I}_2\|_{\mathsf{F}}] \quad (54)
\end{aligned}
$$

where step ① uses the triangle inequality; step ② uses the norm inequality and $\forall t, \|\mathbf{V}^t\|_{\mathsf{F}} \leq \overline{\mathrm{V}}$; step ③ uses the definition of $\forall i, \Omega_{i2} = \mathbf{U}_{\mathsf{B}_{(i)}}^{\top}[\tilde{\mathbf{G}}^t - \tilde{\mathbf{G}}^t][\mathbf{X}^t]^{\top}\mathbf{U}_{\mathsf{B}_{(i)}}$ in (46); step ④ uses the norm inequality and $\forall t, \|\mathbf{X}^t\|_{\mathsf{F}} \leq \overline{\mathrm{X}}$; step ⑤ uses Lemma E.1; step ⑥ uses Part ($\boldsymbol{b}$) in Lemma 2.5 and $\forall t, \|\mathbf{X}^t\|_{\mathsf{F}} \leq \overline{\mathrm{X}}$.

In view of( 51), (52), (53), (54), and (50), we have:

$$\mathbb{E}_{\iota^{t+1}}[\|\nabla_{\mathcal{J}}\mathcal{T}(\mathbf{I}_2; \mathbf{X}^{t+1}, \mathsf{B}^{t+1})\|_{\mathsf{F}}]$$
$$\leq \quad \frac{np}{2}(\overline{\mathrm{X}} + \overline{\mathrm{V}}^2\overline{\mathrm{X}})\mathbb{E}_{\iota^t}[\sqrt{u^t}] + (c_1 + c_2 + c3 + c4) \cdot \mathbb{E}_{\iota^t}[\sum_{i=1}^{n/2} \|\bar{\mathbf{V}}_i^t - \mathbf{I}_2\|_{\mathsf{F}}]$$
$$= \quad \frac{np}{2}(\overline{\mathrm{X}} + \overline{\mathrm{V}}^2\overline{\mathrm{X}})\mathbb{E}_{\iota^t}[\sqrt{u^t}] + \phi\mathbb{E}_{\iota^t}[\sum_{i=1}^{n/2} \|\bar{\mathbf{V}}_i^t - \mathbf{I}_2\|_{\mathsf{F}}]$$

where $c_1 = 2\overline{\mathrm{XG}}, c_2 = (1 + \overline{\mathrm{V}}^2)(L_f + \theta), c_3 = (\overline{\mathrm{X}} + \overline{\mathrm{VX}})\overline{\mathrm{G}}$, and $c_4 = \frac{n}{2}(\overline{\mathrm{X}}^2 + \overline{\mathrm{V}}^2\overline{\mathrm{X}}^2)L_f$. $\quad\square$

**Lemma E.3.** *We have the following results:* $\mathrm{dist}(\mathbf{0}, \nabla_{\mathcal{J}}f(\mathbf{X}^t)) \leq \gamma \cdot \|\nabla_{\mathcal{J}}\mathcal{T}(\mathbf{I}_2; \mathbf{X}^t, \mathsf{B})\|_{\mathsf{F}} + 2\overline{\mathrm{X}}^2\sqrt{\mathbb{E}_{\iota^t}[u^t]}$ *with* $\gamma \triangleq \overline{\mathrm{X}}\sqrt{C_n^2}$.

*Proof.* We have the following inequalities:

$$\|\nabla_{\mathcal{J}}f(\mathbf{X}^t)\|_{\mathsf{F}} \quad \overset{①}{=} \quad \|\nabla f(\mathbf{X}^t) - \mathbf{J}\mathbf{X}^t(\nabla f(\mathbf{X}^t))^{\top}\mathbf{X}^t\mathbf{J}\|_{\mathsf{F}}$$
$$\overset{②}{=} \quad \|\nabla f(\mathbf{X}^t)(\mathbf{X}^t)^{\top}\mathbf{J}\mathbf{X}^t\mathbf{J} - \mathbf{J}\mathbf{X}^t(\nabla f(\mathbf{X}^t))^{\top}\mathbf{J}\mathbf{J}\mathbf{X}^t\mathbf{J}\|_{\mathsf{F}}$$
$$\overset{③}{\leq} \quad \|\nabla f(\mathbf{X}^t)(\mathbf{X}^t)^{\top} - \mathbf{J}\mathbf{X}^t(\nabla f(\mathbf{X}^t))^{\top}\mathbf{J}\|_{\mathsf{F}}\|\mathbf{J}\mathbf{X}^t\mathbf{J}\|_{\mathsf{F}}$$
$$\overset{④}{\leq} \quad \overline{\mathrm{X}}\|\nabla f(\mathbf{X}^t)(\mathbf{X}^t)^{\top} - \mathbf{J}\mathbf{X}^t(\nabla f(\mathbf{X}^t))^{\top}\mathbf{J}\|_{\mathsf{F}}$$

where step ① uses the definition of $\nabla_{\mathcal{J}}f(\mathbf{X}^t)$; step ② uses $\mathbf{J}\mathbf{J} = \mathbf{I}$ and $\mathbf{X}^{\top}\mathbf{J}\mathbf{X} = \mathbf{J} \Rightarrow \mathbf{X}^{\top}\mathbf{J}\mathbf{X}\mathbf{J} = \mathbf{J}\mathbf{J} = \mathbf{I}$; step ③ uses the norm inequality and ; step ④ uses $\forall t, \|\mathbf{X}^t\|_{\mathsf{F}} \leq \overline{\mathrm{X}}$.

We Consider $\|\nabla f(\mathbf{X}^t)(\mathbf{X}^t)^{\top} - \mathbf{J}\mathbf{X}^t(\nabla f(\mathbf{X}^t))^{\top}\mathbf{J}\|_{\mathsf{F}}$:

$$\|\nabla f(\mathbf{X}^t)(\mathbf{X}^t)^{\top} - \mathbf{J}\mathbf{X}^t(\nabla f(\mathbf{X}^t))^{\top}\mathbf{J}\|_{\mathsf{F}}$$
$$\overset{①}{\leq} \quad \|\tilde{\mathbf{G}}^t(\mathbf{X}^t)^{\top} - \mathbf{J}\mathbf{X}^t(\tilde{\mathbf{G}}^t)^{\top}\mathbf{J}\|_{\mathsf{F}} + \|(\nabla f(\mathbf{X}^t) - \tilde{\mathbf{G}}^t)(\mathbf{X}^t)^{\top} - \mathbf{J}\mathbf{X}^t(\nabla f(\mathbf{X}^t) - \tilde{\mathbf{G}}^t)^{\top}\mathbf{J}\|_{\mathsf{F}}$$
$$\overset{②}{\leq} \quad \|\tilde{\mathbf{G}}^t(\mathbf{X}^t)^{\top} - \mathbf{J}\mathbf{X}^t(\tilde{\mathbf{G}}^t)^{\top}\mathbf{J}\|_{\mathsf{F}} + \|\nabla f(\mathbf{X}^t) - \tilde{\mathbf{G}}^t\|_{\mathsf{F}} \cdot \|\mathbf{X}^t\|_{\mathsf{F}} + \|\mathbf{X}^t\|_{\mathsf{F}} \cdot \|\nabla f(\mathbf{X}^t) - \tilde{\mathbf{G}}^t\|_{\mathsf{F}}$$
$$\overset{③}{\leq} \quad \|\tilde{\mathbf{G}}^t(\mathbf{X}^t)^{\top} - \mathbf{J}\mathbf{X}^t(\tilde{\mathbf{G}}^t)^{\top}\mathbf{J}\|_{\mathsf{F}} + 2\overline{\mathrm{X}}\sqrt{\mathbb{E}_{\iota^t}[u^t]}$$

where step ① uses $\forall \mathbf{A}, \mathbf{B}, \|\mathbf{A}\|_{\mathsf{F}} - \|\mathbf{B}\|_{\mathsf{F}} \leq \|\mathbf{A} - \mathbf{B}\|_{\mathsf{F}}$; step ② uses the norm inequality; step ③ uses $\forall t, \|\mathbf{X}^t\|_{\mathsf{F}} \leq \overline{\mathrm{X}}$. Thus,

$$\|\nabla_{\mathcal{J}}F(\mathbf{X}^t)\|_{\mathsf{F}} \quad \leq \quad \overline{\mathrm{X}}\|\tilde{\mathbf{G}}^t(\mathbf{X}^t)^{\top} - \mathbf{J}\mathbf{X}^t(\tilde{\mathbf{G}}^t)^{\top}\mathbf{J}\|_{\mathsf{F}} + 2\overline{\mathrm{X}}^2\sqrt{\mathbb{E}_{\iota^t}[u^t]}$$
$$\overset{①}{\leq} \quad \overline{\mathrm{X}}\sqrt{C_n^2} \cdot \|\sum_{i=1}^{n/2}\mathbf{U}_{\mathsf{B}_{(i)}}^{\top}[\tilde{\mathbf{G}}^t(\mathbf{X}^t)^{\top} - \mathbf{J}\mathbf{X}^t(\tilde{\mathbf{G}}^t)^{\top}\mathbf{J}\mathbf{U}_{\mathsf{B}_{(i)}}\|_{\mathsf{F}}] + 2\overline{\mathrm{X}}^2\sqrt{\mathbb{E}_{\iota^t}[u^t]}$$
$$\overset{②}{=} \quad \overline{\mathrm{X}}\sqrt{C_n^2} \cdot \|\nabla_{\mathcal{J}}\mathcal{T}(\mathbf{I}_2; \mathbf{X}^t, \mathsf{B})\|_{\mathsf{F}} + 2\overline{\mathrm{X}}^2\sqrt{\mathbb{E}_{\iota^t}[u^t]}$$

where step ① uses Lemma A.1 with $\mathbf{W} = \tilde{\mathbf{G}}^t(\mathbf{X}^t)^{\top} - \mathbf{J}\mathbf{X}^t(\tilde{\mathbf{G}}^t)^{\top}\mathbf{J}$ and $k = 2$; step ② uses the definition of $\nabla_{\mathcal{J}}\mathcal{T}(\mathbf{I}_2; \mathbf{X}^t, \mathsf{B})$. $\quad\square$

We now present the following useful lemma.

**Lemma E.4.** *We define* $\mathbf{T}_{\mathbf{X}}\mathcal{J} \triangleq \{\mathbf{Y} \in \mathbb{R}^{n \times n} \mid \mathcal{A}_X(\mathbf{Y}) = \mathbf{0}\}$ *and* $\mathcal{A}_{\mathbf{X}}(\mathbf{Y}) \triangleq \mathbf{X}^{\top}\mathbf{J}\mathbf{Y} + \mathbf{Y}^{\top}\mathbf{J}\mathbf{X}$. *For any* $\mathbf{G} \in \mathbb{R}^{n \times n}$ *and* $\mathbf{X}^{\top}\mathbf{J}\mathbf{X} = \mathbf{J}$, *the unique minimizer of the following optimization problem:*

$$\bar{\mathbf{Y}} = \arg\min_{\mathbf{Y} \in \mathbf{T}_{\mathbf{X}M}\mathcal{J}} h(\mathbf{Y}) = \frac{1}{2}\|\mathbf{Y} - \mathbf{G}\|_{\mathsf{F}}^2,$$

*satisify* $h(\bar{\mathbf{Y}}) \leq h(\mathbf{G} - \mathbf{J}\mathbf{X}\mathbf{G}^{\top}\mathbf{X}\mathbf{J})$.

*Proof.* We note that $\bar{\mathbf{Y}} = \arg\min_{\mathbf{Y}\in T_{\mathbf{X}}\mathcal{J}} \frac{1}{2}\|\mathbf{Y}-\mathbf{G}\|_F^2 = \arg\min_{\mathbf{Y}} \frac{1}{2}\|\mathbf{Y}-\mathbf{G}\|_F^2$, s.t. $\mathbf{X}^\top\mathbf{J}\mathbf{Y} + \mathbf{Y}^\top\mathbf{J}\mathbf{X} = \mathbf{0}$. Introducing a multiplier $\mathbf{\Lambda} \in \mathbb{R}^{n\times n}$ for the linear constraints $\mathbf{X}^\top\mathbf{J}\mathbf{Y} + \mathbf{Y}^\top\mathbf{J}\mathbf{X} = \mathbf{0}$, we have following Lagrangian function: $\tilde{\mathcal{L}}(\mathbf{Y};\mathbf{\Lambda}) = \frac{1}{2}\|\mathbf{Y}-\mathbf{G}\|_F^2 + \langle\mathbf{X}^\top\mathbf{J}\mathbf{Y} + \mathbf{Y}^\top\mathbf{J}\mathbf{X}, \mathbf{\Lambda}\rangle$. We naturally derive the following first-order optimality condition: $\mathbf{Y} - \mathbf{G} + \mathbf{J}\mathbf{X}\mathbf{\Lambda} = \mathbf{0}$, $\mathbf{X}^\top\mathbf{J}\mathbf{Y} + \mathbf{Y}^\top\mathbf{J}\mathbf{X} = \mathbf{0}$. Incorporating the term $\mathbf{Y} = \mathbf{G} - \mathbf{J}\mathbf{X}\mathbf{\Lambda}$ into $\mathbf{X}^\top\mathbf{J}\mathbf{Y} + \mathbf{Y}^\top\mathbf{J}\mathbf{X} = \mathbf{0}$, we obtain:

$$\mathbf{X}^\top\mathbf{X}\mathbf{\Lambda} + \mathbf{\Lambda}^\top\mathbf{X}^\top\mathbf{X} = \mathbf{G}^\top\mathbf{J}\mathbf{X} + \mathbf{X}^\top\mathbf{J}\mathbf{G} \tag{55}$$

Any $\mathbf{\Lambda}$ satisfying formula (55) is a feasible point, so we can easily find :

$$\begin{aligned}
&\mathbf{X}^\top\mathbf{X}\mathbf{\Lambda} = \mathbf{X}^\top\mathbf{J}\mathbf{G} \\
\overset{①}{\Rightarrow}\ &\mathbf{X}\mathbf{\Lambda} = \mathbf{J}\mathbf{G} \\
\overset{②}{\Rightarrow}\ &\mathbf{X}^\top\mathbf{J}\mathbf{X}\mathbf{\Lambda} = \mathbf{X}^\top\mathbf{J}\mathbf{J}\mathbf{G} \\
\overset{③}{\Rightarrow}\ &\mathbf{J}\mathbf{\Lambda} = \mathbf{X}^\top\mathbf{G} \\
\overset{④}{\Rightarrow}\ &\mathbf{\Lambda} = \mathbf{J}\mathbf{X}^\top\mathbf{G} \\
\overset{⑤}{\Rightarrow}\ &\mathbf{\Lambda} = \mathbf{G}^\top\mathbf{X}\mathbf{J}
\end{aligned} \tag{56}$$

where step ① uses the fact that any matrix $\mathbf{X}$ satisfying the J-orthogonality constraint has a determinant of 1 or -1, thus $\mathrm{inv}(\mathbf{X})$ exists; step ② multiply both sides of the equation by $\mathbf{X}\mathbf{J}$;step ③ uses $\mathbf{X}^T\mathbf{J}\mathbf{X} = \mathbf{J}$ and $\mathbf{J}\mathbf{J} = \mathbf{I}$; step ④ multiply both sides of the equation by $\mathbf{J}$ and uses $\mathbf{J}\mathbf{J} = \mathbf{I}$; step ⑤ uses the fact that $\mathbf{\Lambda}$ is a symmetric matrix.

Therefore, a feasible solution $\mathbf{Y}$ can be computed as $\mathbf{Y} = \mathbf{G} - \mathbf{J}\mathbf{X}\mathbf{\Lambda} = \mathbf{G} - \mathbf{J}\mathbf{X}\mathbf{G}^\top\mathbf{X}\mathbf{J}$. Since $\bar{\mathbf{Y}}$ is the optimal solution, there must be $h(\bar{\mathbf{Y}}) \leq h(\mathbf{G} - \mathbf{J}\mathbf{X}\mathbf{G}^\top\mathbf{X}\mathbf{J})$. $\square$

We now present the proof of this lemma.

**Lemma E.5.** *For any $\mathbf{X} \in \mathbb{R}^{n\times n}$, it holds that $\mathrm{dist}(\mathbf{0}, \nabla f^\circ(\mathbf{X})) \leq \mathrm{dist}(\mathbf{0}, \nabla_{\mathcal{J}}f(\mathbf{X}))$.*

*Proof.* For the purpose of analysis, we define the nearest J orthogonal matrix to an arbitrary matrix $\mathbf{Y} \in \mathbb{R}^{n\times n}$ is given by $\mathcal{P}_{\mathcal{J}}(\mathbf{X})$. Similarly, we have $\mathcal{P}_{\mathbf{T}_{\mathbf{X}}\mathcal{J}}(\nabla f(\mathbf{X}))$ for projecting gradient $\nabla f(\mathbf{X})$ into space $\mathbf{T}_{\mathbf{X}}\mathcal{J}$.

We recall that the following first-order optimality conditions are equivalent for all $\mathbf{X} \in \mathbb{R}^{n\times n}$ :

$$(\mathbf{0} \in \nabla f^\circ(\mathbf{X})) \Leftrightarrow (\mathbf{0} \in \mathcal{P}_{\mathbf{T}_{\mathbf{X}}\mathcal{J}}(\nabla f(\mathbf{X}))). \tag{57}$$

Therefore, we derive the following results:

$$\begin{aligned}
\mathrm{dist}(\mathbf{0}, \nabla f^\circ(\mathbf{X})) &= \inf_{\mathbf{Y}\in\nabla f^\circ(\mathbf{X})} \|\mathbf{Y}\|_F \tag{58} \\
&= \inf_{\mathbf{Y}\in\mathcal{P}_{(\mathbf{T}_{\mathbf{X}}\mathcal{J})}(\nabla f(\mathbf{X}))} \|\mathbf{Y}\|_F \tag{59}
\end{aligned}$$

We let $\mathbf{G} \in \nabla f(\mathbf{X})$ and obtain the following results from the above equality:

$$\begin{aligned}
\mathrm{dist}(\mathbf{0}, \nabla f^\circ(\mathbf{X})) &\overset{①}{\leq} \|\mathbf{G} - \mathbf{J}\mathbf{X}\mathbf{G}^\top\mathbf{X}\mathbf{J}\|_F, \tag{60} \\
&\overset{②}{=} \|\nabla_{\mathcal{J}}f(\mathbf{X})\|_F \triangleq \mathrm{dist}(\mathbf{0}, \nabla_{\mathcal{J}}f(\mathbf{X})). \tag{61}
\end{aligned}$$

where step ① uses Lemma E.4; step ② uses $\nabla_{\mathcal{J}}f(\mathbf{X}) = \mathbf{G} - \mathbf{J}\mathbf{X}\mathbf{G}^\top\mathbf{X}\mathbf{J}$ with $\mathbf{G} \in \nabla f(\mathbf{X})$. $\square$

First of all, since $f^\circ(\mathbf{X}) \triangleq f(\mathbf{X}) + \mathcal{I}_{\mathcal{J}}(\mathbf{X})$ is a KL function, we have from Proposition 4.8 that:

$$\begin{aligned}
\frac{1}{\varphi'(f^\circ(\mathbf{X}') - f^\circ(\mathbf{X}))} &\leq \mathrm{dist}(0, \nabla f^\circ(\mathbf{X}')) \\
&\overset{①}{=} \|\nabla_{\mathcal{J}}f(\mathbf{X}')\|_F, \tag{62}
\end{aligned}$$

where step ① uses Lemma E.5. Here, $\varphi(\cdot)$ is some certain concave desingularization function. Since $\varphi(\cdot)$ is concave, we have:

$$\forall \Delta \in \mathbb{R}, \Delta^+ \in \mathbb{R}, \varphi(\Delta^+) + (\Delta - \Delta^+)\varphi'(\Delta) \leq \varphi(\Delta). \tag{63}$$

Applying the inequality above with $\Delta = f(\mathbf{X}^t) - f(\bar{\mathbf{X}})$ and $\Delta^+ = f(\mathbf{X}^{t+1}) - f(\bar{\mathbf{X}})$, we have:

$$(f(\mathbf{X}^t) - f(\mathbf{X}^{t+1}))\varphi'(f(\mathbf{X}^t) - f(\bar{\mathbf{X}}))$$
$$\leq \quad \varphi(f(\mathbf{X}^t) - f(\bar{\mathbf{X}})) - \varphi(f(\mathbf{X}^{t+1}) - f(\bar{\mathbf{X}})) \triangleq \mathcal{E}^t. \tag{64}$$

With the sufficient descent condition as shown in Theorem 4.7, we derive the following inequalities:

$$\mathbb{E}_{\iota^t}[\tfrac{\theta}{2} \sum_{i=1}^{n/2} \|\bar{\mathbf{V}}_i^t - \mathbf{I}_2\|_{\mathsf{F}}^2]$$
$$\leq \quad \mathbb{E}_{\iota^t}[f(\mathbf{X}^t) - f(\mathbf{X}^{t+1})] + \tfrac{1}{2}\mathbb{E}_{\iota^t}[\|\mathbf{X}^t\|_{\mathsf{F}}^2]\mathbb{E}_{\iota^t}[\sum_{i=1}^{n/2} \|\bar{\mathbf{V}}_i^t - \mathbf{I}_2\|_{\mathsf{F}}^2] + \tfrac{1}{2}\mathbb{E}_{\iota^t}[u^t] \tag{65}$$
$$\overset{①}{\Rightarrow} \quad \mathbb{E}_{\iota^t}[\tfrac{\theta - \overline{\mathrm{X}}^2}{2} \sum_{i=1}^{n/2} \|\bar{\mathbf{V}}_i^t - \mathbf{I}_2\|_{\mathsf{F}}^2] \leq \mathbb{E}_{\iota^t}[f(\mathbf{X}^t) - f(\mathbf{X}^{t+1})] + \tfrac{1}{2}\mathbb{E}_{\iota^t}[u^t] \tag{66}$$
$$\tag{67}$$

where step ① uses $\forall t, \|\mathbf{X}^t\|_{\mathsf{F}} \leq \overline{\mathrm{X}}$.

$$\mathbb{E}_{\iota^t}[\tfrac{\theta - \overline{\mathrm{X}}^2}{2} \sum_{i=1}^{n/2} \|\bar{\mathbf{V}}_i^t - \mathbf{I}_2\|_{\mathsf{F}}^2]$$
$$\overset{①}{\leq} \quad \mathbb{E}_{\iota^t}[\tfrac{\mathcal{E}^t}{\varphi'(f(\mathbf{X}^t) - f(\bar{\mathbf{X}}))}] + \tfrac{1}{2}\mathbb{E}_{\iota^t}[u^t]$$
$$\overset{②}{\leq} \quad \mathbb{E}_{\iota^t}[\mathcal{E}^t \|\nabla_{\mathcal{J}} f(\mathbf{X}^t)\|_{\mathsf{F}}] + \tfrac{1}{2}\mathbb{E}_{\iota^t}[u^t]$$
$$\overset{③}{\leq} \quad \mathbb{E}_{\iota^t}[\mathcal{E}^t \gamma \|\nabla_{\mathcal{J}} \mathcal{T}(\mathbf{I}_2; \mathbf{X}^t, \mathsf{B})\|_{\mathsf{F}} + 2\mathcal{E}^t \overline{\mathrm{X}}^2 \sqrt{\mathbb{E}_{\iota^t}[u^t]}] + \tfrac{1}{2}\mathbb{E}_{\iota^t}[u^t]$$
$$\overset{④}{\leq} \quad \mathbb{E}_{\iota^t}[\mathcal{E}^t \gamma \phi \sum_{i=1}^{n/2} \|\bar{\mathbf{V}}_i^{t-1} - \mathbf{I}_2\|_{\mathsf{F}} + \mathcal{E}^t \gamma \tfrac{np}{2}(\overline{\mathrm{X}} + \overline{\mathrm{V}}^2 \overline{\mathrm{X}})\sqrt{\mathbb{E}_{\iota^t}[u^t]}]$$
$$\qquad + 2\mathcal{E}^t \overline{\mathrm{X}}^2 \sqrt{\mathbb{E}_{\iota^t}[u^t]} + \tfrac{1}{2}\mathbb{E}_{\iota^t}[u^t]$$
$$\overset{⑤}{\leq} \quad \mathbb{E}_{\iota^t}[\mathcal{E}^t \gamma \phi \sqrt{\tfrac{n}{2}} \sqrt{\sum_{i=1}^{n/2} \|\bar{\mathbf{V}}_i^{t-1} - \mathbf{I}_2\|_{\mathsf{F}}^2}$$
$$\qquad + \mathcal{E}^t (2\overline{\mathrm{X}}^2 + \gamma \tfrac{np}{2}\overline{\mathrm{X}} + \gamma \tfrac{np}{2}\overline{\mathrm{V}}^2 \overline{\mathrm{X}})\sqrt{\mathbb{E}_{\iota^t}[u^t]}] + \tfrac{1}{2}\mathbb{E}_{\iota^t}[u^t]$$
$$\overset{⑥}{\leq} \quad \mathbb{E}_{\iota^t}[\tfrac{n\mathcal{E}^{t^2}\gamma^2 \phi^2}{4\theta'} + \tfrac{\theta'}{2} \sum_{i=1}^{n/2} \|\bar{\mathbf{V}}_i^{t-1} - \mathbf{I}_2\|_{\mathsf{F}}^2 + \tfrac{\bar{\theta}\mathbb{E}_{\iota^t}[u^t]}{2}$$
$$\qquad + \tfrac{\mathcal{E}^{t^2}(2\overline{\mathrm{X}}^2 + \gamma \frac{np}{2}\overline{\mathrm{X}} + \gamma \frac{np}{2}\overline{\mathrm{V}}^2 \overline{\mathrm{X}})^2}{2\bar{\theta}}] + \tfrac{1}{2}\mathbb{E}_{\iota^t}[u^t]$$
$$\overset{⑦}{=} \quad \mathbb{E}_{\iota^t}[\mathcal{E}^{t^2}\mathfrak{A}^2 + \tfrac{\theta'}{2} \sum_{i=1}^{n/2} \|\bar{\mathbf{V}}_i^{t-1} - \mathbf{I}_2\|_{\mathsf{F}}^2] + \tfrac{\bar{\theta}+1}{2}\mathbb{E}_{\iota^t}[u^t] \tag{68}$$

where step ① uses the sufficient descent condition as shown in Theorem 4.7; step ② uses Inequality (64) and (62) with $\mathbf{X}' = \mathbf{X}^t$ and $\mathbf{X} = \bar{\mathbf{X}}$; step ③ uses lemma E.3 ; step ④ uses Lemma E.2 ; step ⑤ uses $\forall x_i \in \mathbb{R}, \frac{x_1 + \cdots + x_n}{n} \leqslant \sqrt{\frac{x_1^2 + \cdots + x_n^2}{n}}$ ; step ⑥ applies the inequality that $\forall \theta' > 0, a, b, ab \leq \frac{\theta' a^2}{2} + \frac{b^2}{2\theta'}$ with $a = \sqrt{\sum_{i=1}^{n/2} \|\bar{\mathbf{V}}_i^{t-1} - \mathbf{I}_2\|_{\mathsf{F}}^2}, b = \mathcal{E}^t \gamma \phi \sqrt{\frac{n}{2}}; a = \sqrt{\mathbb{E}_{\iota^t}[u^t]}, b = \mathcal{E}^t(2\overline{\mathrm{X}}^2 + \gamma \frac{np}{2}\overline{\mathrm{X}} + \gamma \frac{np}{2}\overline{\mathrm{V}}^2 \overline{\mathrm{X}})$; step ⑦ denote $\mathfrak{A}^2 \triangleq \frac{(2\overline{\mathrm{X}}^2 + \gamma \frac{np}{2}\overline{\mathrm{X}} + \gamma \frac{np}{2}\overline{\mathrm{V}}^2 \overline{\mathrm{X}})^2}{2\bar{\theta}} + \frac{n\gamma^2 \phi^2}{4\theta'}$. To simplify the formula, we define $\aleph^t = \sum_{i=1}^{n/2} \|\bar{\mathbf{V}}_i^t - \mathbf{I}_2\|_{\mathsf{F}}^2$ .

Multiplying both sides by 2 and taking the square root of both sides, we have:

$$\mathbb{E}_{\iota^t}[\sqrt{\theta - \overline{\mathrm{X}}^2} \sqrt{\aleph^t}] \quad \leq \quad \sqrt{\mathbb{E}_{\iota^t}[\mathcal{E}^{t^2}\mathfrak{A}^2 + \theta'\aleph^{t-1}] + (\bar{\theta} + 1)\mathbb{E}_{\iota^t}[u^t]}$$
$$\leq \quad \sqrt{\mathbb{E}_{\iota^t}[\mathcal{E}^{t^2}\mathfrak{A}^2]} + \mathbb{E}_{\iota^{t-1}}[\sqrt{\theta'\aleph^{t-1}}] + \sqrt{(\bar{\theta}+1)\mathbb{E}_{\iota^t}[u^t]}$$
$$\leq \quad \mathcal{E}^t \mathfrak{A} + \sqrt{\theta'}\mathbb{E}_{\iota^{t-1}}[\sqrt{\aleph^{t-1}}] + \sqrt{(\bar{\theta}+1)}\sqrt{\mathbb{E}_{\iota^t}[u^t]} \tag{69}$$

To recursively eliminate term $\sqrt{(\bar{\theta}+1)\mathbb{E}_{\iota^t}[u^t]}$, we take the root of both sides of the Inequality in Lemma 4.5:

$$\sqrt{\mathbb{E}_{\iota^t}[u^t]} \leq \sqrt{\frac{p(N-b)}{b(N-1)}\sigma^2} + \sqrt{(1-p)\mathbb{E}_{\iota^{t-1}}[u^{t-1}]} + \sqrt{\frac{L_f^2\overline{X}^2(1-p)}{b'}\mathbb{E}_{\iota^{t-1}}[\aleph^{t-1}]}$$

$$\leq \sqrt{\frac{p(N-b)}{b(N-1)}\sigma^2} + \sqrt{(1-p)}\sqrt{\mathbb{E}_{\iota^{t-1}}[u^{t-1}]} + \sqrt{\frac{L_f^2\overline{X}^2(1-p)}{b'}}\sqrt{\mathbb{E}_{\iota^{t-1}}[\aleph^{t-1}]} \quad (70)$$

Adding Inequality $\frac{\sqrt{\bar{\theta}+1}}{1-\sqrt{1-p}} \times$ (70) to (69)

$$\mathbb{E}_{\iota^t}[\sqrt{\theta-\overline{X}^2}\sqrt{\aleph^t}] \leq \mathcal{E}^t\mathfrak{A} + (\sqrt{\theta'} + \sqrt{\frac{L_f^2\overline{X}^2(1-p)}{b'}}\frac{\sqrt{\bar{\theta}+1}}{1-\sqrt{1-p}})\mathbb{E}_{\iota^{t-1}}[\sqrt{\aleph^{t-1}}] +$$
$$\frac{\sqrt{1-p}\sqrt{(\bar{\theta}+1)}}{1-\sqrt{1-p}}(\sqrt{\mathbb{E}_{\iota^{t-1}}[u^{t-1}]} - \sqrt{\mathbb{E}_{\iota^t}[u^t]}) + \frac{\sqrt{\bar{\theta}+1}}{1-\sqrt{1-p}}\sqrt{\frac{p(N-b)}{b(N-1)}\sigma^2}(71)$$

With the choice $\sqrt{\theta'} = \frac{\sqrt{\theta-\overline{X}^2}}{2} - \sqrt{\frac{L_f^2\overline{X}^2(1-p)}{b'}}\frac{\sqrt{\bar{\theta}+1}}{1-\sqrt{1-p}}$, we have:

$$\mathbb{E}_{\iota^t}[\sqrt{\theta-\overline{X}^2}\sqrt{\aleph^t}] \leq \mathcal{E}^t\mathfrak{A} + (\frac{\sqrt{\theta-\overline{X}^2}}{2})\mathbb{E}_{\iota^{t-1}}[\sqrt{\aleph^{t-1}}] +$$
$$\frac{\sqrt{1-p}\sqrt{(\bar{\theta}+1)}}{1-\sqrt{1-p}}(\sqrt{\mathbb{E}_{\iota^{t-1}}[u^{t-1}]} - \sqrt{\mathbb{E}_{\iota^t}[u^t]}) + \frac{\sqrt{\bar{\theta}+1}}{1-\sqrt{1-p}}\sqrt{\frac{p(N-b)}{b(N-1)}\sigma^2}(72)$$

Rearranging terms, we have:

$$\mathbb{E}_{\iota^t}[\sqrt{\theta-\overline{X}^2}\sqrt{\aleph^t}] - \mathbb{E}_{\iota^{t-1}}[\frac{\sqrt{\theta-\overline{X}^2}}{2}\sqrt{\aleph^{t-1}}]$$
$$\leq \mathcal{E}^t\mathfrak{A} + \frac{\sqrt{1-p}\sqrt{(\bar{\theta}+1)}}{1-\sqrt{1-p}}(\sqrt{\mathbb{E}_{\iota^{t-1}}[u^{t-1}]} - \sqrt{\mathbb{E}_{\iota^t}[u^t]}) + \frac{\sqrt{\bar{\theta}+1}}{1-\sqrt{1-p}}\sqrt{\frac{p(N-b)}{b(N-1)}\sigma^2} \quad (73)$$

Summing the inequality above over $t = i, 2\ldots, T$, we have:

$$\mathbb{E}_{\iota^T}[\sqrt{\theta-\overline{X}^2}\sqrt{\aleph^T}] + \mathbb{E}_{\iota^{T-1}}[\frac{\sqrt{\theta-\overline{X}^2}}{2}\sum_{t=i}^{T-1}\sqrt{\aleph^t}]$$
$$\leq \mathfrak{A}\sum_{t=i}^T\mathcal{E}^t + \frac{\sqrt{1-p}\sqrt{(\bar{\theta}+1)}}{1-\sqrt{1-p}}(\sqrt{\mathbb{E}_{\iota^{i-1}}[u^{i-1}]} - \sqrt{\mathbb{E}_{\iota^T}[u^T]}) +$$
$$\frac{(T-i+1)\sqrt{\bar{\theta}+1}}{1-\sqrt{1-p}}\sqrt{\frac{p(N-b)}{b(N-1)}\sigma^2} + \frac{\sqrt{\theta-\overline{X}^2}}{2}\mathbb{E}_{\iota^{i-1}}[\sqrt{\aleph^{i-1}}]$$
$$\overset{①}{\leq} \mathfrak{A}\sum_{t=i}^T\mathcal{E}^t + \frac{\sqrt{1-p}\sqrt{(\bar{\theta}+1)}}{1-\sqrt{1-p}}\sqrt{\mathbb{E}_{\iota^{i-1}}[u^{i-1}]} + \frac{(T-i+1)\sqrt{\bar{\theta}+1}}{1-\sqrt{1-p}}\sqrt{\frac{p(N-b)}{b(N-1)}\sigma^2} + \frac{\sqrt{\theta-\overline{X}^2}}{2}\mathbb{E}_{\iota^{i-1}}[\sqrt{\aleph^{i-1}}]$$

where step ① uses the fact that $\mathbb{E}_{\iota^T}[u^T] \geq 0$.

Since $b = N, b' = \sqrt{b}$, $p = \frac{b'}{b+b'}$, we have $\frac{(T+i-1)\sqrt{\bar{\theta}+1}}{1-\sqrt{1-p}}\sqrt{\frac{p(N-b)}{b(N-1)}\sigma^2} = 0$. Rearranging terms, we have:

$$\mathbb{E}_{\iota^T}[\frac{\theta-\overline{X}^2}{2}\sum_{t=i}^T\sqrt{\aleph^t}] \leq \mathfrak{A}\sum_{t=i}^T\mathcal{E}^t + \frac{\sqrt{1-p}\sqrt{(\bar{\theta}+1)}}{1-\sqrt{1-p}}\sqrt{\mathbb{E}_{\iota^{i-1}}[u^{i-1}]} + \frac{\sqrt{\theta-\overline{X}^2}}{2}\mathbb{E}_{\iota^{i-1}}[\sqrt{\aleph^{i-1}}] \quad (74)$$

Considering $\mathfrak{A}\sum_{t=i}^T\mathcal{E}^t$, we have:

$$\mathfrak{A}\sum_{t=i}^T\mathcal{E}^t \overset{①}{=} \mathfrak{A}\sum_{t=i}^T\varphi(f(\mathbf{X}^t) - f(\bar{\mathbf{X}})) - \varphi(f(\mathbf{X}^{t+1}) - f(\bar{\mathbf{X}}))$$
$$\overset{②}{=} \mathfrak{A}[\varphi(f(\mathbf{X}^i) - f(\bar{\mathbf{X}})) - \varphi(f(\mathbf{X}^{T+1}) - f(\bar{\mathbf{X}}))]$$
$$\overset{③}{\leq} \mathfrak{A}\varphi(f(\mathbf{X}^i) - f(\bar{\mathbf{X}})) \quad (75)$$

where step ① uses the definition of $\mathcal{E}^i$ in (64); step ② uses a basic recursive reduction; step ③ uses the fact the desingularization function $\varphi(\cdot)$ is positive. Combining Inequality (74) and (75), we obtain :

$$\mathbb{E}_{\iota^T}[\frac{\theta-\overline{X}^2}{2}\sum_{t=i}^T\sqrt{\aleph^t}] \leq \mathfrak{A}\varphi(f(\mathbf{X}^i) - f(\bar{\mathbf{X}})) +$$
$$\frac{\sqrt{1-p}\sqrt{(\bar{\theta}+1)}}{1-\sqrt{1-p}}\sqrt{\mathbb{E}_{\iota^{i-1}}[u^{i-1}]} + \frac{\sqrt{\theta-\overline{X}^2}}{2}\mathbb{E}_{\iota^i}[\sqrt{\aleph^{i-1}}] \quad (76)$$

Using $\forall t, \|\mathbf{V}^t\|_{\mathsf{F}} \leq \overline{\mathrm{V}}$, we have the fact that $\|\mathbf{V}_i - \mathbf{I}_2\|_{\mathsf{F}}^2 \leq (\|\mathbf{V}_i\|_{\mathsf{F}} + \|\mathbf{I}_2\|_{\mathsf{F}})^2 \leq (\overline{\mathrm{X}} + \sqrt{2})^2$ and $\sum_{i=1}^{n/2} \|\bar{\mathbf{V}}_i^0 - \mathbf{I}_2\|_{\mathsf{F}}^2 \leq \frac{n}{2}(\overline{\mathrm{V}} + \sqrt{2})^2$. Using the inequality that $\frac{\|\mathbf{X}^+ - \mathbf{X}\|_{\mathsf{F}}^2}{\overline{\mathrm{X}}^2} \leq \frac{\|\mathbf{X}^+ - \mathbf{X}\|_{\mathsf{F}}^2}{\|\mathbf{X}\|_{\mathsf{F}}^2} \leq \sum_{i=1}^{n/2} \|\bar{\mathbf{V}}_i - \mathbf{I}_2\|_{\mathsf{F}}^2$ as shown in Part ($b$) in Lemma 2.5 and letting $i = 1$, we have:

$$\mathbb{E}_{\iota^t}\big[\tfrac{\theta - \overline{\mathrm{X}}^2}{2\overline{\mathrm{X}}} \textstyle\sum_{t=1}^T \|\mathbf{X}^{t+1} - \mathbf{X}^t\|_{\mathsf{F}}\big] \quad \leq \quad \mathfrak{A}\varphi(f(\mathbf{X}^1) - f(\bar{\mathbf{X}})) +$$

$$\frac{\sqrt{1-p}\sqrt{(\bar{\theta}+1)}}{1-\sqrt{1-p}}\sqrt{\mathbb{E}_{\iota^0}[u^0]} + \frac{\sqrt{\theta - \overline{\mathrm{X}}^2}}{2}\sqrt{\tfrac{n}{2}(\overline{\mathrm{V}} + \sqrt{2})^2}$$

Since $\mathbb{E}_{\iota^0}[u^0] \leq \frac{N-b}{b(N-1)}\sigma^2 = 0$, we have:

$$\mathbb{E}_{\iota^t}\big[\tfrac{\theta - \overline{\mathrm{X}}^2}{2\overline{\mathrm{X}}} \textstyle\sum_{t=1}^T \|\mathbf{X}^{t+1} - \mathbf{X}^t\|_{\mathsf{F}}\big] \leq \mathfrak{A}\varphi(f(\mathbf{X}^1) - f(\bar{\mathbf{X}})) + \frac{\sqrt{\theta - \overline{\mathrm{X}}^2}}{2}\sqrt{\tfrac{n}{2}(\overline{\mathrm{V}} + \sqrt{2})^2}$$

We can get the expression for C:

$$\mathbb{E}_{\iota^t}\big[\textstyle\sum_{j=1}^t \|\mathbf{X}^{j+1} - \mathbf{X}^j\|_{\mathsf{F}}\big] \leq C$$

where $C \triangleq \frac{2\overline{\mathrm{X}}}{\theta - \overline{\mathrm{X}}^2}(\mathfrak{A}\varphi(f(\mathbf{X}^1) - f(\bar{\mathbf{X}})) + \frac{\sqrt{\theta - \overline{\mathrm{X}}^2}}{2}\sqrt{\tfrac{n}{2}(\overline{\mathrm{V}} + \sqrt{2})^2})$. Considering that: $\sqrt{\theta'} = \frac{\sqrt{\theta - \overline{\mathrm{X}}^2}}{2} - \sqrt{\frac{L_f^2 \overline{\mathrm{X}}^2 (1-p)}{b'}} \frac{\sqrt{\bar{\theta}+1}}{1 - \sqrt{1-p}} = \frac{\sqrt{\theta - \overline{\mathrm{X}}^2}}{2} - \sqrt{L_f^2 \overline{\mathrm{X}}^2 (1 + \bar{\theta})}((1 + N^{\frac{1}{2}})^{\frac{1}{2}} + N^{\frac{1}{4}})$, we have: $\mathfrak{A} = \sqrt{\frac{(2\overline{\mathrm{X}}^2 + \gamma \frac{np}{2}\overline{\mathrm{X}} + \gamma \frac{np}{2}\overline{\mathrm{V}}^2 \overline{\mathrm{X}})^2}{2\bar{\theta}} + \frac{n\gamma^2\phi^2}{4\theta'}} \leq \mathcal{O}(\frac{1}{N^{1/4}})$. Finally, we have $C \leq \mathcal{O}(\frac{\varphi(f(\mathbf{X}^1) - f(\bar{\mathbf{X}}))}{N^{1/4}})$ $\qquad\square$

E.5   PROOF OF THEOREM 4.9

*Proof.* For simplicity, we use $\mathtt{B}$ instead of $\mathtt{B}^t$. Initially, we prove the following important lemmas.

**Lemma E.6.** *(Riemannian gradient Lower Bound for the Iterates Gap) We define* $\phi \triangleq (3\overline{\mathrm{X}} + \overline{\mathrm{VX}})\overline{\mathrm{G}} + (1 + \overline{\mathrm{X}}^2 + \overline{\mathrm{V}}^2 + \overline{\mathrm{V}}^2\overline{\mathrm{X}}^2)L_f + (1 + \overline{\mathrm{V}}^2)\theta$. *It holds that:* $\mathbb{E}_{\xi^{t+1}}[\mathrm{dist}(\mathbf{0}, \nabla_{\mathcal{J}}\mathcal{G}(\mathbf{I}_2; \mathbf{X}^{t+1}, \mathtt{B}^{t+1}))] \leq \phi \cdot \mathbb{E}_{\xi^t}[\|\bar{\mathbf{V}}^t - \mathbf{I}_2\|_{\mathsf{F}}].$

*Proof.* The proof process is exactly the same as in lemma E.2 and will not be repeated here. $\qquad\square$

The following lemma is useful to outline the relation of $\|\nabla_{\mathcal{J}}f(\mathbf{X}^t)\|_{\mathsf{F}}$ and $\|\nabla_{\mathcal{J}}\mathcal{G}(\mathbf{I}_2; \mathbf{X}^t, \mathtt{B})\|_{\mathsf{F}}$.

**Lemma E.7.** *We have the following results:* $\mathrm{dist}(\mathbf{0}, \nabla_{\mathcal{J}}f(\mathbf{X}^t)) \leq \gamma \cdot \mathbb{E}_{\xi^{t-1}}[\mathrm{dist}(\mathbf{0}, \nabla_{\mathcal{J}}\mathcal{G}(\mathbf{I}_2; \mathbf{X}^t, \mathtt{B}))]$ *with* $\gamma \triangleq \overline{\mathrm{X}}\sqrt{C_n^2}$.

*Proof.* We have the following inequalities:

$$
\begin{aligned}
\|\nabla_{\mathcal{J}}f(\mathbf{X}^t)\|_{\mathsf{F}}^2 \quad &\overset{①}{=} \quad \|\mathbf{G}^t - \mathbf{J}\mathbf{X}^t(\mathbf{G}^t)^\top \mathbf{X}^t \mathbf{J}\|_{\mathsf{F}}^2 \\
&\overset{②}{=} \quad \|\mathbf{G}^t(\mathbf{X}^t)^\top \mathbf{J}\mathbf{X}^t \mathbf{J} - \mathbf{J}\mathbf{X}^t(\mathbf{G}^t)^\top \mathbf{J}\mathbf{J}\mathbf{X}^t \mathbf{J}\|_{\mathsf{F}}^2 \\
&\overset{③}{\leq} \quad \|\mathbf{G}^t(\mathbf{X}^t)^\top - \mathbf{J}\mathbf{X}^t(\mathbf{G}^t)^\top \mathbf{J}\|_{\mathsf{F}}^2 \|\mathbf{J}\mathbf{X}^t \mathbf{J}\|_{\mathsf{F}}^2 \\
&\overset{④}{\leq} \quad \|\mathbf{X}^t\|_{\mathsf{F}}^2 \|\mathbf{W}\|_{\mathsf{F}}^2, \text{ with } \mathbf{W} \triangleq \mathbf{G}^t(\mathbf{X}^t)^\top - \mathbf{J}\mathbf{X}^t(\mathbf{G}^t)^\top \mathbf{J} \\
&\overset{⑤}{\leq} \quad \|\mathbf{X}^t\|_{\mathsf{F}}^2 C_n^2 \cdot \mathbb{E}_{\xi^{t-1}}[\|\mathbf{U}_{\mathtt{B}}^\top [\mathbf{G}^t(\mathbf{X}^t)^\top - \mathbf{J}\mathbf{X}^t(\mathbf{G}^t)^\top \mathbf{J}]\mathbf{U}_{\mathtt{B}}\|_{\mathsf{F}}^2] \\
&\overset{⑥}{=} \quad \|\mathbf{X}^t\|_{\mathsf{F}}^2 C_n^2 \cdot \mathbb{E}_{\xi^{t-1}}[\|\nabla_{\mathcal{J}}\mathcal{G}(\mathbf{I}_2; \mathbf{X}^t, \mathtt{B})\|_{\mathsf{F}}^2] \qquad\qquad (77) \\
&\overset{⑦}{\leq} \quad \overline{\mathrm{X}}^2 C_n^2 \cdot \mathbb{E}_{\xi^{t-1}}[\|\nabla_{\mathcal{J}}\mathcal{G}(\mathbf{I}_2; \mathbf{X}^t, \mathtt{B})\|_{\mathsf{F}}^2]
\end{aligned}
$$

where step ① uses the definition of $\nabla_{\mathcal{J}}f(\mathbf{X}^t)$; step ② uses $\mathbf{J}\mathbf{J} = \mathbf{I}$ and $\mathbf{X}^\top \mathbf{J}\mathbf{X} = \mathbf{J} \Rightarrow \mathbf{X}^\top \mathbf{J}\mathbf{X}\mathbf{J} = \mathbf{J}\mathbf{J} = \mathbf{I}$; step ③ uses the norm inequality and ; step ④ uses the definition of $\mathbf{W} \triangleq \mathbf{G}^t(\mathbf{X}^t)^\top - \mathbf{J}\mathbf{X}^t(\mathbf{G}^t)^\top \mathbf{J}$; step ⑤ uses Lemma (A.1) with $k = 2$; step ⑥ uses the definition of $\nabla_{\mathcal{J}}\mathcal{G}(\mathbf{I}_2; \mathbf{X}^t, \mathtt{B})$. Taking the square root of both sides, we finish the proof of this lemma; step ⑦ uses $\forall t, \|\mathbf{X}^t\|_{\mathsf{F}} \leq \overline{\mathrm{X}}$. $\qquad\square$

Finally, we obtain our main convergence results. First of all, since $f^\circ(\mathbf{X}) \triangleq f(\mathbf{X}) + \mathcal{I}_\mathcal{J}(\mathbf{X})$ is a KL function, we have from Proposition 4.8 that:

$$\frac{1}{\varphi'(f^\circ(\mathbf{X}') - f^\circ(\mathbf{X}))} \leq \operatorname{dist}(0, \nabla f^\circ(\mathbf{X}')) \overset{①}{\leq} \|\nabla_\mathcal{J} f(\mathbf{X}')\|_\mathsf{F}, \tag{78}$$

where step ① uses Lemma E.5. Here, $\varphi(\cdot)$ is some certain concave desingularization function. Since $\varphi(\cdot)$ is concave, we have:

$$\forall \Delta \in \mathbb{R}, \Delta^+ \in \mathbb{R}, \varphi(\Delta^+) + (\Delta - \Delta^+)\varphi'(\Delta) \leq \varphi(\Delta).$$

Applying the inequality above with $\Delta = f(\mathbf{X}^t) - f(\bar{\mathbf{X}})$ and $\Delta^+ = f(\mathbf{X}^{t+1}) - f(\bar{\mathbf{X}})$, we have:

$$\begin{aligned}
& (f(\mathbf{X}^t) - f(\mathbf{X}^{t+1}))\varphi'(f(\mathbf{X}^t) - f(\bar{\mathbf{X}})) \\
\leq \quad & \varphi(f(\mathbf{X}^t) - f(\bar{\mathbf{X}})) - \varphi(f(\mathbf{X}^{t+1}) - f(\bar{\mathbf{X}})) \triangleq \mathcal{E}^t.
\end{aligned} \tag{79}$$

We derive the following inequalities:

$$\begin{aligned}
\mathbb{E}_{\xi^t}[\tfrac{\theta}{2}\|\bar{\mathbf{V}}^t - \mathbf{I}_2\|_\mathsf{F}^2] & \overset{①}{\leq} & \mathbb{E}_{\xi^t}[f(\mathbf{X}^t) - f(\mathbf{X}^{t+1})] \\
& \overset{②}{\leq} & \mathbb{E}_{\xi^t}[\frac{\mathcal{E}^t}{\varphi'(f(\mathbf{X}^t) - f(\bar{\mathbf{X}}))}] \\
& \overset{③}{\leq} & \mathbb{E}_{\xi^t}[\mathcal{E}^t\|\nabla_\mathcal{J} f(\mathbf{X}^t)\|_\mathsf{F}] \\
& \overset{④}{\leq} & \mathbb{E}_{\xi^t}[\mathcal{E}^t\gamma\|\nabla_\mathcal{J}\mathcal{G}(\mathbf{I}_2; \mathbf{X}^t, \mathsf{B})\|_\mathsf{F}] \\
& \overset{⑤}{\leq} & \mathbb{E}_{\xi^{t-1}}[\mathcal{E}^t\gamma\phi\|\bar{\mathbf{V}}^{t-1} - \mathbf{I}_2\|_\mathsf{F}] \\
& \overset{⑥}{\leq} & \mathbb{E}_{\xi^{t-1}}[\tfrac{\theta'}{2}\|\bar{\mathbf{V}}^{t-1} - \mathbf{I}_2\|_\mathsf{F}^2 + \frac{(\mathcal{E}^t\gamma\phi)^2}{2\theta'}], \forall \theta' > 0,
\end{aligned}$$

where step ① uses the sufficient descent condition as shown in Theorem 4.6; step ② uses Inequality (79); step ③ uses Inequality (78) with $\mathbf{X}' = \mathbf{X}^t$ and $\mathbf{X} = \bar{\mathbf{X}}$; step ④ uses Lemma E.7; step ⑤ uses Lemma E.6; step ⑥ applies the inequality that $\forall \theta' > 0, a, b, ab \leq \frac{\theta' a^2}{2} + \frac{b^2}{2\theta'}$ with $a = \|\bar{\mathbf{V}}^{t-1} - \mathbf{I}_2\|_\mathsf{F}$ and $b = \mathcal{E}^t\gamma\phi$.

Multiplying both sides by 2 and taking the square root of both sides, we have:

$$\begin{aligned}
\sqrt{\theta}\mathbb{E}_{\xi^t}[\|\bar{\mathbf{V}}^t - \mathbf{I}_2\|_\mathsf{F}] & \leq & \sqrt{\frac{(\mathcal{E}^t\gamma\phi)^2}{\theta'} + \theta'\mathbb{E}_{\xi^{t-1}}[\|\bar{\mathbf{V}}^{t-1} - \mathbf{I}_2\|_\mathsf{F}^2]}, \forall \theta' > 0 \\
& \overset{①}{\leq} & \sqrt{\theta'}\mathbb{E}_{\xi^{t-1}}[\|\bar{\mathbf{V}}^{t-1} - \mathbf{I}_2\|_\mathsf{F}] + \frac{\mathcal{E}^t\gamma\phi}{\sqrt{\theta'}}, \forall \theta' > 0,
\end{aligned}$$

where step ① uses the inequality that $\sqrt{a+b} \leq \sqrt{a} + \sqrt{b}$ for all $a \geq 0$ and $b \geq 0$. Summing the inequality above over $t = i, 2\ldots, T$, we have:

$$\begin{aligned}
& \sqrt{\theta}\mathbb{E}_{\xi^T}[\|\bar{\mathbf{V}}^T - \mathbf{I}_2\|_\mathsf{F}] - \sqrt{\theta'}\mathbb{E}_{\xi^{i-1}}[\|\bar{\mathbf{V}}^{i-1} - \mathbf{I}_2\|_\mathsf{F}] + \sum_{t=i}^{T-1}(\sqrt{\theta} - \sqrt{\theta'})\mathbb{E}_{\xi^t}[\|\bar{\mathbf{V}}^t - \mathbf{I}_2\|_\mathsf{F}] \\
\leq \quad & \frac{\gamma\phi}{\sqrt{\theta'}}\sum_{t=i}^T \mathcal{E}^t \\
\overset{①}{=} \quad & \frac{\gamma\phi}{\sqrt{\theta'}}\sum_{t=i}^T \varphi(f(\mathbf{X}^t) - f(\bar{\mathbf{X}})) - \varphi(f(\mathbf{X}^{t+1}) - f(\bar{\mathbf{X}})) \\
\overset{②}{=} \quad & \frac{\gamma\phi}{\sqrt{\theta'}}[\varphi(f(\mathbf{X}^i) - f(\bar{\mathbf{X}})) - \varphi(f(\mathbf{X}^{T+1}) - f(\bar{\mathbf{X}}))] \\
\overset{③}{\leq} \quad & \frac{\gamma\phi}{\sqrt{\theta'}}\varphi(f(\mathbf{X}^i) - f(\bar{\mathbf{X}})),
\end{aligned}$$

where step ① uses the definition of $\mathcal{E}^i$ in (79); step ② uses a basic recursive reduction; step ③ uses the fact the desingularization function $\varphi(\cdot)$ is positive. With the choice $\theta' = \frac{\theta}{4}$, we have:

$$\begin{aligned}
& \sqrt{\theta}\mathbb{E}_{\xi^T}[\|\bar{\mathbf{V}}^T - \mathbf{I}_2\|_\mathsf{F}] + \frac{\sqrt{\theta}}{2}\sum_{t=i}^{T-1}\mathbb{E}_{\xi^t}[\|\bar{\mathbf{V}}^t - \mathbf{I}_2\|_\mathsf{F}] \\
\leq \quad & \frac{2\gamma\phi}{\sqrt{\theta}}\varphi(f(\mathbf{X}^i) - f(\bar{\mathbf{X}})) + \frac{\sqrt{\theta}}{2}\mathbb{E}_{\xi^{i-1}}[\|\bar{\mathbf{V}}^{i-1} - \mathbf{I}_2\|_\mathsf{F}]
\end{aligned} \tag{80}$$

We obtain from Inequality (80):

$$\frac{1}{2}\sum_{t=i}^{T}\mathbb{E}_{\xi^t}[\|\bar{\mathbf{V}}^t - \mathbf{I}_2\|_{\mathsf{F}}] \leq \frac{2\gamma\phi}{\theta}\varphi(f(\mathbf{X}^i) - f(\bar{\mathbf{X}})) + \frac{1}{2}\mathbb{E}_{\xi^{i-1}}[\|\bar{\mathbf{V}}^{i-1} - \mathbf{I}_2\|_{\mathsf{F}}] \quad (81)$$

$$\overset{①}{\Rightarrow} \quad \frac{1}{2}\sum_{t=i}^{T}\mathbb{E}_{\xi^t}[\|\mathbf{X}^{t+1} - \mathbf{X}^t\|_{\mathsf{F}}] \leq (\frac{2\overline{\mathbf{X}}\gamma\phi}{\theta}\varphi(f(\mathbf{X}^i) - f(\bar{\mathbf{X}})) + \frac{\overline{\mathbf{X}}}{2}(\overline{\mathbf{V}} + \sqrt{2}))$$

where step ① uses $\forall t, \|\mathbf{V}\|_{\mathsf{F}} \leq \overline{\mathbf{V}}$, then $\|\mathbf{V} - \mathbf{I}_2\|_{\mathsf{F}} \leq \|\mathbf{V}\|_{\mathsf{F}} + \|\mathbf{I}\|_{\mathsf{F}} \leq \overline{\mathbf{V}} + \sqrt{2}$ and the inequality that $\frac{\|\mathbf{X}^{i+1} - \mathbf{X}^i\|_{\mathsf{F}}}{\overline{\mathbf{X}}} \leq \|\bar{\mathbf{V}}^i - \mathbf{I}_2\|_{\mathsf{F}}$ as shown in Part (*b*) in Lemma 2.1. Finally, let $i = 1$ we can get:

$$\sum_{t=1}^{T}\mathbb{E}_{\xi^t}[\|\mathbf{X}^{t+1} - \mathbf{X}^t\|_{\mathsf{F}}] \leq C$$

where $C \triangleq \frac{4\overline{\mathbf{X}}\gamma\phi}{\theta}\varphi(f(\mathbf{X}^1) - f(\bar{\mathbf{X}})) + \overline{\mathbf{X}}(\overline{\mathbf{V}} + \sqrt{2}) \leq \mathcal{O}(\varphi(f(\mathbf{X}^1) - f(\bar{\mathbf{X}})))$. $\qquad\square$

# F  Additional Experiment Details and Results

## F.1  Additional Details for Hyperbolic Structural Probe Problem

To begin with, we give the definition of the Ultrahyperbolic manifold $\mathbb{U}_{\alpha}^{p,q}$, which will be used in Ultra-hyperbolic geodesic distance $\mathbf{d}_{\alpha}(\mathbf{x}, \mathbf{y})$ and Diffeomorphism $\varphi(\cdot)$.

▶ **Ultrahyperbolic manifold.**  Vectors in an ultrahyperbolic manifold is defined as $\mathbb{U}_{\alpha}^{p,q} = \{\mathbf{x} = (x_1, x_2, \cdots, x_{p+q})^{\top} \in \mathbb{R}^{p,q} : \|\mathbf{x}\|_q^2 = -\alpha^2\}$[49], where $\alpha$ is a non-negative real number denoting the radius of curvature. $\|\mathbf{x}\|_q^2 = \langle\mathbf{x}, \mathbf{x}\rangle_q$, $\forall\mathbf{x}, \mathbf{y} \in \mathbb{R}^{p,q}, \langle\mathbf{x}, \mathbf{y}\rangle_q = \sum_{i=1}^{p}\mathbf{x}_i\mathbf{y}_i - \sum_{j=p+1}^{p+q}\mathbf{x}_j\mathbf{y}_j$ is a norm of the induced scalar product. The hyperbolic and spherical manifolds can be defined as : $\mathbb{H}_{\alpha} = \mathbb{U}_{\alpha}^{p,1}$, $\mathbb{S}_{\alpha} = \mathbb{U}_{\alpha}^{0,q}$.

▶ **Ultra-hyperbolic geodesic distance.**  The ultra-hyperbolic geodesic distance [27][28] $\mathbf{d}_{\gamma}(\cdot, \cdot)$ is formulated: $\forall\mathbf{x} \in \mathbb{U}_{\alpha}^{p,q}, \mathbf{y} \in \mathbb{U}_{\alpha}^{p,q}$ and $\alpha > 0$, $\mathbf{d}_{\alpha}(\mathbf{x}, \mathbf{y}) = \begin{cases} \alpha\cosh^{-1}(|\frac{\langle\mathbf{x},\mathbf{y}\rangle_q}{\alpha^2}|) & \text{if } |\frac{\langle\mathbf{x},\mathbf{y}\rangle_q}{\alpha^2}| \geq 1 \\ \alpha\cos^{-1}(|\frac{\langle\mathbf{x},\mathbf{y}\rangle_q}{\alpha^2}|) & \text{otherwise.}\end{cases}$

▶ **Diffeomorphism.**  [Theorem 1 Diffeomorphism of [50]]: Any vector $\mathbf{x} \in \mathbb{R}^p \times \mathbb{R}_*^q$ can be mapped into $\mathbb{U}_{\alpha}^{p,q}$ by a double projection $\varphi = \phi^{-1} \circ \phi$, with $\psi(\mathbf{x}) = (\begin{smallmatrix}\mathbf{s} \\ \alpha\frac{\mathbf{t}}{\|\mathbf{t}\|}\end{smallmatrix})$, $\psi^{-1}(\mathbf{z}) = (\begin{smallmatrix}\mathbf{v} \\ \frac{\sqrt{\alpha^2 + \|\mathbf{v}\|^2}}{\alpha}\mathbf{u}\end{smallmatrix})$, where $\mathbf{x} = (\begin{smallmatrix}\mathbf{s} \\ \mathbf{t}\end{smallmatrix}) \in \mathbb{U}_{\alpha}^{p,q}$ with $\mathbf{s} \in \mathbb{R}^p$ and $\mathbf{t} \in \mathbb{R}_*^q \cdot \mathbf{z} = (\begin{smallmatrix}\mathbf{v} \\ \mathbf{u}\end{smallmatrix}) \in \mathbb{R}^p \times \mathbb{S}_{\alpha}^q$ with $\mathbf{v} \in \mathbb{R}^p$ and $\mathbf{u} \in \mathbb{S}_{\alpha}^q$.

## F.2  Additional application: Ultra-hyperbolic Knowledge Graph Embedding

The J orthogonal matrix can be used as an isometric linear operator in the Ultrahyperbolic manifold, [49] et al. extended the knowledge graph model from hyperbolic space to Ultra-hyperbolic space (named as **UltraE**) by this property. The **UltraE** model is formulated as follows:

$$\min_{\mathbf{R},\mathbf{E},\mathbf{b}}\mathcal{L}(\mathbf{R},\mathbf{E},\mathbf{b}) \triangleq -\frac{1}{N}\sum_{(h,r,t)\in\Delta}(\log s(h,r,t) + \sum_{(h',r',r')\in\Delta'_{(h,r,t)}}\log(1 - s(h',r',t')))$$

$$s.t. \begin{cases} s(h,r,t) = \sigma(-d_{\alpha}^2(\mathbf{R}_r\mathbf{E}_h, \mathbf{E}_t) + \mathbf{b}_h + \mathbf{b}_t + \delta) \\ \mathbf{R}_r^{\top}\mathbf{J}\mathbf{R}_r = \mathbf{J}\end{cases}$$

where $\mathbf{E} \in \mathbb{R}^{n_e \times n}$ with $\mathbf{E}_h = \mathbf{E}(h, :) \in \mathbb{U}_{\alpha}^{p,q}$, $\mathbf{b} \in \mathbb{R}^{n_r}$ with $\mathbf{b}_h = \mathbf{b}(r) \in \mathbb{R}$, $\mathbf{R} \in \mathbb{R}^{n_r \times n \times n}$ with $\mathbf{R}_r = \mathbf{R}(r, :, :) \in \mathbb{R}^{n \times n}$ and $\mathbf{J} = [\begin{smallmatrix}\mathbf{I}_p & \mathbf{0} \\ \mathbf{0} & -\mathbf{I}_q\end{smallmatrix}]$; $\Delta \in \mathbb{N}^{N \times 3}$ is the set of positive triplets, $\Delta'_{(h,r,t)} \in \mathbb{N}^{N \times k \times 3}$ denotes the set of negative triples constructed by corrupting $(h, r, t)$; $\delta$ is a global margin hyper-parameter, $\sigma(\cdot)$ is the sigmoid function, $n_e$ represents the number of entities and $n_r$ represents the number of relations; $d_{\alpha}(\cdot)$ stands for the Ultra-hyperbolic geodesic distance (refer to F.1).

▶ **Experiment Details.** We selected a batch of **FB15K** and **WN18RR** respectively as the data set for the Ultra-hyperbolic Knowledge Graph Embedding problem, (training set size, test set size, number of entities, number of relations) are (719,308,135,22) and (545,233,208,5) respectively. $n = 36$, $p = 18$, $\delta = 5$, $\alpha = 1$ and $k = 50$. In order to highlight the difference between J orthogonal optimization, in the **UltraE** model, all entities and biases of the optimization algorithm are optimized using **ADMM** by **Pytorch**, $lr = 5e-4$. We use the **Adagrad** optimizer in Pytorch to optimize the J-orthogonality constraint variable in the **CS** model.

### F.3 Implementation of ADMM algorithm for problem (1)

We consider the following smooth J-orthogonality constraint problem: $\min_{\mathbf{X} \in \mathbb{R}^{n \times n}} f(\mathbf{X})$, s.t.$\mathbf{X}^\top \mathbf{J} \mathbf{X} = \mathbf{J}$. Defining $\mathbf{Y} = \mathbf{J} \mathbf{X} \in \mathbb{R}^{n \times n}$, we have: $\min_{\mathbf{X}, \mathbf{Y} \in \mathbb{R}^{n \times n}} f(\mathbf{X})$ s.t. $\mathbf{X}^\top \mathbf{Y} = \mathbf{J}, \mathbf{Y} = \mathbf{J} \mathbf{X}$. Introducing Lagrange multipliers $\mathbf{Z} \in \mathbb{R}^{n \times n}$, $\mathbf{W} \in \mathbb{R}^{n \times n}$ and $\beta \in \mathbb{R}$, we have the following Lagrange function:

$$\mathcal{L}(\mathbf{X}, \mathbf{Y}; \mathbf{Z}, \mathbf{W}) = f(\mathbf{X}) + \langle \mathbf{X}^\top \mathbf{Y} - \mathbf{J}, \mathbf{Z} \rangle + \langle \mathbf{J} \mathbf{X} - \mathbf{Y}, \mathbf{W} \rangle + \frac{\beta}{2} \|\mathbf{X}^\top \mathbf{Y} - \mathbf{J}\|_{\mathsf{F}}^2 + \frac{\beta}{2} \|\mathbf{J} \mathbf{X} - \mathbf{Y}\|_{\mathsf{F}}^2$$

Supposing $f(\mathbf{X})$ is $l$-Lipschitz gradient continuous: $f(\mathbf{X}) \leqslant f(\mathbf{X}^t) + \langle \mathbf{X} - \mathbf{X}^t, \nabla f(\mathbf{X}^t) \rangle + \frac{l}{2} \|\mathbf{X} - \mathbf{X}^t\|_{\mathsf{F}}^2$, We get the following majorization function of $\mathcal{L}(\mathbf{X}, \mathbf{Y}; \mathbf{Z}, \mathbf{W})$ at $(\mathbf{X}^t, \mathbf{Y}; \mathbf{Z}, \mathbf{W})$:

$$\begin{aligned} \mathcal{L}(\mathbf{X}, \mathbf{Y}; \mathbf{Z}, \mathbf{W}) \quad \leq \quad & f(\mathbf{X}^{\mathbf{t}}) + \langle \mathbf{X} - \mathbf{X}^t, \nabla f(\mathbf{X}^t) \rangle + \frac{l}{2} \|\mathbf{X} - \mathbf{X}^t\|_{\mathsf{F}}^2 + \langle \mathbf{X}^\top \mathbf{Y} - \mathbf{J}, \mathbf{Z} \rangle + \\ & \langle \mathbf{J} \mathbf{X} - \mathbf{Y}, \mathbf{W} \rangle + \frac{\beta}{2} \|\mathbf{X}^\top \mathbf{Y} - \mathbf{J}\|_{\mathsf{F}}^2 + \frac{\beta}{2} \|\mathbf{J} \mathbf{X} - \mathbf{Y}\|_{\mathsf{F}}^2 \end{aligned}$$

We solve the following subproblem to update $\mathbf{X}^{t+1}$ and $\mathbf{Y}^{t+1}$ alternately:

$$\begin{aligned} \mathbf{X}^{t+1} \quad = \quad & \arg\min_{\mathbf{X}} \mathcal{L}_{\mathbf{X}}(\mathbf{X}, \mathbf{Y}^t; \mathbf{Z}^t, \mathbf{W}^t) \triangleq \langle \mathbf{X} - \mathbf{X}^t, \nabla f(\mathbf{X}^t) \rangle + \frac{l}{2} \|\mathbf{X} - \mathbf{X}^t\|_F^2 + \\ & \langle \mathbf{X}^\top \mathbf{Y} - \mathbf{J}, \mathbf{Z}^t \rangle + \langle \mathbf{J} \mathbf{X} - \mathbf{Y}, \mathbf{W}^t \rangle + \frac{\beta}{2} \|\mathbf{X}^\top \mathbf{Y} - \mathbf{J}\|_{\mathsf{F}}^2 + \frac{\beta}{2} \|\mathbf{J} \mathbf{X} - \mathbf{Y}\|_{\mathsf{F}}^2 \\ \mathbf{Y}^{t+1} \quad = \quad & \arg\min_{\mathbf{Y}} \mathcal{L}_{\mathbf{Y}}(\mathbf{X}^{t+1}, \mathbf{Y}; \mathbf{Z}^t, \mathbf{W}^t) \triangleq \langle \mathbf{X}^\top \mathbf{Y} - \mathbf{J}, \mathbf{Z}^t \rangle + \langle \mathbf{J} \mathbf{X}^{t+1} - \mathbf{Y}, \mathbf{W}^t \rangle \\ & + \frac{\beta}{2} \|\mathbf{X}^{t+1\top} \mathbf{Y} - \mathbf{J}\|_F^2 + \frac{\beta}{2} \|\mathbf{J} \mathbf{X}^{t+1} - \mathbf{Y}\|_{\mathsf{F}}^2 \\ \mathbf{Z}^{t+1} \quad = \quad & \mathbf{Z}^t + \beta \cdot (\mathbf{X}^{t+1\top} \mathbf{Y}^{t+1} - \mathbf{J}) \\ \mathbf{W}^{t+1} \quad = \quad & \mathbf{W}^t + \beta \cdot (\mathbf{J} \mathbf{X}^{t+1} - \mathbf{Y}^{t+1}) \end{aligned}$$

Considering first-order optimality conditions for functions $\mathcal{L}_{\mathbf{X}}(\mathbf{X}, \mathbf{Y}^t; \mathbf{Z}^t, \mathbf{W}^t)$ and $\mathcal{L}_{\mathbf{Y}}(\mathbf{X}^{t+1}, \mathbf{Y}; \mathbf{Z}^t, \mathbf{W}^t)$, we can get the updated formula for $\mathbf{X}^{t+1}$ and $\mathbf{Y}^{t+1}$:

$$\begin{aligned} \mathbf{X}^{t+1} \quad = \quad & -(l\mathbf{I} + \beta(\mathbf{Y}^t \mathbf{Y}^{t\top} + \mathbf{I}))^{-1}(\nabla f(\mathbf{X}^{t\top}) - l\mathbf{X}^t + \mathbf{Y}^t \mathbf{X}^{t\top} + \mathbf{J} \mathbf{W}^t - \beta \mathbf{Y}^t \mathbf{J} - \beta \mathbf{J} \mathbf{Y}^t) \\ \mathbf{Y}^{t+1} \quad = \quad & -(\beta(\mathbf{X}^{t+1} \mathbf{X}^{t+1\top} + \mathbf{I}))^{-1}(\mathbf{X}^{t+1} \mathbf{Z}^t - \mathbf{J} \mathbf{W}^t - \beta \mathbf{X}^{t+1} \mathbf{J} - \beta \mathbf{J} \mathbf{X}^{t+1}) \end{aligned}$$

### F.4 Implementation of UMCM algorithm for problem (1)

We consider the following smooth J-Orthogonality constraint problem: $\min_{\mathbf{X} \in \mathbb{R}^{n \times n}} f(\mathbf{X})$, s.t. $\mathbf{X}^\top \mathbf{J} \mathbf{X} = \mathbf{J}$. We consider the Lagrangian function of the above J-Orthogonality constraint problem with $\Lambda \in \mathbb{R}^{n \times n}$:

$$\mathcal{L}(\mathbf{X}, \Lambda) = f(\mathbf{X}) - \tfrac{1}{2} \langle \Lambda, \mathbf{X}^\top \mathbf{J} \mathbf{X} - \mathbf{J} \rangle.$$

Setting the gradient of $\mathcal{L}(\mathbf{X}, \Lambda)$ w.r.t. $\mathbf{X}$ to zero yields:$\nabla f(\mathbf{X}) - \mathbf{J} \mathbf{X} \Lambda = 0$.

Multiplying both sides by $\mathbf{X}^\top$ and using the fact that $\mathbf{X}^\top \mathbf{J} \mathbf{X} = \mathbf{J}$, we have $\mathbf{J} \Lambda = \mathbf{X}^\top \nabla f(\mathbf{X})$. Multiplying both sides by $\mathbf{J}^\top$ and using $\mathbf{J}^\top \mathbf{J} = \mathbf{I}$, we have $\Lambda = \mathbf{J} \mathbf{X}^\top \nabla f(\mathbf{X})$. Thus, we obtain equivalent unconstrained optimization problem:

$$\min_{\mathbf{X} \in \mathbb{R}^{n \times n}} f(\mathbf{X}) - \tfrac{1}{2} \langle \mathbf{J} \mathbf{X}^\top \nabla f(\mathbf{X}), \mathbf{X}^\top \mathbf{J} \mathbf{X} - \mathbf{J} \rangle.$$

**Algorithm 3:** Alternating Direction Method of Multipliers for Problem (1)

1: Input: $\mathbf{X}^0 \in \mathbb{R}^{n \times n}$, $\mathbf{Y}^0 = \mathbf{X}^0$, $\mathbf{Z}^0 \in \mathbb{R}^{n \times n}$, $\mathbf{W}^0 \in \mathbb{R}^{n \times n}$ and positive constants $l$, $\beta$.

**while** *not converged* **do**

  2: Update $\mathbf{X}^{t+1}$ and $\mathbf{Y}^{t+1}$.

$$\mathbf{X}^{t+1} = -(l\mathbf{I} + \beta(\mathbf{Y}^t \mathbf{Y}^{t\top} + \mathbf{I}))^{-1}(\nabla f(\mathbf{X}^{t\top}) - l\mathbf{X}^t + \mathbf{Y}^t \mathbf{X}^{t\top} + \mathbf{J}\mathbf{W}^t - \beta\mathbf{Y}^t\mathbf{J} - \beta\mathbf{J}\mathbf{Y}^t)$$
$$\mathbf{Y}^{t+1} = -(\beta(\mathbf{X}^{t+1}\mathbf{X}^{t+1\top} + \mathbf{I}))^{-1}(\mathbf{X}^{t+1}\mathbf{Z}^t - \mathbf{J}\mathbf{W}^t - \beta\mathbf{X}^{t+1}\mathbf{J} - \beta\mathbf{J}\mathbf{X}^{t+1})$$

  3: Update $\mathbf{Z}^{t+1}$ and $\mathbf{W}^{t+1}$.

$$\mathbf{Z}^{t+1} = \mathbf{Z}^t + \beta \cdot (\mathbf{X}^{t+1\top}\mathbf{Y}^{t+1} - \mathbf{J})$$
$$\mathbf{W}^{t+1} = \mathbf{W}^t + \beta \cdot (\mathbf{J}\mathbf{X}^{t+1} - \mathbf{Y}^{t+1})$$

  4: if $t\%5 == 0$ and $\beta \leq 10^9$: $\beta = 2 * \beta$.
  5: $t = t + 1$

**end**

With the quadratic term $\frac{\beta}{2}\|\mathbf{X}^\top \mathbf{J}\mathbf{X} - \mathbf{J}\|_\mathsf{F}^2$, we get the objective function of UMCM as follows:

$$\min_{\mathbf{X} \in \mathbb{R}^{n \times n}} f(\mathbf{X}) - \tfrac{1}{2}\langle \mathbf{J}\mathbf{X}^\top \nabla f(\mathbf{X}), \mathbf{X}^\top \mathbf{J}\mathbf{X} - \mathbf{J}\rangle + \tfrac{\beta}{2}\|\mathbf{X}^\top \mathbf{J}\mathbf{X} - \mathbf{J}\|_\mathsf{F}^2.$$

Finally, we solve it by gradient-based approach. Exactly, we use the Adagrad optimizer built into PYTORCH in the our paper.

F.5   THE SELECTION OF PARAMETER $\beta$ IN ADMM AND UMCM

In the ADMM algorithm, $\beta$ is an important parameter that balances the constraint adherence and the optimization objective. We offer two methods for choosing $\beta$: one is a fixed $\beta$ that remains constant throughout the entire ADMM iteration process, and the other is a dynamic $\beta$ that increases every specified number of iterations until it reaches an upper limit.

| datasetname | (m-n-p) | fixed $\beta$:1e5 | fixed $\beta$:1e7 | fixed $\beta$:1e9 | dynamic $\beta$:1e3 | dynamic $\beta$:1e5 | dynamic $\beta$:1e7 |
|---|---|---|---|---|---|---|---|
| cifar | (1000-100-50) | -9.40e+09(1.5e+06) | -1.29e+07(9.2e-07) | -6.63e+03(2.9e-14) | -5.23e+06(3.6e-10) | -7.71e+03(2.3e-14) | -6.52e+03(2.9e-14) |
| mnist | (1000-780-390) | -9.26e+11(5.8e+05) | -6.31e+04(5.5e-11) | -3.95e+04(2.5e-14) | -1.36e+07(1.5e+02) | -9.88e+04(1.1e-14) | -3.97e+04(9.7e-15) |
| randn1000 | (1000-1000-500) | -1.09e+06(2.1e-07) | -5.03e+05(1.2e-11) | -5.04e+05(1.1e-14) | -4.18e+06(1.7e-09) | -5.06e+05(7.5e-15) | -5.01e+05(8.7e-15) |

Table 3: Supplementary experiments for **HEVP** (limited to 30s). The data in the cell stand for the convergence values of **ADMM** under different $\beta$ Settings and the value in parentheses represents $\frac{1}{n^2}\sum_{ij}^n |\mathbf{X}^\top \mathbf{J}\mathbf{X} - \mathbf{J}|_{ij}$. The values with an error of less than 1e-13 are colored with red.

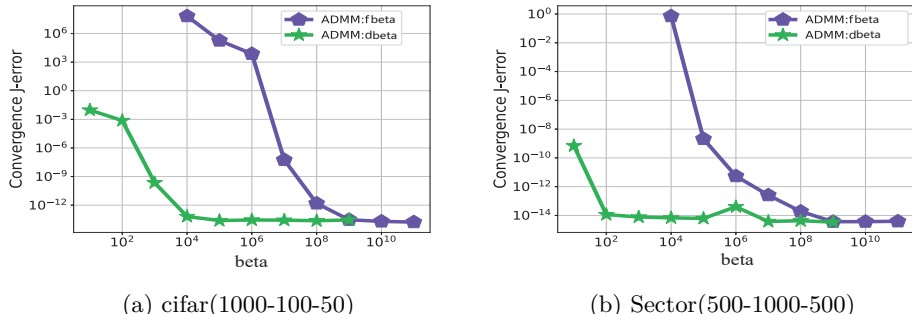

(a) cifar(1000-100-50)          (b) Sector(500-1000-500)

Figure 3: Supplementary experiments for **HEVP** (limited to 30s). The X-axis stands for the initial value of $\beta$, and the Y-axis stands for the convergence error of **ADMM**: $\frac{1}{n^2}\sum_{ij}^n |\mathbf{X}^\top \mathbf{J}\mathbf{X} - \mathbf{J}|_{ij}$.

Table 3 and Figure 3 show the objective function values and constraint violation for solving the HEVP problem with different $\beta$ values when ADMM converges. Generally, the dynamic $\beta$ setting performs better than the fixed $\beta$ setting. However, in the dynamic $\beta$ initial setting, a smaller $\beta$ can lead to larger constraint violation.

To effectively compare with feasible methods such as CSDM and JOBCD, we chose the ADMM algorithm with a dynamic $\beta$ setting, initializing $\beta$ from range $[1e2, 1e5]$. The selection of parameter $\beta$ in the UMCM algorithm shares similar characteristics with that in ADMM. We also use a dynamic setting, with the initial value of $\beta$ chosen from range $[1e2, 1e5]$.

## F.6 EXPERIMENT RESULT

▶ **Hyperbolic Eigenvalue Problem.** Table 4 and Figure 4, 5, 6 are supplementary experiments for HEVP. Several conclusions can be drawn. (i) **GS-JOBCD** often greatly improves upon **UMCM**, **ADMM** and **CSDM**. This is because our methods find stronger stationary points than them. (ii) **J-JOBCD** is a parallel version of **GS-JOBCD** and thus exhibits significantly faster convergence. (iii) The proposed methods generally give the best performance.

▶ **Hyperbolic Structural Probe Problem.** Table 5 and Figure 7, 8 are supplementary experiments for HSPP. Several conclusions can be drawn. (i) **J-JOBCD** often greatly improves upon **UMCM**, **ADMM** and **CSDM** (ii) **VR-J-JOBCD** is a reduced variance version of **J-JOBCD** and thus exhibits significantly faster convergence for problems with large samples. (iii) The proposed methods generally give the best performance.

▶ **Ultra-hyperbolic Knowledge Graph Embedding Problem.** Figure 9, 10, 11 and 12 are supplementary experiments for **UltraE**. Several conclusions can be drawn. (i) In terms of Epoch performance, **J-JOBCD** and **VR-J-JOBCD** often greatly improves upon **CSDM**, thus they show better MRR and hits results. (ii) In models with limited sample sizes, the computational efficiency of **VR-J-JOBCD** is inferior to that of **J-JOBCD**. This discrepancy arises because each iteration in **VR-J-JOBCD** necessitates two instances of backpropagation, thus consuming substantial computational resources. (iii) The proposed methods generally give the best performance.

### F.6.1 Hyperbolic Eigenvalue Problem

Table 4: The convergence curve of the compared methods for solving HEVP. (+) indicates that after the convergence of the **CSDM**, **UMCM** and **ADMM**, utilizing the **GS-JOBCD** for optimization markedly enhances the objective value. The $1^{st}$, $2^{nd}$ , and $3^{rd}$ best results are colored with red, green and blue, respectively. $(n, p)$ represents the dimension and p-value of the $\mathbf{J}$ orthogonal matrix (square matrix). The value in () stands for $\frac{1}{n^2}\sum_{ij}^{n}|\mathbf{X}^{\top}\mathbf{J}\mathbf{X} - \mathbf{J}|_{ij}$ and cells with this value greater than 1e-7 are highlighted in gray.

| dataname | (m-n-p) | UMCM | ADMM | CSDM | GS-JOBCD | J-JOBCD | UMCM+GS-JOBCD | ADMM+GS-JOBCD | CSDM+GS-JOBCD |
|---|---|---|---|---|---|---|---|---|---|
| time limit=30s | | | | | | | | | |
| cifar | (1000-100-50) | -1.11e+04(5.3e-10) | -9.71e+04(1.2e-07) | -1.38e+04(1.8e-09) | -6.33e+03(2.0e-09) | -1.41e+04(1.9e-08) | -2.31e+04(6.2e-09)(+) | -9.71e+04(1.2e-07) | -2.81e+04(6.6e-09)(+) |
| CnnCaltech | (2000-1000-500) | -2.62e+02(3.5e-07) | -2.51e+02(6.9e-07) | -2.61e+02(1.2e-10) | -2.50e+02(9.3e-11) | -1.09e+03(5.3e-09) | -2.65e+02(3.5e-07)(+) | -2.51e+02(6.9e-07) | -2.63e+02(1.3e-10)(+) |
| gisette | (3000-1000-500) | -1.02e+08(2.7e-04) | -2.84e+11(7.2e+01) | -1.56e+06(1.1e-10) | -1.41e+06(1.0e-10) | -2.37e+06(1.7e-09) | -1.03e+08(2.7e-04)(+) | -6.72e+11(7.2e+01)(+) | -1.57e+06(1.4e-10)(+) |
| mnist | (1000-780-390) | -2.13e+05(5.1e-09) | -2.06e+09(1.3e+01) | -4.51e+04(1.5e-10) | -4.00e+04(1.3e-10) | -1.81e+05(7.2e-09) | -2.14e+05(5.2e-09)(+) | -1.81e+09(1.3e+01)(+) | -5.25e+04(2.2e-10)(+) |
| randn10 | (10-10-5) | -4.23e+01(5.8e-09) | -4.21e+01(1.1e-07) | -2.29e+02(2.8e-01) | -4.23e+01(1.5e-08) | -7.85e+01(3.7e-07) | -4.24e+01(1.2e-08)(+) | -4.22e+01(1.2e-07)(+) | -1.06e+11(7.0e+01)(+) |
| randn100 | (100-100-50) | -5.01e+03(1.1e-09) | -4.93e+03(1.1e-07) | -7.29e+03(2.0e-09) | -4.89e+03(2.0e-09) | -1.27e+04(5.5e-08) | -5.01e+03(1.6e-09) | -4.93e+03(1.1e-07) | -8.04e+03(9.7e-09)(+) |
| randn1000 | (1000-1000-500) | -5.64e+05(5.7e-07) | -5.63e+05(4.0e-06) | -5.15e+05(1.2e-10) | -5.01e+05(9.0e-11) | -1.42e+06(1.6e-08) | -5.64e+05(5.7e-07) | -5.63e+05(4.0e-06) | -5.16e+05(1.4e-10)(+) |
| sector | (500-1000-500) | -1.42e+03(2.6e-07) | -1.76e+03(9.4e-07) | -1.67e+03(1.2e-10) | -1.54e+03(1.0e-10) | -1.53e+03(5.8e-10) | -1.66e+03(2.6e-07)(+) | -1.76e+03(9.4e-07) | -1.68e+03(1.3e-10)(+) |
| TDT2 | (1000-1000-500) | -1.05e+08(2.6e-05) | -2.31e+10(5.5e+00) | -1.97e+06(1.1e-10) | -1.81e+06(9.9e-11) | -1.80e+06(6.8e-10) | -1.05e+08(2.6e-05) | -1.38e+11(5.5e+00)(+) | -1.98e+06(1.2e-10)(+) |
| w1a | (2470-290-145) | -1.46e+04(2.1e-09) | -1.72e+04(8.8e-08) | -1.50e+04(1.3e-05) | -1.38e+04(8.4e-10) | -1.97e+05(3.0e-08) | -1.48e+04(3.9e-09)(+) | -1.73e+04(8.8e-08)(+) | -1.59e+04(1.3e-05)(+) |
| cifar | (1000-100-70) | -9.88e+03(1.6e-09) | -7.04e+04(1.1e-07) | -1.04e+04(1.2e-09) | -6.30e+03(3.7e-09) | -1.27e+04(1.9e-08) | -2.15e+04(7.6e-09)(+) | -7.04e+04(1.1e-07)(+) | -2.76e+04(8.5e-09)(+) |
| CnnCaltech | (2000-1000-700) | -2.58e+02(5.2e-07) | -2.50e+02(2.1e-07) | -2.57e+02(8.8e-11) | -2.50e+02(7.0e-11) | -1.01e+03(5.0e-09) | -2.61e+02(5.2e-07)(+) | -2.51e+02(2.1e-07)(+) | -2.59e+02(1.1e-10)(+) |
| gisette | (3000-1000-700) | -6.86e+07(1.7e-04) | -1.11e+10(1.8e+01) | -1.49e+06(8.9e-11) | -1.40e+06(7.2e-11) | -2.00e+06(1.6e-09) | -6.90e+07(1.7e-04)(+) | -3.45e+10(1.8e+01)(+) | -1.52e+06(1.2e-10)(+) |
| mnist | (1000-780-500) | -1.70e+05(4.8e-09) | -1.05e+08(8.4e-01) | -4.24e+04(1.2e-10) | -3.98e+04(1.1e-10) | -1.66e+05(6.7e-09) | -1.71e+05(5.0e-09)(+) | -7.70e+07(8.4e-01) | -4.98e+04(2.0e-10)(+) |
| randn10 | (10-10-7) | -4.24e+01(8.2e-09) | -4.21e+01(1.6e-07) | -1.66e+02(1.7e-01) | -4.23e+01(2.3e-08) | -9.03e+01(3.7e-07) | -4.24e+01(9.9e-09) | -4.21e+01(1.6e-07) | -2.34e+02(1.7e-01)(+) |
| randn100 | (100-100-70) | -4.99e+03(1.1e-09) | -4.92e+03(9.3e-08) | -6.26e+03(1.3e-09) | -4.89e+03(2.5e-09) | -1.18e+04(5.2e-08) | -4.99e+03(1.6e-09) | -4.92e+03(9.3e-08) | -6.99e+03(9.4e-09)(+) |
| randn1000 | (1000-1000-700) | -5.53e+05(1.7e-07) | -5.52e+05(3.9e-06) | -5.09e+05(9.2e-11) | -5.01e+05(6.6e-11) | -1.31e+06(1.5e-08) | -5.53e+05(1.7e-07) | -5.52e+05(3.9e-06) | -5.10e+05(1.2e-10)(+) |
| sector | (500-1000-700) | -1.64e+03(3.2e-08) | -1.78e+03(6.2e-04) | -1.62e+03(8.3e-11) | -1.53e+03(7.1e-11) | -1.53e+03(5.6e-10) | -1.75e+03(3.3e-08)(+) | -1.77e+03(6.2e-04) | -1.63e+03(1.0e-10)(+) |
| TDT2 | (1000-1000-700) | -6.93e+07(1.4e-04) | -2.74e+12(2.1e+02) | -1.91e+06(8.1e-11) | -1.81e+06(7.2e-11) | -1.80e+06(6.2e-10) | -6.93e+07(1.4e-04) | -2.03e+11(2.1e+02)(+) | -1.93e+06(1.0e-10)(+) |
| w1a | (2470-290-200) | -1.44e+04(1.6e-09) | -1.61e+04(7.1e-08) | -1.45e+04(3.3e-10) | -1.38e+04(5.5e-10) | -2.03e+05(2.9e-08) | -1.46e+04(3.8e-09)(+) | -1.61e+04(7.1e-08) | -1.58e+04(1.3e-09)(+) |
| cifar | (1000-100-90) | -7.78e+03(1.5e-09) | -2.55e+04(8.9e-08) | -7.75e+03(7.4e-10) | -7.51e+03(1.3e-08) | -1.15e+04(3.2e-08) | -1.35e+04(1.4e-08)(+) | -2.55e+04(8.9e-08) | -1.71e+04(1.0e-08)(+) |
| CnnCaltech | (2000-1000-900) | -2.52e+02(3.8e-06) | -2.50e+02(3.5e-07) | -2.52e+02(6.1e-11) | -2.50e+02(4.6e-11) | -7.55e+02(4.3e-09) | -2.56e+02(3.8e-06)(+) | -2.50e+02(3.5e-07) | -2.53e+02(8.0e-11)(+) |
| gisette | (3000-1000-900) | -1.30e+07(2.1e-04) | -2.82e+10(3.6e+01) | -1.43e+06(5.9e-11) | -1.40e+06(3.9e-11) | -1.67e+06(1.3e-09) | -1.36e+07(2.1e-04)(+) | -2.95e+10(3.6e+01)(+) | -1.45e+06(8.5e-11)(+) |
| mnist | (1000-780-650) | -4.97e+04(2.3e-08) | -2.82e+05(7.9e-05) | -3.95e+04(8.7e-11) | -3.98e+04(8.2e-11) | -1.16e+05(4.6e-09) | -5.52e+04(2.3e-08)(+) | -2.81e+05(7.9e-05) | -4.39e+04(1.5e-10)(+) |
| randn10 | (10-10-9) | -4.21e+01(6.9e-09) | -4.21e+01(1.2e-07) | -9.39e+01(1.8e-01) | -1.96e+02(5.2e-07) | -7.26e+01(6.5e-07) | -4.65e+02(6.7e-07)(+) | -4.35e+01(2.9e-07)(+) | -1.65e+02(1.8e-01)(+) |
| randn100 | (100-100-90) | -4.93e+03(1.0e-09) | -4.90e+03(8.3e-08) | -5.37e+03(7.6e-10) | -4.89e+03(6.8e-09) | -8.26e+03(4.2e-08) | -4.93e+03(1.1e-09) | -4.90e+03(8.3e-08) | -5.87e+03(7.4e-09)(+) |
| randn1000 | (1000-1000-900) | -5.22e+05(1.3e-08) | -5.23e+05(2.2e-06) | -5.03e+05(5.6e-11) | -5.01e+05(3.8e-11) | -9.00e+05(7.3e-09) | -5.22e+05(1.3e-08) | -5.23e+05(2.2e-06) | -5.04e+05(7.6e-11)(+) |
| sector | (500-1000-900) | -1.47e+03(2.0e-08) | -1.63e+03(2.0e-07) | -1.56e+03(6.0e-11) | -1.53e+03(3.5e-11) | -1.53e+03(2.8e-10) | -1.67e+03(2.1e-08)(+) | -1.63e+03(2.0e-07) | -1.57e+03(8.1e-11)(+) |
| TDT2 | (1000-1000-900) | -1.29e+07(1.1e-04) | -1.44e+11(4.9e+01) | -1.83e+06(5.7e-11) | -1.80e+06(3.8e-11) | -1.80e+06(3.0e-10) | -1.29e+07(1.1e-04) | -1.50e+11(4.9e+01)(+) | -1.85e+06(8.1e-11)(+) |
| w1a | (2470-290-250) | -1.40e+04(2.2e-09) | -1.53e+04(6.4e-08) | -1.40e+04(2.5e-10) | -1.38e+04(6.9e-10) | -1.01e+05(1.6e-08) | -1.41e+04(4.4e-09)(+) | -1.53e+04(6.4e-08) | -1.52e+04(1.2e-09)(+) |
| time limit=60s | | | | | | | | | |
| cifar | (1000-100-50) | -1.11e+04(6.2e-10) | -9.71e+04(1.3e-07) | -2.07e+04(2.2e-09) | -6.50e+03(3.5e-09) | -1.62e+04(3.4e-08) | -2.96e+04(7.2e-09)(+) | -9.71e+04(1.3e-07) | -4.36e+04(8.4e-09)(+) |
| CnnCaltech | (2000-1000-500) | -2.62e+02(2.1e-10) | -2.50e+02(6.4e-08) | -2.71e+02(1.2e-10) | -2.50e+02(1.1e-10) | -1.56e+03(1.0e-08) | -2.67e+02(8.2e-10)(+) | -2.50e+02(6.4e-08) | -2.76e+02(1.5e-10)(+) |
| gisette | (3000-1000-500) | -1.02e+08(1.1e-04) | -4.09e+08(1.1e+01) | -1.67e+06(1.3e-10) | -1.41e+06(1.4e-10) | -3.02e+06(3.8e-09) | -1.04e+08(1.1e-04)(+) | -3.10e+11(1.1e+01)(+) | -1.70e+06(1.8e-10)(+) |
| mnist | (1000-780-390) | -2.13e+05(1.5e-09) | -6.60e+07(4.2e-01) | -4.91e+04(1.6e-10) | -4.11e+04(1.8e-10) | -3.28e+05(1.5e-08) | -2.14e+05(2.0e-09)(+) | -5.82e+07(4.2e-01)(+) | -6.82e+04(3.5e-10)(+) |
| randn10 | (10-10-5) | -4.23e+01(9.0e-09) | -4.21e+01(1.2e-07) | -2.29e+02(2.7e-01) | -4.23e+01(1.5e-08) | -8.32e+01(4.7e-07) | -4.24e+01(1.2e-08)(+) | -4.22e+01(1.2e-07)(+) | -2.80e+03(2.7e-01)(+) |
| randn100 | (100-100-50) | -5.01e+03(1.2e-09) | -4.93e+03(1.0e-07) | -9.13e+03(2.6e-09) | -4.89e+03(2.4e-09) | -1.51e+04(8.3e-08) | -5.01e+03(1.8e-09) | -4.93e+03(1.0e-07) | -1.12e+04(2.2e-08)(+) |
| randn1000 | (1000-1000-500) | -5.64e+05(7.3e-07) | -5.63e+05(7.1e-08) | -5.24e+05(1.3e-10) | -5.01e+05(9.8e-11) | -2.26e+06(3.8e-08) | -5.64e+05(7.3e-07) | -5.63e+05(7.1e-08) | -5.27e+05(1.8e-10)(+) |
| sector | (500-1000-500) | -1.42e+03(2.1e-08) | -1.82e+03(7.0e-08) | -1.76e+03(1.2e-10) | -1.54e+03(1.3e-10) | -1.53e+03(8.1e-10) | -1.72e+03(2.1e-08)(+) | -1.82e+03(7.0e-08)(+) | -1.77e+03(1.3e-10) |
| TDT2 | (1000-1000-500) | -1.05e+08(4.8e-06) | -4.03e+10(5.4e+00) | -2.06e+06(1.2e-10) | -1.81e+06(1.3e-10) | -1.80e+06(8.2e-10) | -1.05e+08(4.8e-06) | -2.16e+11(5.4e+00)(+) | -2.07e+06(1.3e-10) |
| w1a | (2470-290-145) | -1.46e+04(2.0e-09) | -1.72e+04(8.4e-08) | -1.60e+04(2.2e-05) | -1.38e+04(1.5e-09) | -9.40e+05(1.9e-07) | -1.49e+04(4.6e-09)(+) | -1.80e+04(8.4e-08)(+) | -1.85e+04(2.2e-05)(+) |
| cifar | (1000-100-70) | -9.88e+03(1.6e-09) | -7.04e+04(1.1e-07) | -1.39e+04(1.3e-09) | -1.04e+04(1.2e-08) | -1.48e+04(3.6e-08) | -2.71e+04(8.8e-09)(+) | -7.04e+04(1.1e-07) | -3.89e+04(9.4e-09)(+) |
| CnnCaltech | (2000-1000-700) | -2.58e+02(6.6e-10) | -2.50e+02(5.1e-08) | -2.62e+02(9.6e-11) | -2.50e+02(1.0e-10) | -1.47e+03(1.0e-08) | -2.62e+02(1.7e-09)(+) | -2.50e+02(5.1e-08) | -2.68e+02(1.5e-10)(+) |
| gisette | (3000-1000-700) | -6.85e+07(1.6e-04) | -6.31e+07(5.3e-01) | -1.55e+06(9.3e-11) | -1.41e+06(1.2e-10) | -2.38e+06(3.4e-09) | -6.91e+07(1.6e-04)(+) | -8.34e+08(5.3e-01)(+) | -1.63e+06(1.8e-10)(+) |
| mnist | (1000-780-500) | -1.70e+05(8.6e-10) | -8.43e+05(5.4e-04) | -4.48e+04(1.3e-10) | -4.08e+04(1.8e-10) | -2.89e+05(1.4e-08) | -1.73e+05(1.5e-09)(+) | -8.41e+05(5.4e-04) | -6.46e+04(3.0e-10)(+) |
| randn10 | (10-10-7) | -4.24e+01(6.9e-09) | -4.21e+01(2.0e-07) | -1.19e+02(9.6e-02) | -4.22e+01(1.4e-08) | -1.05e+02(7.7e-07) | -4.24e+01(1.2e-08) | -1.23e+02(2.4e-07)(+) | -1.54e+02(9.6e-02)(+) |
| randn100 | (100-100-70) | -4.99e+03(1.2e-09) | -4.92e+03(9.6e-08) | -7.97e+03(1.8e-09) | -4.89e+03(4.1e-09) | -1.50e+04(9.4e-08) | -4.99e+03(1.9e-09) | -4.92e+03(9.6e-08) | -1.04e+04(2.0e-08)(+) |
| randn1000 | (1000-1000-700) | -5.53e+05(3.9e-08) | -5.52e+05(5.1e-08) | -5.18e+05(9.3e-11) | -5.01e+05(8.9e-11) | -2.11e+06(3.6e-08) | -5.53e+05(3.9e-08) | -5.52e+05(5.1e-08) | -5.21e+05(1.6e-10)(+) |
| sector | (500-1000-700) | -1.64e+03(6.3e-09) | -1.77e+03(5.1e-08) | -1.67e+03(9.1e-11) | -1.54e+03(1.3e-10) | -1.53e+03(8.6e-10) | -1.78e+03(7.0e-09)(+) | -1.77e+03(5.1e-08) | -1.73e+03(1.5e-10)(+) |
| TDT2 | (1000-1000-700) | -6.93e+07(1.4e-04) | -4.06e+07(1.5e-01) | -1.98e+06(9.3e-11) | -1.81e+06(1.3e-10) | -1.80e+06(8.4e-10) | -6.93e+07(1.4e-04) | -2.64e+08(1.5e-01)(+) | -2.04e+06(1.5e-10)(+) |
| w1a | (2470-290-200) | -1.44e+04(1.6e-09) | -1.67e+04(7.2e-08) | -1.55e+04(4.1e-10) | -1.38e+04(1.3e-09) | -2.10e+06(2.7e-07) | -1.47e+04(5.2e-09)(+) | -1.72e+04(7.2e-08)(+) | -1.95e+04(4.5e-09)(+) |
| cifar | (1000-100-90) | -7.78e+03(1.5e-09) | -2.55e+04(8.4e-08) | -9.37e+03(7.3e-10) | -1.22e+04(2.0e-08) | -1.24e+04(5.4e-08) | -1.75e+04(1.6e-08)(+) | -2.55e+04(8.4e-08) | -2.52e+04(1.1e-08)(+) |
| CnnCaltech | (2000-1000-900) | -2.52e+02(4.0e-09) | -2.50e+02(8.7e-04) | -2.54e+02(6.3e-11) | -2.50e+02(1.1e-10) | -9.76e+02(8.2e-09) | -2.58e+02(5.0e-09)(+) | -2.50e+02(8.7e-04)(+) | -2.58e+02(1.6e-10)(+) |
| gisette | (3000-1000-900) | -1.25e+07(8.1e-05) | -3.98e+04(2.0e-03) | -1.45e+06(6.0e-11) | -1.40e+06(1.1e-10) | -2.14e+06(3.2e-09) | -1.33e+07(8.1e-05)(+) | -2.10e+06(2.0e-03)(+) | -1.52e+06(1.8e-10)(+) |
| mnist | (1000-780-650) | -4.97e+04(1.0e-09) | -2.79e+05(4.8e-08) | -4.00e+04(8.5e-11) | -4.20e+04(2.1e-10) | -1.88e+05(1.0e-08) | -6.14e+04(2.0e-09)(+) | -2.79e+05(4.8e-08) | -5.68e+04(2.9e-10)(+) |
| randn10 | (10-10-9) | -4.22e+01(4.8e-09) | -4.21e+01(1.4e-07) | -6.70e+01(1.1e-01) | -2.68e+02(9.1e-07) | -8.76e+01(6.3e-07) | -1.18e+02(4.9e-07)(+) | -4.21e+01(1.6e-07) | -9.60e+01(1.1e-01)(+) |
| randn100 | (100-100-90) | -4.93e+03(9.9e-10) | -4.90e+03(8.4e-08) | -5.94e+03(8.5e-10) | -4.89e+03(8.3e-09) | -8.83e+03(7.8e-08) | -4.93e+03(1.3e-09) | -4.90e+03(8.4e-08) | -7.35e+03(1.6e-08)(+) |
| randn1000 | (1000-1000-900) | -5.22e+05(9.5e-10) | -5.23e+05(1.8e-07) | -5.07e+05(6.0e-11) | -5.01e+05(8.2e-11) | -1.34e+06(1.9e-08) | -5.22e+05(1.0e-09) | -5.23e+05(1.8e-07) | -5.08e+05(1.3e-10)(+) |
| sector | (500-1000-900) | -1.47e+03(1.6e-08) | -1.63e+03(4.3e-08) | -1.58e+03(5.9e-11) | -1.53e+03(8.2e-11) | -1.53e+03(4.7e-10) | -1.71e+03(1.7e-08)(+) | -1.63e+03(4.3e-08) | -1.63e+03(1.4e-10)(+) |
| TDT2 | (1000-1000-900) | -1.26e+07(4.2e-05) | -1.44e+10(4.4e+00) | -1.86e+06(6.4e-11) | -1.80e+06(9.1e-11) | -1.80e+06(4.7e-10) | -1.26e+07(4.2e-05) | -1.50e+10(4.4e+00)(+) | -1.92e+06(1.6e-10)(+) |
| w1a | (2470-290-250) | -1.40e+04(2.1e-09) | -1.53e+04(6.3e-08) | -1.45e+04(2.7e-10) | -1.38e+04(2.1e-09) | -4.79e+05(9.4e-08) | -1.42e+04(6.3e-09)(+) | -1.53e+04(6.4e-08) | -1.86e+04(4.2e-09)(+) |
| time limit=90s | | | | | | | | | |
| cifar | (1000-100-50) | -1.11e+04(9.9e-10) | -7.32e+03(1.0e-07) | -1.07e+04(1.4e-09) | -1.53e+04(1.3e-08) | -3.43e+04(6.7e-08) | -5.71e+04(1.1e-08)(+) | -7.62e+04(1.0e-07)(+) | -6.97e+04(1.6e-08)(+) |
| CnnCaltech | (2000-1000-500) | -2.62e+02(1.9e-10) | -2.51e+02(6.5e-08) | -2.80e+02(1.4e-10) | -2.51e+02(1.2e-10) | -1.96e+03(1.6e-08) | -2.68e+02(9.6e-10)(+) | -2.51e+02(6.4e-08) | -2.87e+02(1.8e-10)(+) |
| gisette | (3000-1000-500) | -1.02e+08(5.4e-05) | -1.49e+11(2.5e+00) | -1.77e+06(1.3e-10) | -1.41e+06(1.9e-10) | -3.58e+06(6.0e-09) | -1.05e+08(5.4e-05)(+) | -3.50e+11(2.5e+00)(+) | -1.81e+06(2.4e-10)(+) |
| mnist | (1000-780-390) | -2.13e+05(1.4e-09) | -3.32e+06(1.4e-02) | -5.22e+04(1.6e-10) | -4.41e+04(2.9e-10) | -4.63e+05(2.1e-08) | -2.15e+05(2.0e-09)(+) | -3.16e+06(1.4e-02)(+) | -8.88e+04(5.6e-10)(+) |
| randn10 | (10-10-5) | -4.23e+01(8.4e-09) | -4.21e+01(1.1e-07) | -2.29e+02(2.7e-01) | -4.23e+01(1.5e-08) | -8.46e+01(7.5e-07) | -4.24e+01(1.2e-08)(+) | -4.22e+01(1.1e-07)(+) | -3.01e+02(2.7e-01)(+) |
| randn100 | (100-100-50) | -5.01e+03(1.4e-09) | -4.93e+03(1.1e-07) | -1.34e+04(3.4e-09) | -4.90e+03(2.6e-09) | -1.71e+04(1.2e-07) | -5.01e+03(2.1e-09) | -4.93e+03(1.1e-07) | -2.13e+04(5.8e-08)(+) |
| randn1000 | (1000-1000-500) | -5.64e+05(4.9e-07) | -5.63e+05(6.6e-08) | -5.46e+05(1.4e-10) | -5.01e+05(1.1e-10) | -2.84e+06(6.0e-08) | -5.64e+05(4.9e-07) | -5.63e+05(6.6e-08) | -5.52e+05(2.4e-10)(+) |
| sector | (500-1000-500) | -1.42e+03(7.5e-10) | -1.51e+03(5.2e-08) | -1.63e+03(1.1e-10) | -1.54e+03(1.6e-10) | -2.52e+03(3.4e-09) | -1.76e+03(1.6e-09)(+) | -1.54e+03(5.2e-08)(+) | -1.63e+03(1.4e-10) |
| TDT2 | (1000-1000-500) | -1.05e+08(2.8e-06) | -7.34e+08(1.5e+00) | -1.93e+06(1.1e-10) | -1.81e+06(1.8e-10) | -3.21e+06(3.6e-09) | -1.05e+08(2.8e-06) | -5.87e+11(5.8e+00)(+) | -1.93e+06(1.4e-10) |
| w1a | (2470-290-145) | -1.46e+04(2.0e-09) | -1.39e+04(7.8e-08) | -1.45e+04(1.1e-05) | -1.38e+04(3.1e-09) | -3.60e+06(5.3e-07) | -1.52e+04(6.8e-09)(+) | -1.39e+04(7.8e-08) | -1.63e+04(1.1e-05)(+) |
| cifar | (1000-100-70) | -9.88e+03(1.8e-09) | -7.17e+03(9.1e-08) | -8.80e+03(1.1e-09) | -4.18e+04(2.2e-08) | -3.04e+04(6.8e-08) | -5.84e+04(1.4e-08)(+) | -5.02e+04(9.3e-08)(+) | -5.70e+04(1.7e-08)(+) |
| CnnCaltech | (2000-1000-700) | -2.58e+02(2.6e-10) | -2.50e+02(5.1e-08) | -2.67e+02(9.4e-11) | -2.50e+02(1.4e-10) | -1.91e+03(1.6e-08) | -2.63e+02(1.7e-09)(+) | -2.51e+02(5.1e-08)(+) | -2.77e+02(1.8e-10)(+) |
| gisette | (3000-1000-700) | -6.85e+07(1.5e-04) | -6.49e+06(3.5e+00) | -1.61e+06(9.3e-11) | -1.41e+06(2.4e-10) | -3.13e+06(5.5e-09) | -6.94e+07(1.5e-04)(+) | -7.22e+09(3.5e+00)(+) | -1.74e+06(2.8e-10)(+) |
| mnist | (1000-780-500) | -1.70e+05(8.1e-10) | -8.40e+05(3.8e-06) | -4.72e+04(1.3e-10) | -4.25e+04(2.6e-10) | -4.45e+05(2.1e-08) | -1.74e+05(1.6e-09)(+) | -8.40e+05(3.8e-06) | -8.55e+04(5.6e-10)(+) |
| randn10 | (10-10-7) | -4.24e+01(6.9e-09) | -4.21e+01(1.4e-07) | -1.66e+02(1.5e-01) | -4.22e+01(1.4e-08) | -8.33e+01(8.5e-07) | -4.24e+01(1.2e-08) | -4.21e+01(1.4e-07) | -2.35e+02(1.5e-01)(+) |
| randn100 | (100-100-70) | -4.99e+03(1.3e-09) | -4.92e+03(9.7e-08) | -1.01e+04(2.0e-09) | -4.89e+03(4.6e-09) | -1.62e+04(1.3e-07) | -4.99e+03(2.2e-09) | -4.92e+03(9.7e-08) | -2.89e+04(6.1e-08)(+) |
| randn1000 | (1000-1000-700) | -5.53e+05(1.7e-08) | -5.54e+05(5.0e-08) | -5.28e+05(1.0e-10) | -5.01e+05(1.1e-10) | -2.54e+06(5.4e-08) | -5.53e+05(1.7e-08) | -5.54e+05(5.0e-08) | -5.33e+05(2.2e-10)(+) |
| sector | (500-1000-700) | -1.64e+03(1.9e-10) | -1.55e+03(4.1e-08) | -1.59e+03(8.9e-11) | -1.54e+03(2.0e-10) | -2.44e+03(3.4e-09) | -1.79e+03(1.2e-09)(+) | -1.55e+03(4.1e-08) | -1.62e+03(1.7e-10)(+) |
| TDT2 | (1000-1000-700) | -6.93e+07(1.4e-04) | -7.53e+05(1.9e-02) | -1.88e+06(8.4e-11) | -1.81e+06(2.0e-10) | -3.07e+06(4.0e-09) | -6.93e+07(1.4e-04) | -5.35e+07(1.9e-02)(+) | -1.91e+06(1.8e-10)(+) |
| w1a | (2470-290-200) | -1.44e+04(1.6e-09) | -1.39e+04(6.2e-08) | -1.41e+04(3.4e-10) | -1.38e+04(2.3e-09) | -8.13e+05(1.7e-07) | -1.48e+04(6.6e-09)(+) | -1.39e+04(6.2e-08) | -1.61e+04(5.0e-09)(+) |
| cifar | (1000-100-90) | -7.78e+03(1.6e-09) | -6.57e+03(8.5e-08) | -7.14e+03(5.1e-10) | -1.66e+04(2.2e-08) | -2.11e+04(1.0e-07) | -2.07e+04(1.8e-08)(+) | -1.87e+04(8.6e-08)(+) | -1.95e+04(1.5e-08)(+) |
| CnnCaltech | (2000-1000-900) | -2.52e+02(5.9e-10) | -2.50e+02(4.4e-08) | -2.55e+02(6.1e-11) | -2.50e+02(2.4e-10) | -1.18e+03(1.2e-08) | -2.58e+02(2.0e-09)(+) | -2.50e+02(4.4e-08) | -2.65e+02(2.7e-10)(+) |
| gisette | (3000-1000-900) | -1.24e+07(5.3e-05) | -3.44e+07(8.6e-03) | -1.47e+06(5.8e-11) | -1.40e+06(3.5e-10) | -2.25e+06(4.7e-09) | -1.34e+07(5.3e-05)(+) | -3.79e+07(8.6e-03)(+) | -1.61e+06(4.5e-10)(+) |
| mnist | (1000-780-650) | -4.97e+04(4.7e-10) | -2.79e+05(4.8e-08) | -4.04e+04(8.8e-11) | -4.78e+04(5.7e-10) | -2.83e+05(1.6e-08) | -6.66e+04(1.7e-09)(+) | -2.79e+05(4.8e-08) | -6.80e+04(5.5e-10)(+) |
| randn10 | (10-10-9) | -4.22e+01(4.2e-09) | -4.21e+01(1.3e-07) | -7.61e+01(1.4e-01) | -5.03e+01(3.9e-07) | -4.35e+01(7.6e-07) | -5.02e+01(3.7e-07)(+) | -5.01e+01(4.5e-07)(+) | -1.13e+02(1.4e-01)(+) |
| randn100 | (100-100-90) | -4.93e+03(9.9e-10) | -4.90e+03(8.7e-08) | -6.65e+03(1.1e-09) | -4.89e+03(8.8e-09) | -9.01e+03(9.9e-08) | -4.93e+03(1.4e-09) | -4.90e+03(8.7e-08) | -1.03e+04(2.3e-08)(+) |
| randn1000 | (1000-1000-900) | -5.22e+05(5.6e-10) | -5.23e+05(4.3e-08) | -5.10e+05(5.9e-11) | -5.01e+05(1.4e-10) | -1.52e+06(2.7e-08) | -5.22e+05(6.2e-10) | -5.23e+05(4.3e-08) | -5.13e+05(2.2e-10)(+) |
| sector | (500-1000-900) | -1.47e+03(2.2e-10) | -1.54e+03(3.9e-08) | -1.55e+03(6.1e-11) | -1.53e+03(2.3e-10) | -2.18e+03(3.3e-09) | -1.72e+03(1.9e-09)(+) | -1.54e+03(3.9e-08) | -1.58e+03(2.8e-10)(+) |
| TDT2 | (1000-1000-900) | -1.26e+07(5.1e-05) | -9.14e+06(2.0e-03) | -1.82e+06(5.7e-11) | -1.80e+06(2.5e-10) | -2.69e+06(5.0e-09) | -1.26e+07(5.1e-05) | -9.83e+06(2.0e-03)(+) | -1.86e+06(2.8e-10)(+) |
| w1a | (2470-290-250) | -1.40e+04(2.1e-09) | -1.53e+04(6.2e-08) | -1.48e+04(2.8e-10) | -1.38e+04(3.7e-09) | -2.22e+06(3.4e-07) | -1.49e+04(8.2e-09)(+) | -1.56e+04(6.2e-08) | -2.70e+04(7.3e-09)(+) |

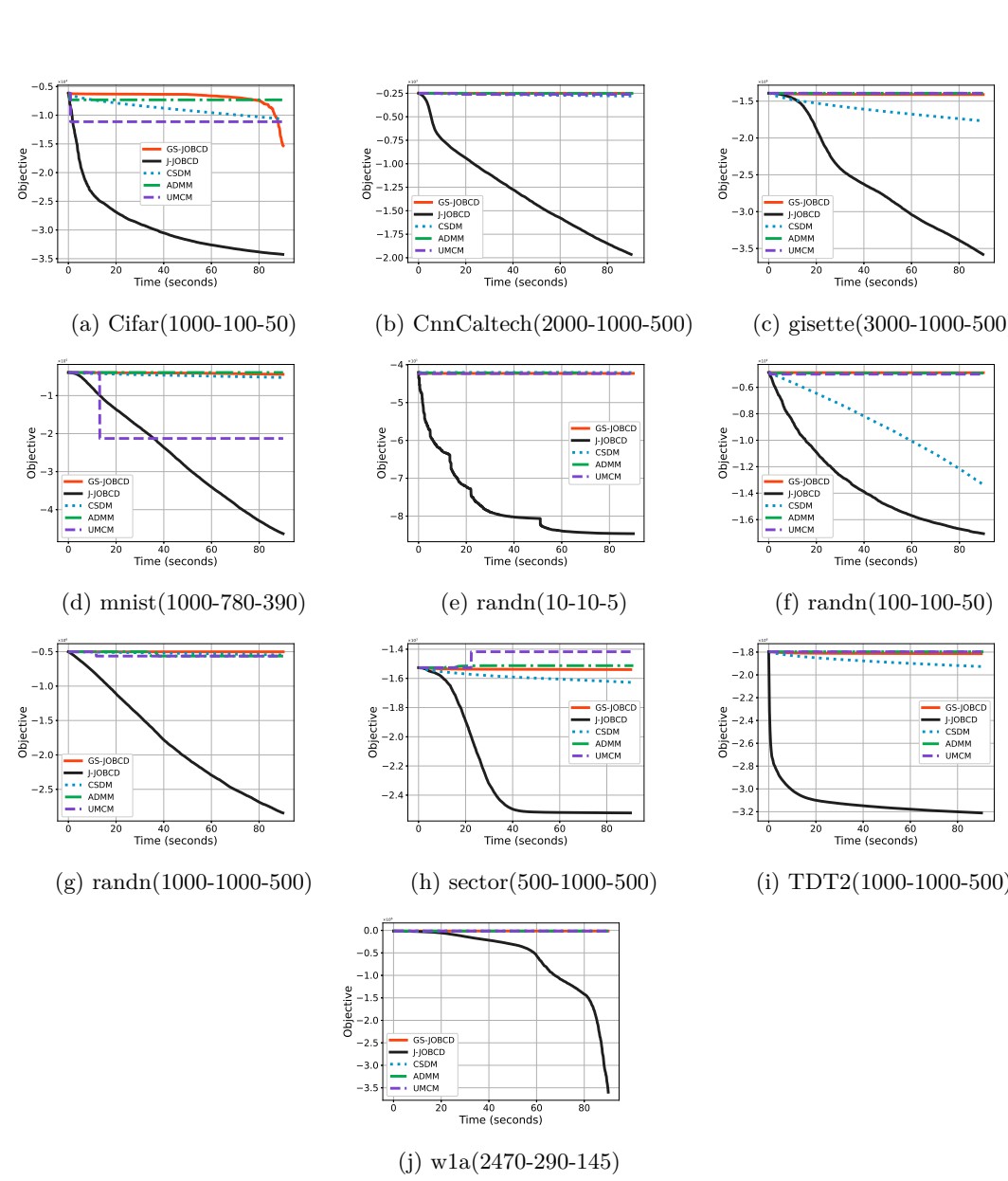

(a) Cifar(1000-100-50)

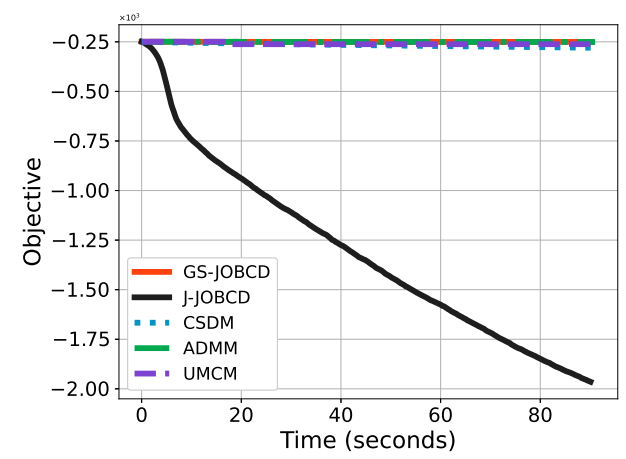

(b) CnnCaltech(2000-1000-500)

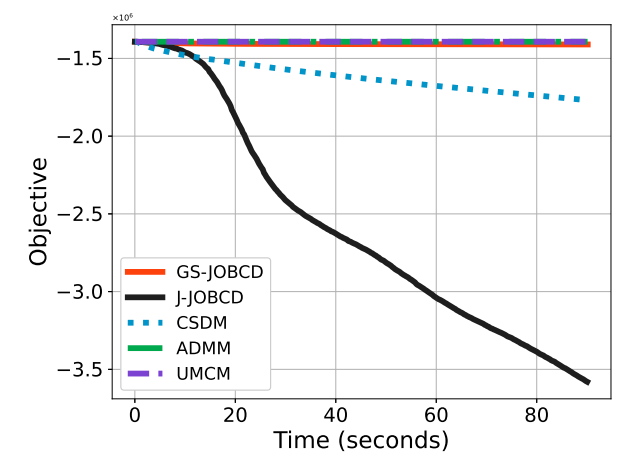

(c) gisette(3000-1000-500)

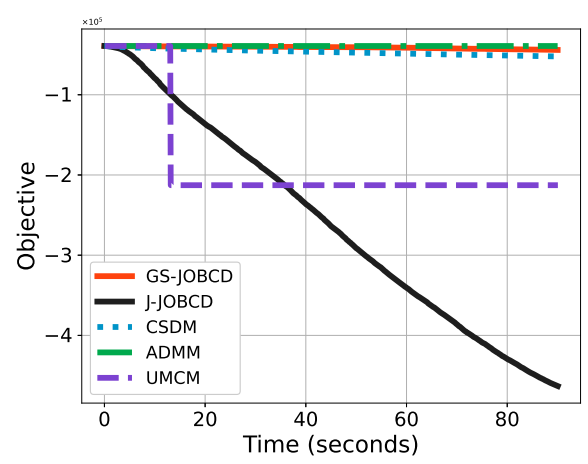

(d) mnist(1000-780-390)

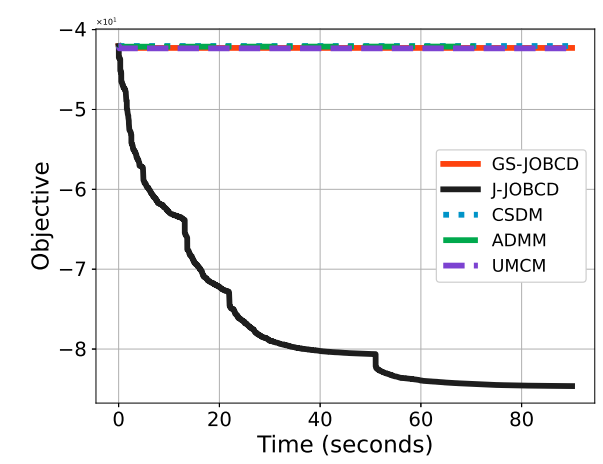

(e) randn(10-10-5)

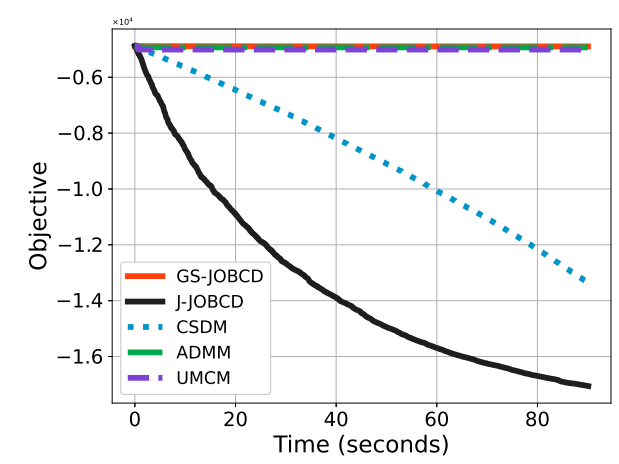

(f) randn(100-100-50)

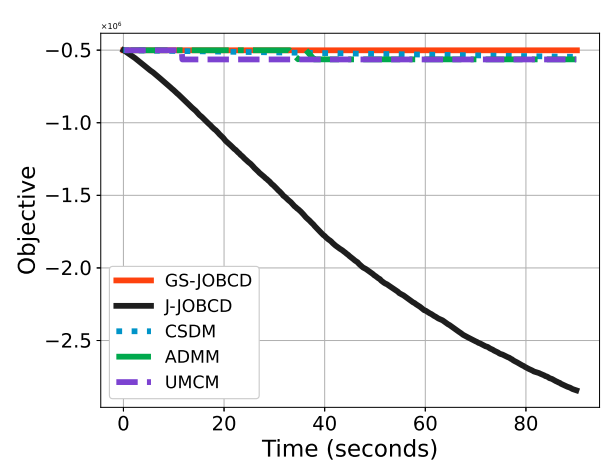

(g) randn(1000-1000-500)

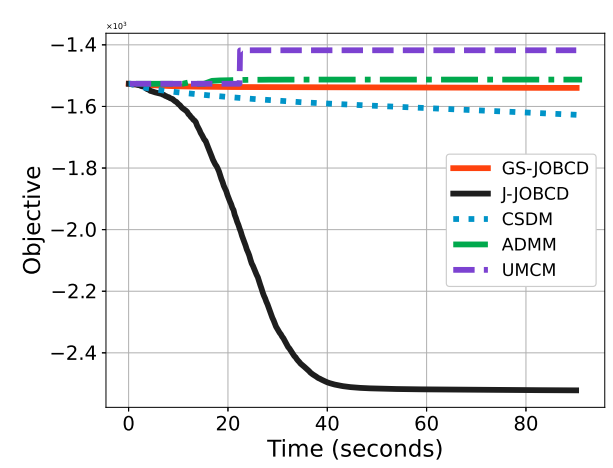

(h) sector(500-1000-500)

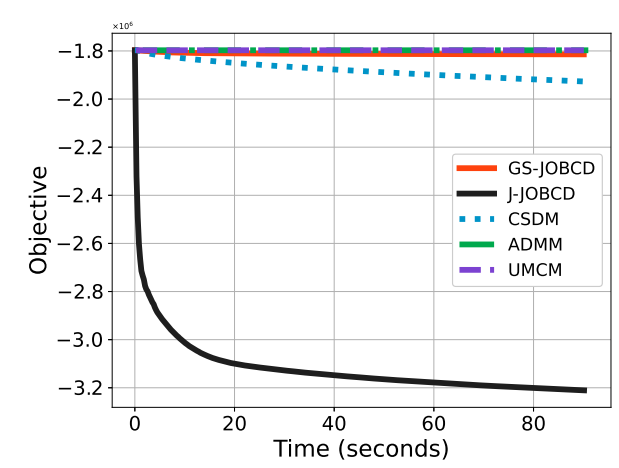

(i) TDT2(1000-1000-500)

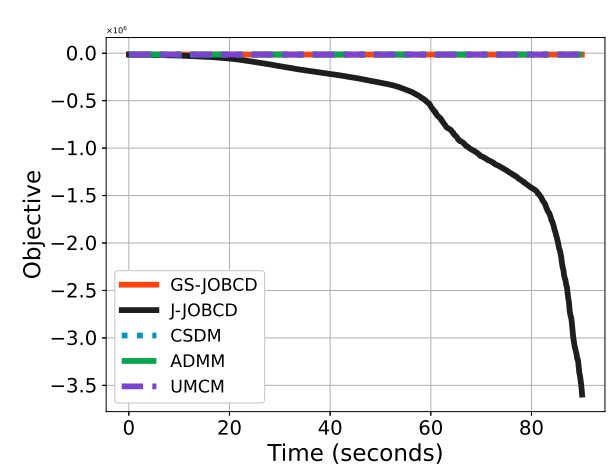

(j) w1a(2470-290-145)

Figure 4: The convergence curve of the compared methods for solving HEVP with varying $(m, n, p)$.

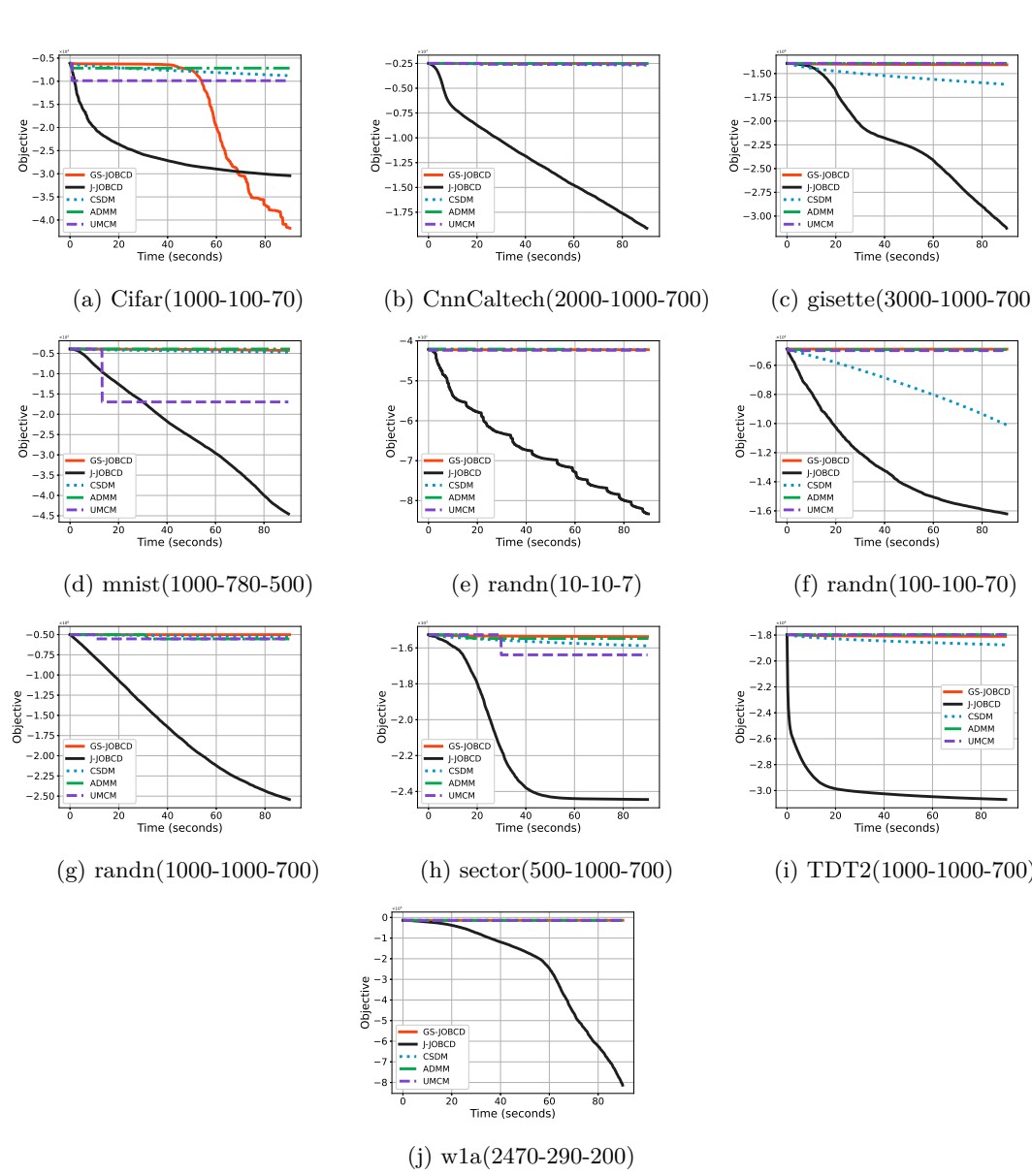

(a) Cifar(1000-100-70)  (b) CnnCaltech(2000-1000-700)  (c) gisette(3000-1000-700)

(d) mnist(1000-780-500)  (e) randn(10-10-7)  (f) randn(100-100-70)

(g) randn(1000-1000-700)  (h) sector(500-1000-700)  (i) TDT2(1000-1000-700)

(j) w1a(2470-290-200)

Figure 5: The convergence curve of the compared methods for solving HEVP with varying $(m, n, p)$.

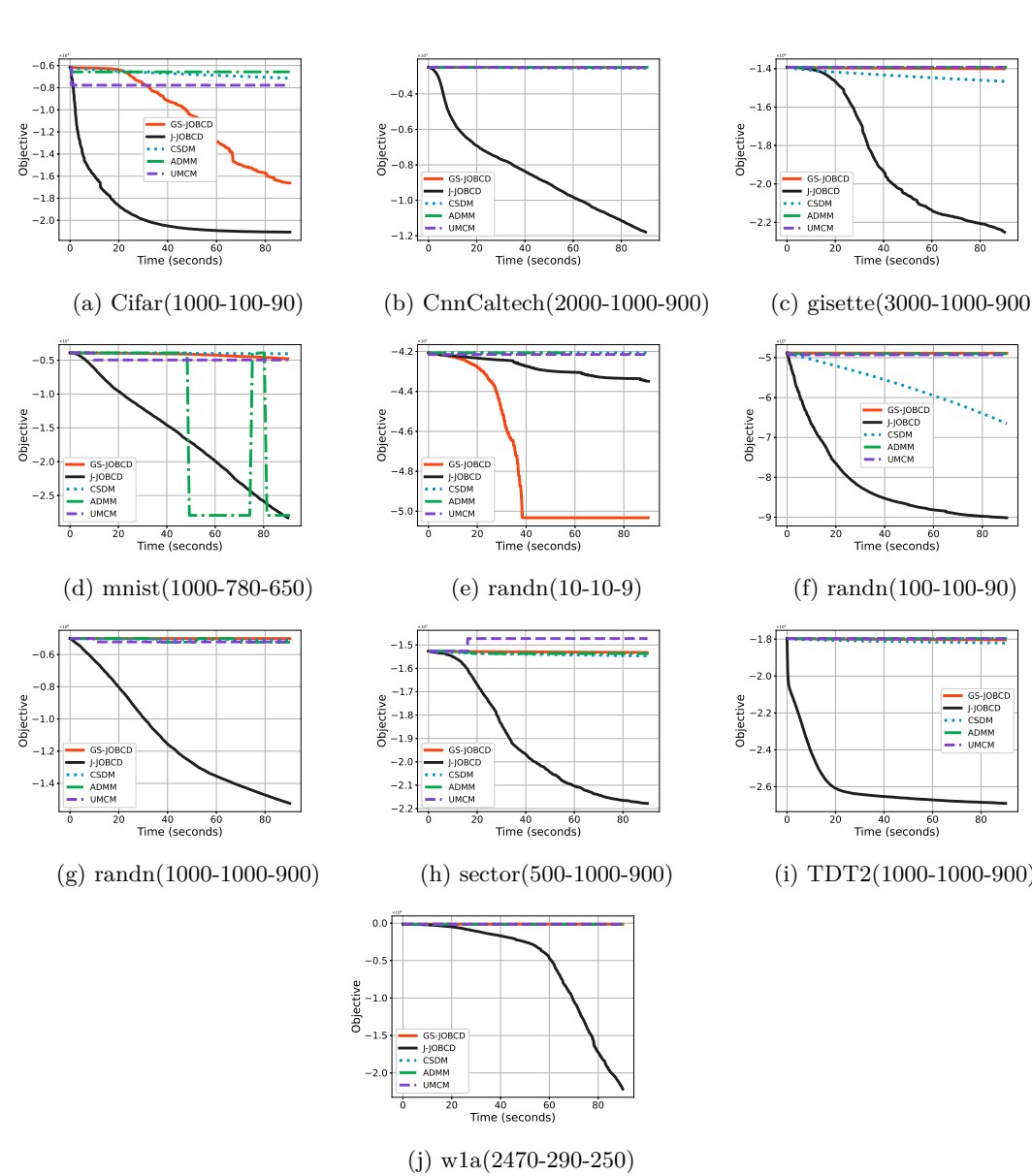

(a) Cifar(1000-100-90)  (b) CnnCaltech(2000-1000-900)  (c) gisette(3000-1000-900)

(d) mnist(1000-780-650)  (e) randn(10-10-9)  (f) randn(100-100-90)

(g) randn(1000-1000-900)  (h) sector(500-1000-900)  (i) TDT2(1000-1000-900)

(j) w1a(2470-290-250)

Figure 6: The convergence curve of the compared methods for solving HEVP with varying $(m, n, p)$.

## F.6.2 Hyperbolic Structural Probe Problem

Table 5: The convergence curve of the compared methods for solving HSPP. (+) indicates that after the convergence of the **CSDM**, utilizing the **J-OBCD** for optimization markedly enhances the objective value. The $1^{st}, 2^{nd}$, and $3^{rd}$ best results are colored with red, green and blue, respectively. $(n, p)$ represents the dimension and p-value of the **J** orthogonal matrix (square matrix). The value in () stands for $\frac{1}{n^2} \sum_{ij}^{n} |\mathbf{X}^\top \mathbf{J} \mathbf{X} - \mathbf{J}|_{ij}$ and cells with this value greater than 1e-7 are highlighted in gray.

| datasetname | (m-n-p) | ADMM | UMCM | CSDM | J-JOBCD | VR-J-JOBCD | CSDM+J-JOBCD |
|---|---|---|---|---|---|---|---|
| | | | | time limit=30s | | | |
| 20News | (9423-50-25) | -7.56e+00(7.6e-06) | -5.59e+01(3.0e-04) | -3.21e+01(1.0e-04) | -2.52e+02(2.4e-08) | -2.96e+02(3.5e-08) | -3.21e+01(1.0e-04) |
| Cifar | (10000-50-25) | -4.81e+04(1.3e-05) | -1.00e+04(4.1e-04) | -5.15e+03(2.4e-04) | -5.26e+04(1.6e-08) | -3.19e+04(1.7e-08) | -5.15e+03(2.4e-04)(+) |
| cnnCaltech | (3000-96-48) | -9.35e+00(4.0e-06) | -1.90e+02(5.6e-06) | -2.11e+02(7.1e-06) | -1.65e+04(2.8e-08) | -1.34e+04(2.3e-08) | -2.11e+02(7.1e-06)(+) |
| gisette | (6000-50-25) | -1.76e+04(1.1e-05) | -4.50e+03(1.3e-05) | -5.00e+03(3.3e-04) | -5.81e+04(1.6e-08) | -5.11e+04(1.7e-08) | -5.00e+03(3.3e-04)(+) |
| Mnist | (6000-92-46) | -4.90e+04(6.8e-06) | -1.67e+04(9.5e-06) | -2.27e+03(6.3e-06) | -3.08e+03(1.1e-08) | -1.99e+04(3.1e-08) | -2.27e+03(6.3e-06) |
| news20 | (7967-50-25) | -3.91e-03(7.6e-06) | -1.25e+00(4.2e-04) | -1.06e+00(1.0e-04) | -7.03e+00(6.2e-08) | -6.92e+00(5.6e-08) | -1.06e+00(1.0e-04)(+) |
| randn5000 | (5000-100-50) | -8.82e+02(1.6e-06) | -6.37e+02(4.2e-04) | -1.56e+02(2.6e-06) | -1.56e+02(3.9e-09) | -4.71e+03(1.9e-08) | -1.56e+02(2.6e-06) |
| w1a | (2477-100-50) | -3.38e+02(3.6e-06) | -9.70e+02(3.1e-04) | -1.65e+02(5.9e-06) | -2.97e+03(9.0e-09) | -3.44e+03(2.1e-08) | -1.65e+02(5.9e-06) |
| 20News | (9423-50-35) | -1.01e+01(7.6e-06) | -6.95e+01(1.5e-03) | -1.32e+01(7.0e-06) | -1.50e+02(8.1e-09) | -1.95e+02(1.2e-08) | -1.32e+01(7.0e-06) |
| Cifar | (10000-50-35) | -6.60e+04(1.2e-05) | -1.11e+04(4.1e-04) | -4.50e+03(1.9e-04) | -2.99e+04(1.5e-08) | -3.78e+04(1.7e-08) | -4.50e+03(1.9e-04)(+) |
| cnnCaltech | (3000-96-70) | -1.19e+01(4.0e-06) | -1.79e+02(7.2e-06) | -1.33e+02(4.9e-06) | -1.43e+04(2.3e-08) | -1.79e+04(2.7e-08) | -1.33e+02(4.9e-06)(+) |
| gisette | (6000-50-35) | -2.46e+04(1.0e-05) | -4.87e+03(1.2e-05) | -3.87e+03(8.7e-05) | -5.87e+04(3.0e-08) | -6.76e+04(3.0e-08) | -3.87e+03(8.7e-05)(+) |
| Mnist | (6000-92-70) | -7.45e+04(6.3e-06) | -1.71e+04(3.2e-04) | -2.15e+03(3.0e-06) | -9.38e+03(1.1e-08) | -5.00e-01(3.3e-09) | -2.15e+03(3.0e-06) |
| news20 | (7967-50-35) | -1.56e-02(7.6e-06) | -1.34e+00(3.0e-04) | -9.14e-01(1.2e-04) | -9.70e+00(6.1e-08) | -1.06e+01(6.2e-08) | -9.14e-01(1.2e-04)(+) |
| randn5000 | (5000-100-75) | -1.57e+03(4.5e-06) | -6.27e+02(8.9e-04) | -9.15e+01(5.4e-06) | -1.63e+03(1.7e-08) | -4.35e+01(7.1e-10) | -9.15e+01(5.4e-06) |
| w1a | (2477-100-75) | -6.77e+02(3.3e-06) | -1.06e+03(6.8e-06) | -1.90e+02(4.5e-05) | -1.80e+03(3.5e-09) | -1.80e+03(5.1e-09) | -1.90e+02(4.5e-05) |
| 20News | (9423-50-45) | -1.51e+01(7.6e-06) | -8.44e+01(1.1e-03) | -9.15e+00(1.4e-05) | -3.84e+01(1.1e-09) | -1.96e+02(1.1e-08) | -9.15e+00(1.4e-05) |
| Cifar | (10000-50-45) | -8.45e+04(1.4e-05) | -8.15e+03(3.2e-04) | -3.75e+03(1.9e-04) | -2.69e+03(2.3e-09) | 5.00e+00(3.9e-09) | -3.75e+03(1.9e-04)(+) |
| cnnCaltech | (3000-96-85) | -7.38e+00(4.0e-06) | -9.28e+01(8.2e-06) | -8.77e+01(5.4e-05) | -1.79e+04(3.4e-08) | -1.43e+04(3.1e-08) | -8.77e+01(5.4e-05)(+) |
| gisette | (6000-50-45) | -3.29e+04(9.6e-06) | -3.88e+03(1.3e-05) | -2.65e+03(2.3e-04) | -5.02e+04(1.5e-06) | -4.18e+04(1.6e-08) | -2.65e+03(2.3e-04)(+) |
| Mnist | (6000-92-85) | -8.69e+04(7.2e-06) | -2.49e+04(1.1e-03) | -6.49e+02(3.0e-06) | -3.13e+04(1.1e-06) | -3.10e+04(1.3e-08) | -6.49e+02(3.0e-06) |
| news20 | (7967-50-45) | -3.91e-02(7.6e-06) | -1.19e+00(3.0e-04) | -8.98e-01(6.0e-05) | -4.00e+00(1.5e-07) | -3.60e+00(4.7e-07) | -8.98e-01(6.0e-05)(+) |
| randn5000 | (5000-100-85) | -1.66e+03(2.7e-06) | -3.77e+02(9.2e-04) | -1.05e+02(9.8e-06) | -2.68e+03(2.3e-09) | -1.99e+03(2.0e-10) | -1.05e+02(9.8e-06) |
| w1a | (2477-100-90) | -4.26e+02(3.7e-06) | -7.05e+02(3.1e-04) | -1.31e+02(3.0e-06) | -6.27e+02(2.0e-09) | -6.26e+02(2.4e-09) | -1.31e+02(3.0e-06) |
| | | | | time limit=60s | | | |
| 20News | (9423-50-25) | -7.77e+00(7.6e-06) | -5.79e+01(4.3e-05) | -3.21e+01(1.0e-04) | -3.57e+02(4.6e-08) | -2.96e+02(3.5e-08) | -3.21e+01(1.0e-04) |
| Cifar | (10000-50-25) | -4.88e+04(1.1e-05) | -1.01e+04(2.2e-05) | -7.96e+03(3.6e-04) | -8.47e+04(4.1e-08) | -5.92e+04(3.6e-08) | -7.96e+03(3.6e-04)(+) |
| cnnCaltech | (3000-96-48) | -1.19e+01(4.0e-06) | -1.89e+02(5.6e-06) | -3.58e+02(7.6e-06) | -1.34e+04(2.3e-08) | -5.11e+04(1.7e-08) | -3.58e+02(7.6e-06)(+) |
| gisette | (6000-50-25) | -1.81e+04(8.9e-06) | -4.49e+03(1.4e-05) | -7.58e+03(4.9e-04) | -6.78e+04(2.1e-08) | -5.11e+04(1.7e-08) | -7.58e+03(4.9e-04)(+) |
| Mnist | (6000-92-46) | -5.04e+04(6.8e-06) | -1.67e+04(8.7e-06) | -2.27e+03(6.3e-06) | -4.25e+04(5.2e-08) | -2.28e+04(3.8e-08) | -2.27e+03(6.3e-06) |
| news20 | (7967-50-25) | -3.91e-03(7.6e-06) | -1.21e+00(1.3e-05) | -1.21e+00(1.0e-04) | -1.06e+01(1.1e-07) | -1.17e+01(1.1e-07) | -1.21e+00(1.0e-04)(+) |
| randn5000 | (5000-100-50) | -9.31e+02(2.3e-06) | -6.37e+02(4.2e-04) | -1.56e+02(2.6e-06) | -8.38e+00(6.9e-11) | -4.71e+03(1.9e-08) | -1.56e+02(2.6e-06) |
| w1a | (2477-100-50) | -3.71e+02(3.5e-06) | -9.70e+02(3.1e-04) | -1.65e+02(5.9e-06) | -2.19e+03(4.8e-09) | -3.44e+03(2.1e-08) | -1.65e+02(5.9e-06) |
| 20News | (9423-50-35) | -1.03e+01(7.6e-06) | -6.95e+01(1.5e-03) | -1.32e+01(7.0e-06) | -4.52e+02(5.2e-08) | -1.95e+02(1.2e-08) | -1.32e+01(7.0e-06) |
| Cifar | (10000-50-35) | -6.68e+04(1.0e-05) | -1.10e+04(1.1e-05) | -6.80e+03(3.2e-04) | -9.09e+04(4.0e-08) | -6.64e+04(3.4e-08) | -6.80e+03(3.2e-04)(+) |
| cnnCaltech | (3000-96-70) | -1.46e+01(4.0e-06) | -1.79e+02(7.0e-06) | -2.29e+02(4.6e-06) | -1.87e+04(2.5e-08) | -1.79e+04(2.7e-08) | -2.29e+02(4.6e-06)(+) |
| gisette | (6000-50-35) | -2.52e+04(1.0e-05) | -4.89e+03(1.3e-05) | -5.53e+03(9.5e-05) | -7.67e+04(3.6e-08) | -6.76e+04(3.0e-08) | -5.53e+03(9.5e-05)(+) |
| Mnist | (6000-92-70) | -7.71e+04(6.1e-06) | -1.71e+04(3.2e-04) | -2.15e+03(3.0e-06) | -9.38e+03(1.1e-08) | -5.00e-01(3.3e-09) | -2.15e+03(3.0e-06) |
| news20 | (7967-50-35) | -1.56e-02(7.7e-06) | -1.42e+00(1.3e-05) | -1.26e+00(1.0e-04) | -1.46e+01(1.0e-07) | -1.54e+01(1.1e-07) | -1.26e+00(1.0e-04)(+) |
| randn5000 | (5000-100-75) | -1.65e+03(5.0e-06) | -6.27e+02(8.9e-04) | -9.15e+01(5.4e-06) | -2.01e+03(1.7e-08) | -4.35e+01(7.1e-10) | -9.15e+01(5.4e-06) |
| w1a | (2477-100-75) | -7.26e+02(3.2e-06) | -1.06e+03(6.8e-06) | -1.90e+02(4.5e-05) | -1.58e+03(3.0e-09) | -1.80e+03(5.1e-09) | -1.90e+02(4.5e-05) |
| 20News | (9423-50-45) | -1.54e+01(7.6e-06) | -8.44e+01(1.1e-03) | -9.15e+00(1.4e-05) | -8.68e+00(5.4e-10) | -1.96e+02(1.1e-08) | -9.15e+00(1.4e-05) |
| Cifar | (10000-50-45) | -8.59e+04(1.3e-05) | -8.08e+03(1.6e-05) | -5.64e+03(2.9e-04) | -9.56e+03(5.0e-07) | -2.65e+03(4.4e-07) | -5.64e+03(2.9e-04)(+) |
| cnnCaltech | (3000-96-85) | -9.45e+00(4.0e-06) | -9.26e+01(8.4e-06) | -1.67e+02(5.5e-05) | -1.65e+04(2.8e-08) | -1.43e+04(3.1e-08) | -1.67e+02(5.5e-05)(+) |
| gisette | (6000-50-45) | -3.32e+04(9.4e-06) | -3.87e+03(1.3e-05) | -3.81e+03(3.0e-04) | -4.50e+04(7.2e-07) | -4.18e+04(1.5e-07) | -3.81e+03(3.0e-04)(+) |
| Mnist | (6000-92-85) | -9.06e+04(6.8e-06) | -2.49e+04(1.1e-03) | -6.49e+02(3.0e-06) | -1.63e+04(1.2e-08) | -3.10e+04(1.3e-08) | -6.49e+02(3.0e-06) |
| news20 | (7967-50-45) | -3.91e-02(7.6e-06) | -1.33e+00(1.1e-05) | -3.20e+00(8.0e-05) | -6.17e+00(5.0e-07) | -5.50e+00(6.0e-07) | -3.20e+00(8.0e-05)(+) |
| randn5000 | (5000-100-85) | -1.76e+03(3.2e-06) | -3.77e+02(9.2e-04) | -1.05e+02(9.8e-06) | -1.05e+02(9.8e-06) | -1.05e+02(9.8e-06) | -1.05e+02(9.8e-06) |
| w1a | (2477-100-90) | -4.38e+02(3.6e-06) | -7.05e+02(3.1e-04) | -1.31e+02(3.0e-06) | -3.07e+01(9.3e-11) | -6.26e+02(2.4e-09) | -1.31e+02(3.0e-06) |
| | | | | time limit=90s | | | |
| 20News | (9423-50-25) | -7.96e+00(7.6e-06) | -5.79e+01(4.3e-05) | -3.21e+01(1.0e-04) | -4.70e+01(2.9e-09) | -2.96e+02(3.5e-08) | -3.21e+01(1.0e-04) |
| Cifar | (10000-50-25) | -4.93e+04(1.2e-05) | -1.01e+04(9.8e-06) | -1.05e+04(4.9e-04) | -1.07e+05(6.5e-08) | -8.44e+04(5.9e-08) | -1.05e+04(4.9e-04)(+) |
| cnnCaltech | (3000-96-48) | -1.55e+01(4.0e-06) | -1.89e+02(5.3e-06) | -4.99e+02(8.3e-06) | -6.07e+04(1.4e-08) | -5.11e+04(1.7e-08) | -4.99e+02(8.3e-06)(+) |
| gisette | (6000-50-25) | -1.87e+04(9.3e-06) | -4.51e+03(1.4e-05) | -9.52e+03(6.1e-04) | -6.07e+04(1.4e-08) | -5.11e+04(1.7e-08) | -9.52e+03(6.1e-04)(+) |
| Mnist | (6000-92-46) | -5.24e+04(6.4e-06) | -1.67e+04(8.7e-06) | -2.27e+03(6.3e-06) | -1.35e+04(2.3e-08) | -2.28e+04(3.8e-08) | -2.27e+03(6.3e-06) |
| news20 | (7967-50-25) | -3.91e-03(7.6e-06) | -1.21e+00(1.4e-05) | -1.34e+00(2.4e-05) | -1.37e+01(1.8e-07) | -1.43e+01(1.6e-07) | -1.34e+00(2.4e-05)(+) |
| randn5000 | (5000-100-50) | -9.98e+02(2.8e-06) | -6.37e+02(4.2e-04) | -1.56e+02(2.6e-06) | -1.09e+03(3.8e-09) | -4.71e+03(1.9e-08) | -1.56e+02(2.6e-06) |
| w1a | (2477-100-50) | -4.16e+02(3.4e-06) | -9.70e+02(3.1e-04) | -1.65e+02(5.9e-06) | -2.80e+03(7.4e-09) | -3.44e+03(2.1e-08) | -1.65e+02(5.9e-06) |
| 20News | (9423-50-35) | -1.04e+01(7.6e-06) | -6.95e+01(1.5e-03) | -1.32e+01(7.0e-06) | -1.94e+02(1.1e-08) | -1.95e+02(1.2e-08) | -1.32e+01(7.0e-06) |
| Cifar | (10000-50-35) | -6.83e+04(1.1e-05) | -1.10e+04(1.2e-05) | -8.83e+03(4.1e-04) | -8.76e+04(5.4e-08) | -9.79e+04(5.4e-08) | -8.83e+03(4.1e-04)(+) |
| cnnCaltech | (3000-96-70) | -1.79e+01(4.0e-06) | -1.79e+02(7.3e-06) | -3.02e+02(5.8e-06) | -1.26e+04(2.3e-08) | -1.79e+04(2.7e-08) | -3.02e+02(5.8e-06)(+) |
| gisette | (6000-50-35) | -2.55e+04(9.7e-06) | -4.89e+03(1.4e-05) | -6.74e+03(9.6e-05) | -7.15e+04(3.3e-08) | -6.74e+03(9.6e-05)(+) | -6.74e+03(9.6e-05)(+) |
| Mnist | (6000-92-70) | -8.05e+04(6.5e-06) | -1.71e+04(3.2e-04) | -2.15e+03(3.0e-06) | -9.38e+03(1.1e-08) | -5.00e-01(3.3e-09) | -2.15e+03(3.0e-06) |
| news20 | (7967-50-35) | -1.56e-02(7.6e-06) | -1.42e+00(1.4e-05) | -1.48e+00(1.8e-05) | -1.98e+01(1.8e-07) | -1.56e+00(1.0e-04) | -1.56e+00(1.0e-04)(+) |
| randn5000 | (5000-100-75) | -1.72e+03(5.0e-06) | -6.27e+02(8.9e-04) | -9.15e+01(5.4e-06) | -3.25e+00(6.9e-10) | -4.35e+01(7.1e-10) | -9.15e+01(5.4e-06) |
| w1a | (2477-100-75) | -8.30e+02(3.3e-06) | -1.06e+03(6.8e-06) | -1.90e+02(4.5e-05) | -1.12e+03(1.9e-09) | -1.80e+03(5.1e-09) | -1.90e+02(4.5e-05) |
| 20News | (9423-50-45) | -1.58e+01(7.6e-06) | -8.44e+01(1.1e-03) | -9.15e+00(1.4e-05) | -1.76e+02(1.1e-08) | -1.96e+02(2.0e-07) | -9.15e+00(1.4e-05) |
| Cifar | (10000-50-45) | -8.72e+04(9.6e-06) | -8.11e+03(1.3e-05) | -6.92e+03(3.5e-04) | -2.79e+03(1.0e-08) | -2.80e+03(8.9e-07) | -6.92e+03(3.5e-04)(+) |
| cnnCaltech | (3000-96-85) | -1.19e+01(4.0e-06) | -9.25e+01(8.7e-06) | -2.48e+02(5.4e-05) | -1.57e+04(3.5e-08) | -1.43e+04(3.1e-08) | -2.48e+02(5.4e-05)(+) |
| gisette | (6000-50-45) | -3.32e+04(9.7e-06) | -3.88e+03(1.4e-05) | -4.39e+03(3.2e-04) | -5.42e+04(3.8e-06) | -6.04e+04(1.2e-06) | -4.39e+03(3.2e-04)(+) |
| Mnist | (6000-92-85) | -9.45e+04(8.8e-06) | -2.49e+04(1.1e-03) | -6.49e+02(3.0e-06) | -4.61e+04(3.8e-10) | -3.10e+04(1.3e-08) | -6.49e+02(3.0e-06) |
| news20 | (7967-50-45) | -3.91e-02(7.6e-06) | -1.33e+00(1.2e-05) | -4.01e+00(1.0e-04) | -7.02e+00(5.9e-07) | -5.68e+00(6.0e-07) | -4.01e+00(1.0e-04)(+) |
| randn5000 | (5000-100-85) | -1.81e+03(3.0e-06) | -3.77e+02(9.2e-04) | -1.05e+02(9.8e-06) | -9.28e+02(5.3e-10) | -1.99e+03(2.0e-10) | -1.05e+02(9.8e-06) |
| w1a | (2477-100-90) | -4.53e+02(3.6e-06) | -7.05e+02(3.1e-04) | -1.31e+02(3.0e-06) | -1.54e+03(1.1e-08) | -6.26e+02(2.4e-09) | -1.31e+02(3.0e-06) |

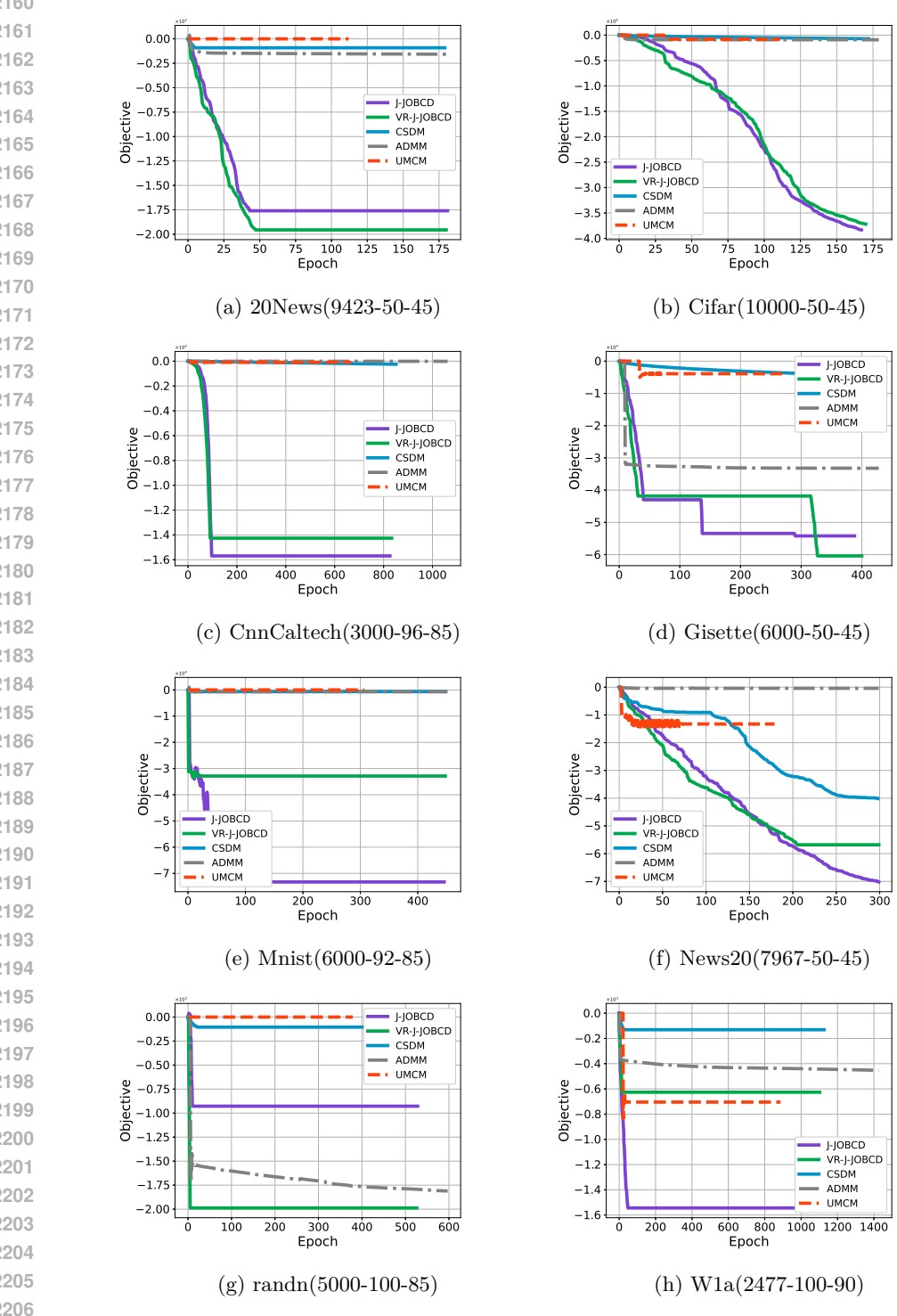

(a) 20News(9423-50-45)

(b) Cifar(10000-50-45)

(c) CnnCaltech(3000-96-85)

(d) Gisette(6000-50-45)

(e) Mnist(6000-92-85)

(f) News20(7967-50-45)

(g) randn(5000-100-85)

(h) W1a(2477-100-90)

Figure 7: Comparisons of objective values $(F(\mathbf{X}) - F^0)$ of HSPP for all the compared methods by epochs with different parameters $(m - n - p)$.

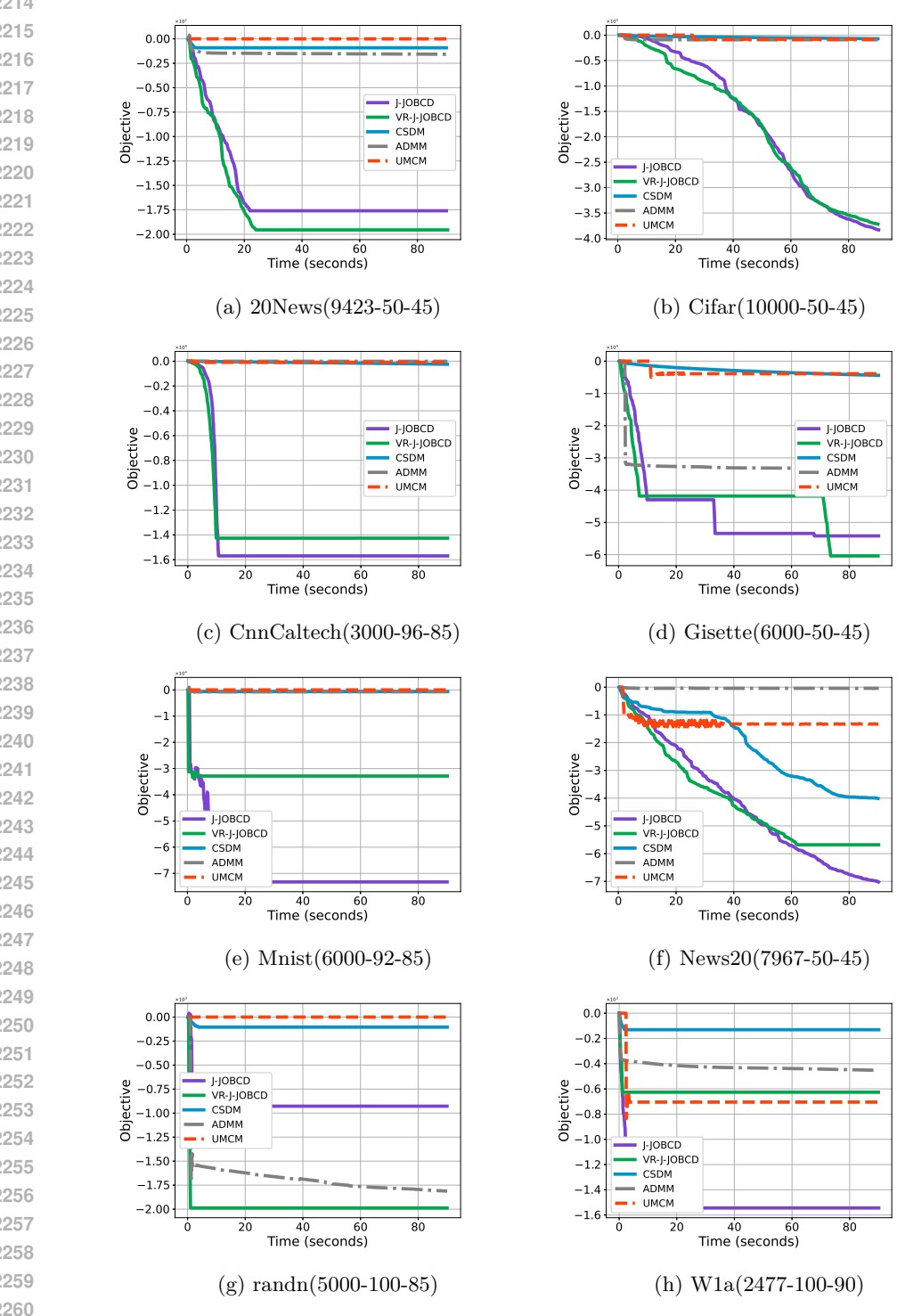

(a) 20News(9423-50-45)

(b) Cifar(10000-50-45)

(c) CnnCaltech(3000-96-85)

(d) Gisette(6000-50-45)

(e) Mnist(6000-92-85)

(f) News20(7967-50-45)

(g) randn(5000-100-85)

(h) W1a(2477-100-90)

Figure 8: Comparisons of objective values $(F(\mathbf{X}) - F^0)$ of HSPP for all the compared methods by time with different parameters $(m - n - p)$.

F.6.3    ULTRA-HYPERBOLIC KNOWLEDGE GRAPH EMBEDDING

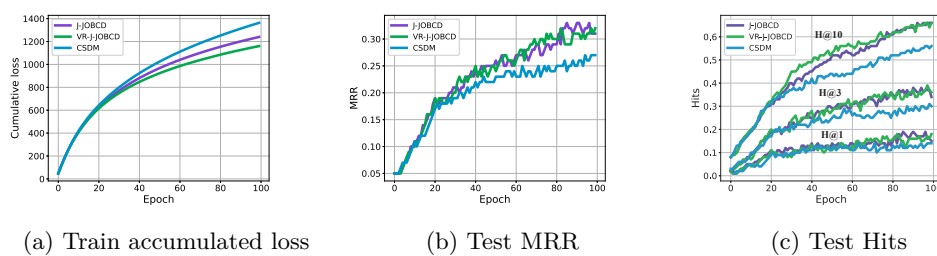

(a) Train accumulated loss        (b) Test MRR        (c) Test Hits

Figure 9: Epoch performance of CS, J-JOBCD, and VR-J-JOBCD in training UltraE on FB15k.

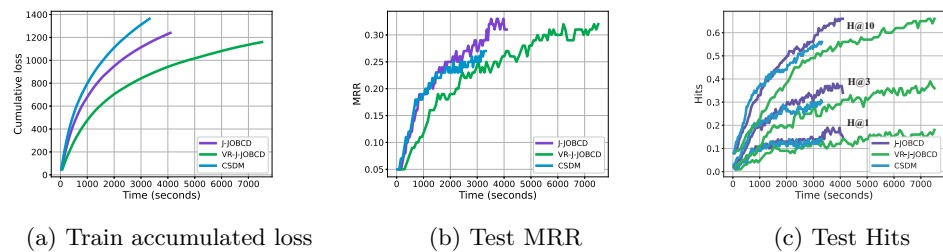

(a) Train accumulated loss        (b) Test MRR        (c) Test Hits

Figure 10: Time performance of CS, J-JOBCD, and VR-J-JOBCD in training UltraE on FB15k.

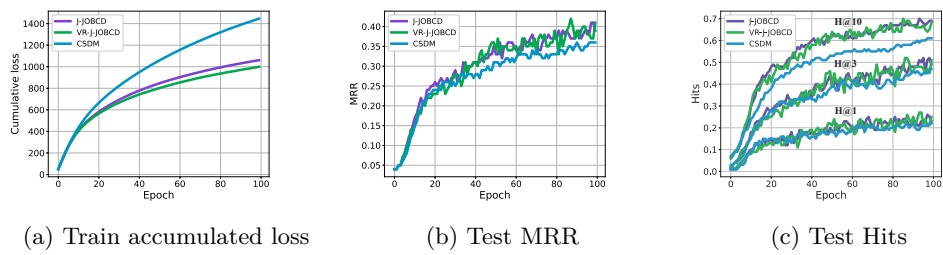

(a) Train accumulated loss        (b) Test MRR        (c) Test Hits

Figure 11: Epoch performance of CSDM, J-JOBCD, and VR-J-JOBCD in training UltraE on WN18RR.

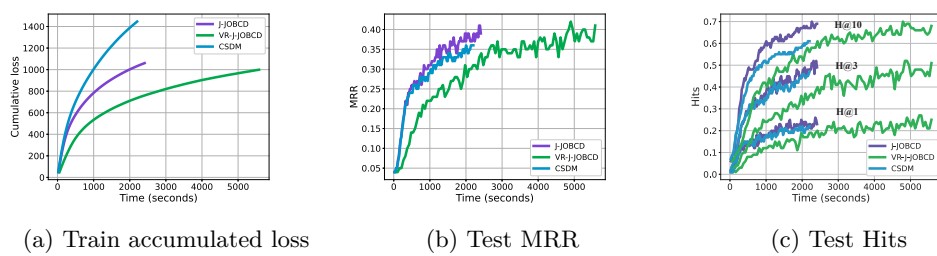

(a) Train accumulated loss        (b) Test MRR        (c) Test Hits

Figure 12: Time performance of CSDM, J-JOBCD, and VR-J-JOBCD in training UltraE on WN18RR.

