# OpenReview forum: "Block Coordinate Descent Methods for Optimization under J-Orthogonality Constraints with Applications"
_ICLR.cc/2025/Conference — Submitted to ICLR 2025_

### Official Review · Reviewer_1LQB · 2024-11-02

**Soundness:** 3
**Presentation:** 3
**Contribution:** 2
**Rating:** 6
**Confidence:** 3

**Summary:**

The paper extends the findings in [52] to the J-orthogonal (hyperbolic orthogonal) case. A new algorithm, JOBCD, is introduced along with a convergence analysis. The authors further develop two JOBCD variants: GS-JOBCD and VR-J-JOBCD which enhance robustness in convergence and efficiency, respectively. Finally, numerical simulations are provided to support the theoretical insights of their study. I would be open to raising my score if the authors can address certain issues I’ve mentioned.

**Strengths:**

The idea makes sense and the problem is of interest to the community. The technical contribution is solid: the authors employed the majorization-minimization method to develop a majorization function for the original forms under hyperbolic orthogonal constraints. They then analyzed the optimal solution of this majorization function, simplifying the hyperbolic orthogonal constraints to the standard orthogonal case as detailed in Proposition 2.2.

**Weaknesses:**

A detailed comparison with [52] would be valuable to better understand the technical challenges posed by the gap between hyperbolic orthogonal and standard orthogonal constraints. I’m also curious about whether the hyperbolic orthogonal constraints will impact the convergence rate compared to [52].

**Questions:**

See weakness.

---

> ### Author Response · Authors · 2024-11-19
> **Rebuttal by Authors**
>
> **Response**:
>
> 1. Difference in form
>
> The structures of Orthogonality and J-orthogonality are quite different.
>
> Orthogonality constraint：the set of  $\mathbf{X}^\top$ $\mathbf{J}$ $\mathbf{X}$= $\mathbf{J}$ where $\mathbf{J}=\mathbf{E}_n$ is fixed。
>
> J-orthogonality constraint：the set of $\mathbf{X}^\top$ $\mathbf{J}$ $\mathbf{X}$= $\mathbf{J}$ where $\mathbf{J}$ is an $n \times n$ diagonal matrix with random$\pm 1s$。This means that for a J-orthogonal optimization problem with dimension n, there are $\mathcal{O}(n^a)$ problem formulations, where $a \geq 2$.
>
> 2. The difficulty in solving the subproblems.
>
> For orthogonal optimization problems, it has been mentioned in [52] that the block size k>2 can be chosen. However, for J-orthogonal optimization problems, if k>2 is selected, an exact solution will not be obtainable. We will use $k=3$ as an example to illustrate. According to Proposition 2.2, when $k=3$, $p$ can take values $\lbrace 0,1,2,3 \rbrace$:
>
> (1)When $p=0$ or $p=3$, the problem degenerates into an optimization problem under orthogonality constraints, for which exact solutions can be obtained based on existing research.
>
> (2)When $p=2$, $U_1$ and $V_1$ are different 2x2 orthogonal matrices, requiring two sets of $\sin(x)$ and $\cos(x)$ functions $\cos(x_1), \sin(x_1), \cos(x_2), \sin(x_2)$ for modeling, while $c$ and $s$ need to be modeled using a set of $\cosh(x)$ and $\sinh(x)$ functions $\cosh(x_3), \sinh(x_3)$. Although we can simplify similarly to what is mentioned in our paper, using $\tan(x)=\sin(x)/\cos(x)$ and $\tanh(x)=\sinh(x)/\cosh(x)$, we ultimately still need to solve a 3-variable optimization problem where $\tan(x_1), \tan(x_2), \tanh(x_3)$ are coupled together (e.g., $\tan(x_1)\times\tan(x_2)\times\tanh(x_3)$, $\tan(x_1)\times\tanh(x_3)^2$), and there is currently no exact method to solve such complex nonlinear relationships.
>
> (3)When $p=1$, the problem to be solved is structurally similar to that in $p=2$.
> The above is a simplified analysis for the case of $k=3$
>
> 3. convergence rate
>
> For a variable with orthogonal optimization, it is bounded: $\|\mathbf{X}\|^2_2=1$. However, J-orthogonal matrices are unbounded. This results in some differences in the final convergence rate. According to the proof of Theorem 4.9 in the appendix of our paper, the sequence $\lbrace \mathbf{X}^t \rbrace^\infty_{t=0}$ of GS-JOBCD has finite length property that: $\forall t,  \sum_{i=1}^t E_{\xi^{t}}[\| \mathbf{X}^{t+1}-\mathbf{X}^t \|_F ] \leq \mathcal{O} ({\overline{\mathbf{X}}}^4 \varphi (\Delta_1 ) ) <+\infty$. Clearly, J-orthogonal optimization problems are more challenging to converge compared to standard orthogonal optimization problems.

---

> > ### Comment · Reviewer_1LQB · 2024-11-20
> >
> > Thanks for your reply. I raised my scores to 6.

---

### Official Review · Reviewer_AV48 · 2024-11-02

**Soundness:** 2
**Presentation:** 1
**Contribution:** 2
**Rating:** 3
**Confidence:** 5

**Summary:**

The paper introduces a block coordinate descent framework named JOBCD to address optimization problems subject to J-orthogonality constraints. The authors propose two variants of JOBCD: GS-JOBCD, based on a Gauss-Seidel strategy, and VR-J-JOBCD, which incorporates variance reduction and a Jacobi strategy. The paper establishes the oracle complexity and strong limit-point convergence results for both algorithms. Extensive experiments demonstrate the effectiveness and efficiency of JOBCD.

**Strengths:**

The framework tailored for the specific structure of the J-orthogonality constraint is interesting, which updates on a low-dimensional space in each iteration. The proposed algorithms also exhibit good scalability. Additionally, the paper is theoretically grounded, and numerical experiments validate the effectiveness and efficiency of the framework.

**Weaknesses:**

1. The paper does not specify the motivations of the study, leaving the reader unclear about the overall impact. Specifically, it is noted in line 95 that "all the aforementioned methods solely identify critical points". However, the theory developed in this work only guarantees convergence to first-order stationary points as well. Additionally, the authors state the unconstrained multiplier correction Methods are "efficient", and the ADMM-based methods are "widely adopted". The discussion suggests that existing work appears to be good enough, and thus weakens the motivation of the authors' study. It would be beneficial to point out the disadvantages of the mentioned work, which are addressed by the proposed JOBCD.
2. The novelty of this work seems limited. An array of work has generalized the coordinate descent framework to Stiefel manifolds, e.g., [1,2,3,4], among which [4] also allows to update two rows per iteration. Additionally, the optimization problems on the symplectic Stiefel manifold have also been well-studied, e.g., [5]. To alleviate the concerns, the authors should clarify the challenges of generalizing the BCD method from the orthogonality constraint to the J-orthogonality constraint, and essentially, identify the difference between the symplecticity constraint [5] and the J-orthogonality constraint.
3. The readability should be significantly improved. For instance, all the algorithm names and matrix notation are formatted in bold throughout the manuscript, which makes the text overwhelming and departs from the article's main points. In addition, there are some typos (see Questions).

**Questions:**

1. The complexity developed in Theorem 4.6 hides the dependency on the problem dimension $n$. The proposed GS-JOBCD updates two rows per iteration (only $2/n$ of the variables), which will result in a trade-off between efficiency and convergence rate. Could the authors provide the dependency on $n$ explicitly in the main text?
2. The inequality (10) serves as an $n/2$-step extension of the inequality (3), and accordingly, the VR-J-JOBCD collects $n/2$ steps of JOBCD as one iteration. As stated by line 298, the inequality (3) is not adopted as the surrogate in stochastic settings, for which the presentation of (10) seems tedious.
3. Typos. In equality (6), the "$\in$" should be "$\subseteq$"; in line 420, the "Proposition 4.8" is not formulated.



If the response addresses the concerns, I would like to change the grade.


[1] Coordinate descent without coordinates: Tangent subspace descent on Riemannian manifolds. MOR, 2023

[2] A block coordinate descent method for nonsmooth composite optimization under orthogonality constraints. 2023

[3] Riemannian coordinate descent algorithms on matrix manifolds. ICML, 2024

[4] Randomized submanifold subgradient method for optimization over Stiefel manifolds. 2024

[5] Riemannian optimization on the symplectic Stiefel manifold. 2021

---

> ### Author Response · Authors · 2024-11-18
> **Rebuttal by Authors**
>
> Weaknesses 1.The paper does not specify the motivations of the study, leaving the reader unclear about the overall impact. Specifically, it is noted in line 95 that "all the aforementioned methods solely identify critical points". However, the theory developed in this work only guarantees convergence to first-order stationary points as well.
>
> **Response**:JOBCD leverages the structured constraints by utilizing a 2-dimensional CS decomposition, thereby obtaining stronger stationary points (as stated in Theorem 3.3 of our paper).
>
> Additionally, the authors state the unconstrained multiplier correction Methods are "efficient", and the ADMM-based methods are "widely adopted". The discussion suggests that existing work appears to be good enough, and thus weakens the motivation of the authors' study. It would be beneficial to point out the disadvantages of the mentioned work, which are addressed by the proposed JOBCD.
>
> **Response**:ADMM and UMCM are both highly efficient but infeasible methods, as they do not always ensure that the solutions lie within the constraint set. In contrast, our method is a feasible approach. In the revised version, we will emphasize that our method is a feasible approach for solving J-orthogonal problems, meaning that all iterates will remain within the feasible set. However, a drawback of our method is that it includes a certain degree of randomness.
>
> 2.The novelty of this work seems limited. An array of work has generalized the coordinate descent framework to Stiefel manifolds, e.g., [1,2,3,4], among which [4] also allows to update two rows per iteration. Additionally, the optimization problems on the symplectic Stiefel manifold have also been well-studied, e.g., [5]. To alleviate the concerns, the authors should clarify the challenges of generalizing the BCD method from the orthogonality constraint to the J-orthogonality constraint, and essentially, identify the difference between the symplecticity constraint [5] and the J-orthogonality constraint.
>
> **Response**:(1).None of the existing methods can solve the J-orthogonal optimization problem, whereas ours is the first block coordinate descent method that can effectively address this issue.
>
> (2).None of these methods employ variance reduction algorithms.
>
> (3).The popular structures of symplectic orthogonality and J-orthogonality are quite different. Our algorithm is constructed based on CS decomposition. Additionally, while symplectic orthogonality has fast projection algorithms, J-orthogonal optimization problems do not have such efficient projection methods.
>
> (a).symplectic orthogonality constraint：the set of $\mathbf{X}^\top \mathbf{J} \mathbf{X} = \mathbf{J}$ where $\mathbf{J}$=[0, $\mathbf{I}_n$ ;0,-$\mathbf{I}_n$ ] is fixed。
>
>
> (b).J-orthogonality constraint：the set of $\mathbf{X}^\top \mathbf{J} \mathbf{X} = \mathbf{J}$ where$\mathbf{J}$ is an $n \times n$ diagonal matrix with random $\pm$1s .
>
> (4) JOBCD incorporates the Jacobi strategy, but [2] does not.
>
> 3.The readability should be significantly improved. For instance, all the algorithm names and matrix notation are formatted in bold throughout the manuscript, which makes the text overwhelming and departs from the article's main points. In addition, there are some typos (see Questions).
>
> **Response**:To make the paper easier to read, we will distinguish various bolded elements by using different fonts.
>
> Questions:
> 1.The complexity developed in Theorem 4.6 hides the dependency on the problem dimension n. The proposed GS-JOBCD updates two rows per iteration (only 2/n of the variables), which will result in a trade-off between efficiency and convergence rate. Could the authors provide the dependency on n explicitly in the main text?
>
> **Response**: During the proof of the convergence rate, we have already obtained intermediate results related to n, and we will explicitly express n in the main text in subsequent revisions.
>
> 2.The inequality (10) serves as an n/2-step extension of the inequality (3), and accordingly, the VR-J-JOBCD collects n/2 steps of JOBCD as one iteration. As stated by line 298, the inequality (3) is not adopted as the surrogate in stochastic settings, for which the presentation of (10) seems tedious.
>
> **Response**:Equation (10) primarily demonstrates that the majorization function can be decomposed for parallel computation.
>
> 3.Typos. In equality (6), the "∈" should be "⊆"; in line 420, the " Proposition 4.8" is not formulated.
>
> **Response**:We simply pick an element from the global optimal set. This approach has been adopted in many papers. In the revised version, we will change “Proposition” to “Assumption” in line 420.

---

> > ### Comment · Reviewer_AV48 · 2024-11-25
> >
> > The reviewer thanks for the authors' reply.  The reviewer would like to maintain the score for the limited novelty and concerns.

---

### Official Review · Reviewer_zdQ7 · 2024-11-03

**Soundness:** 2
**Presentation:** 2
**Contribution:** 2
**Rating:** 3
**Confidence:** 4

**Summary:**

The paper considers solving an optimization problem over the set of J-orthogonal matrices (the set of $X^T J X = J$ where $J$ is an $n\times n$ diagonal matrix with $\pm 1$s), where the objective might consist of a finite sum of losses (i.e. empirical risk minimization) or a single loss. The paper proposes two methods for solving this problem: ** GS-JOBCD ** based on Gauss-Seidel strategy and its extension: **VR-J-JOBCD** based on variance reduction with Jacobi strategy.

The main technical innovation of the paper is based on the observation, that when only two rows are updated at once, it is possible to provide a closed form solution for the optimal update. The second algorithm, VR-J-JOBCD, can then apply this strategy on randomly chosen pairs of row indices in parallel.

**Strengths:**

The idea of using (block) coordinate descent for hyperbolic J-orthogonal manifold is novel as far as I know. Also the concept of an update that does pair-wise rotations coordinates is new. The paper also provides many numerical experiments.

**Weaknesses:**

* It is not clear if JOBCD represents a fundamentally new algorithmic framework or if it is a modification of existing block coordinate descent methods or Riem. coordinate descent tailored for J-orthogonal constraints. In particular, the work is very similiar to the work of (Ganzhao; A block coordinate descent method for nonsmooth composite optimization under orthogonality constraints). Although there is a comparison with this work, the main idea in both of these (pairwise index update) is the same.
* The convergence results are stated without enough detail, e.g., without discussing the individual constants.
* The algorithm needs the knowledge of the probability $p$ for the VR to work, this might not be realistic.

**Questions:**

* What is the impact of choosing suboptimal $p$ in the VR strategy?

---

> ### Author Response · Authors · 2024-11-18
> **Rebuttal by Authors**
>
> Weaknesses:
> 1.It is not clear if JOBCD represents a fundamentally new algorithmic framework or if it is a modification of existing block coordinate descent methods or Riem. coordinate descent tailored for J-orthogonal constraints. In particular, the work is very similiar to the work of (Ganzhao; A block coordinate descent method for nonsmooth composite optimization under orthogonality constraints). Although there is a comparison with this work, the main idea in both of these (pairwise index update) is the same.
>
> **Response**:
> (1) Previous studies were unable to solve the J-orthogonal optimization problem, and our paper is the first to effectively address this issue.
>
> (2) We are the first to use the CS decomposition to design a feasible method for solving the J-orthogonal optimization problem.
>
> (3) We employed a variance reduction strategy.
>
> (4) JOBCD incorporates the Jacobi strategy, but the work of (Ganzhao; A block coordinate descent method for nonsmooth composite optimization under orthogonality constraints) does not.
>
> 2.The convergence results are stated without enough detail, e.g., without discussing the individual constants.
>
> **Response**: Theorems 4.5, 4.6, 4.9 and 4.10 in the paper have clearly given corresponding complexity conclusions, which are easy to compare.
>
> (1) According to theorems 4.5 and 4.6, it is obvious that the Oracle complexity of VR-J-JOBCD is lower. To be specific, GS-JOBCD is linearly dependent on N,  while VR-J-JOBCD is linearly dependent on $ \sqrt{N}$. In addition, both have the same complexity for $\epsilon$.
>
> (2) According to theorems 4.9 and 4.10, it is obvious that VR-J-JOBCD makes full use of the finite sum structure of the problem, while GS-JOBCD does not, so the complexity of the GS-JOBCD algorithm is related to N, while VR-J-JOBCD is not.
>
> 3.The algorithm needs the knowledge of the probability p for the VR to work, this might not be realistic.
>
> **Response**: The value of p is merely a parameter that achieves the theoretically optimal complexity and is not the prior probability of the data. It is only related to the data sample size. Existing research on variance reduction algorithms also typically follows a similar setting.
>
> Questions:
> 4.What is the impact of choosing suboptimal p in the VR strategy?
>
> **Response**: Answer the same as “Weaknesses 3”

---

> > ### Comment · Reviewer_zdQ7 · 2024-11-25
> >
> > Thank you for the reply to my review. However, I keep my score unchanged as I still have concerns regarding novelty and clarity of the paper.

---

### Official Review · Reviewer_LCmg · 2024-11-03

**Soundness:** 2
**Presentation:** 3
**Contribution:** 2
**Rating:** 5
**Confidence:** 4

**Summary:**

This paper focuses on one special optimization problem with J-orthogonal constraints, a generalized version of orthogonality into signature matrix $J= X^TJX$. They provide a method based on the Block Coordinate Descent (BCD) with two variants: the Gauss-Seidel strategy (GS-JOBCD) and variance-reduced and Jacobi strategy (VR-J-JOBCD). They showed complexity and convergence analyses for both variants. They numerically tested their method on real-world and synthetic datasets.

**Strengths:**

Pros:

1. They provide a BCD algorithm that is specially designed for J-orthogonal constraints.

2. They offered complete optimality and convergence analyses.

3. The experiment result is interesting, but the baselines are slightly debatable. (Is there any other STOA in these certain applications?)

**Weaknesses:**

Cons:

1. The main concern is the significance of their applications where the reviewer does not find a special significance for a more general area.

2. When they claimed stronger stationary points than the baselines method due to their special design on the J-orthogonal constraints. They are using the standard ADMM and Unconstrained Multiplier Correction Method (UMCM), which are not necessarily tailored to their problem. Especially as far as the reviewer has known, it's wired to see that BCD significantly outperformed ADMM, since they can both guarantee convergence to a stationary point. In general, BCD is not a lightweight algorithm, the experience is ADMM usually more stable and maybe faster. Could the author further clarify the following questions:

a. Could the authors please provide a more detailed theoretical comparison of the convergence properties of their BCD method vs ADMM specifically for problems with J-orthogonal constraints?

b. Could the author please provide more empirical convergence plots comparing BCD and ADMM across different problem instances to better understand when/why BCD outperforms ADMM?

c. Could the authors please implement a version of ADMM that is more tailored to J-orthogonal constraints as a stronger baseline? Or any other SOTA methods can be applied.

d. Could the authors please discuss potential reasons why BCD may be better suited for J-orthogonality constraints compared to general ADMM? For example, provide the theoretical comparison of the complexity bounds between their methods and the baselines.

3. For the complexity part, it would be suggested to add an analysis for baselines to explain why it outperforms the baselines so much in certain settings.

a. Is there any detailed complexity analysis for the baseline methods (ADMM and UMCM) in addition to their proposed methods?

b. Could the author please discuss specific problem characteristics or settings where their methods are expected to have lower complexity and verify these expectations empirically?

**Questions:**

Please address the questions in the weakness part.

---

> ### Author Response · Authors · 2024-11-19
> **Rebuttal by Authors**
>
> 1. The main concern is the significance of their applications where the reviewer does not find a special significance for a more general area.
>
> **Response:**
>
> J-Orthogonality constraint problems defines an optimization framework that is fundamental to a wide range of models in statistical learning and data science, including hyperbolic structural probe problem[1;2], and ultrahyperbolic knowledge graph embedding [3].
>
> [1] Probing bert in hyperbolic spaces. ICLR, 2021.
>
> [2] A structural probe for finding syntax in word representations. NAACL-HLT,2019
>
> [3] Ultrahyperbolic knowledge graph embeddings. SIGKDD 2022
>
> 2. Could the authors please provide a more detailed theoretical comparison of the convergence properties of their BCD method vs ADMM specifically for problems with J-orthogonal constraints?
>
> **Response:**
>
>
> | Algorithm | Feasible method | Complexity per iteration | Oracle complexity | Reducible variance |
> | --- | --- | --- | --- | --- |
> | UMCM | &times; | $\mathcal{O}(\tfrac{1}{n^2})$ | $\mathcal{O}(\tfrac{1}{\epsilon})$ | &times; |
> | ADMM | &times; | $\mathcal{O}(\tfrac{1}{n^2})$ | $\mathcal{O}(\tfrac{1}{\epsilon})$ | &times; |
> | GS-JOBCD | &#x2713; | $\mathcal{O}(\tfrac{1}{n})$ | $\mathcal{O}(\tfrac{1}{\epsilon})$ | &#x2713; |
> | VR-J-JOBCD | &#x2713; | $\mathcal{O}(\tfrac{1}{n^2})$ | $\mathcal{O}(\tfrac{1}{\epsilon})$ | &#x2713; |
>
> (1) ADMM only satisfies the KKT condition in Lemma 3.1 when t is sufficiently large（possibly-infinite）, whereas JOBCD satisfies the constraint for any iteration point.
>
> (2) Both ADMM and JOBCD converge to the KKT conditions at the same rate of $\mathcal{O}(\tfrac{1}{\epsilon})$. The convergence proof for ADMM can be found in THEOREM 4.7 of [1], and the proof process is consistent.
>
> (3) JOBCD employed a variance reduction strategy.
>
> [1] Parallelizable algorithms for optimization problems with orthogonality constraints. SIAM Journal on Scientific Computing 2019
>
> 3．Could the authors please implement a version of ADMM that is more tailored to J-orthogonal constraints as a stronger baseline? Or any other SOTA methods can be applied.
>
> **Response:**
>
> We kindly refer the reviewer to the design process of the UMCM algorithm. In our design, we have thoroughly considered the KKT conditions for the J-orthogonal optimization problem based on the standard UMCM algorithm. We have already employed the Adagrad method, which is a first-order momentum-based method and is currently a SOTA optimization technique.
>
> 4.Could the author please provide more empirical convergence plots comparing BCD and ADMM across different problem instances to better understand when/why BCD outperforms ADMM?
>
> **Response:**
>
> For more experimental comparisons, we refer the reviewer to Section F.5 of the original manuscript. In this section, we provide a detailed analysis of the choice of penalty parameters for ADMM and have selected the optimal settings for our experiments.

---

> > ### Comment · Reviewer_LCmg · 2024-11-21
> > **Response to the author**
> >
> > The reviewer thanks the author for the response.
> > Does that mean: the theoretical convergence is still the same for both ADMM and JOBCD, but you empirically found it's more effective after fine-tuning the parameter for ADMM?

---

> ### Author Response · Authors · 2024-11-22
> **Response to Reviewer LCmg**
>
> Yes, we did find that in our experiments. Indeed, the well-known ADMM is a viable algorithm. However, since projection techniques for J-orthogonal problems have not been adequately studied yet, we do not recommend it at this stage. ADMM cannot guarantee that the final solution will satisfy the J-orthogonal constraint. In the experiments conducted in this paper, we needed to tune its parameters separately for different problems or datasets to achieve satisfactory results.

---

> > ### Comment · Reviewer_LCmg · 2024-11-24
> >
> > The reviewer thanks the author for the clarification. However, the reviewer wants to keep the score for the significance concern.

---

### Meta-Review · Area_Chair_iELc · 2024-12-14

**Metareview:**

This paper studies block coordinate descent method (BCD) for optimization with J-orthogonality constraint. The authors proposed a new method that minimizes a majorizing surrogate of the block restricted objective, which is obtained by sampling two rows and formulating the subproblem accordingly. The reviewers found that the paper is not well written, and the novelty is limited, as it shares significant similarity with existing work.

**Additional Comments On Reviewer Discussion:**

Discussed novelty and clarity. The reviewers were not convinced.

---

### Decision · Program_Chairs · 2025-01-22

Reject